# A fresh look at the pre-industrial air-sea carbon flux using the alkalinity budget

Alban Planchat<sup>1,2,3</sup>, Laurent Bopp<sup>1</sup>, and Lester Kwiatkowski<sup>4</sup>

<sup>1</sup>LMD-IPSL, CNRS, Ecole Normale Supérieure/PSL Res. Univ, Ecole Polytechnique, Sorbonne Université, Paris, 75005, France

<sup>2</sup>Climate and Environmental Physics, Physics Institute, University of Bern, Bern, Switzerland

<sup>3</sup>Oeschger Centre for Climate Change Research, University of Bern, Bern, Switzerland

<sup>4</sup>LOCEAN, Sorbonne Université-CNRS-IRD-MNHN, Paris, 75005, France

**Correspondence:** Alban Planchat (alban.planchat@unibe.ch)

Abstract. Disparities between observational and model-based estimates of the ocean carbon sink persist, highlighting the need for improved understanding and methodologies to reconcile differences in both magnitude and trends over recent decades. A potential key source of uncertainty lies in the pre-industrial air–sea carbon flux, which is essential for isolating the anthropogenic component from observations. This flux, thought to result globally from an imbalance between riverine discharge and sediment burial of carbon, remains highly uncertain, limiting the confidence in impactful applications such as the Global Carbon Budget (GCB). In this study, we present a new theoretical framework that enables direct estimation of the riverine/burial-driven pre-industrial carbon outgassing using both carbon and alkalinity budgets. This approach is validated with a series of ocean biogeochemical simulations, which also highlight the main factors influencing its regional distribution. We then demonstrate the utility of the framework through two proof-of-concept applications. The first revisits the pre-industrial riverine/burial-driven air–sea carbon flux using existing carbon and alkalinity budgets, offering a simple method for reassessment as these budgets are updated. The second application leverages sensitivity simulations to construct a composite simulated estimate that aligns with both carbon and alkalinity budgets to assess the regional distribution of the pre-industrial riverine/burial-driven air–sea carbon flux. This approach is well suited for model intercomparisons, enabling an efficient reassessment of regional flux patterns and helping to reduce biases related to ocean model physics or biogeochemical parameterizations.

#### 15 1 Introduction

Accurately estimating the anthropogenic carbon sink in the ocean is crucial for gaining a deeper understanding of the underlying mechanisms, and is a prerequisite for projecting its future evolution and the climate response to future emissions scenarios (Canadell et al., 2021). This anthropogenic carbon flux is currently assessed with yearly updates by the Global Carbon Budget (GCB; Friedlingstein et al., 2024), using both observational products and model simulations employing Global Ocean Biogeochemical Models (GOBMs). However, reconstructions derived from surface ocean  $pCO_2$  data – currently the main observation-based approach – tend to yield higher estimates than models, both globally and regionally. This mismatch

has grown steadily since the early 2000s, now reaching a 10–20 % difference over the past decade (e.g. Hauck et al., 2020; DeVries et al., 2023; Gruber et al., 2023; Friedlingstein et al., 2024).

Recent studies have begun to investigate the origins of discrepancies in both the magnitude and trend of observational versus model-based estimates of the ocean anthropogenic carbon sink. Analyses of GOBM-derived estimates (Terhaar et al., 2024) and  $pCO_2$ -based products (Ford et al., 2024) point to multiple sources of uncertainty, including methodological differences, model biases, and data limitations. On the modeling side, GOBMs have been shown to underestimate the global sink magnitude (Terhaar et al., 2022), as well as decadal variability, especially in the Southern Ocean (Mayot et al., 2023, 2024). On the observational side, the sparse and uneven spatial and temporal coverage of surface ocean  $pCO_2$  measurements remains a major limitation (Hauck et al., 2023; Landschützer et al., 2023; Dong et al., 2024). While the causes of these mismatches are likely multifaceted, one less-discussed contributor is the uncertainty surrounding the pre-industrial air–sea carbon flux and its influence on  $pCO_2$ -based estimates.

The net air-sea carbon flux derived from  $pCO_2$ -based data encompasses both anthropogenic and natural components. The natural component originates, at the global scale, from the balance between riverine discharge and the burial of organic matter (OM) and calcium carbonate (CaCO<sub>3</sub>). As these external fluxes together represent a net source of carbon for the ocean, they result in a net carbon outgassing at steady state during the pre-industrial era. Consequently, assessing the anthropogenic carbon flux through observations requires determining the pre-industrial riverine/burial-driven air-sea carbon flux and its spatial distribution (e.g. Hauck et al., 2020; Friedlingstein et al., 2024).

Assessing this outgassing carbon flux remains highly uncertain, with estimates ranging from 0.23 to 0.78 PgC yr<sup>-1</sup> (Aumont et al., 2001; Jacobson et al., 2007; Resplandy et al., 2018; Lacroix et al., 2020; Regnier et al., 2022), depending on the modeling approach used to derive them (forward or inverse) and estimates of riverine and burial fluxes (see Table E1). Specifically, the most recent estimate of  $0.65 \pm 0.30$  Pg Cyr<sup>-1</sup> is that used in the latest GCB release (Regnier et al., 2022; Friedlingstein et al., 2024, Table 1).

The spatial distribution of this riverine/burial-driven air-sea carbon flux is also highly uncertain. It strongly depends on the assumptions and methods used to assess it, including how sediment burial processes are represented, and both the magnitude and characteristics of riverine carbon inputs – particularly the balance between organic and inorganic forms, as well as the lability of terrestrial organic matter – (see Table E1). The earlier estimate, derived from a modeling analysis (Aumont et al., 2001) distributed this flux as follows: 49 % in the southern region, 25 % in the inter-tropical region, and 26 % in the northern region. In contrast, the most recent estimate, currently used in the GCB and also based on a modeling study (Lacroix et al., 2020, Table 1), suggests a very different partitioning: 14 %, 64 % and 22 %, respectively, reshaping our understanding of the regional distribution of this flux.

Uncertainties in estimating the riverine/burial-driven pre-industrial outgassing may contribute to the persistent discrepancies between observation-based and model-derived estimates of the anthropogenic ocean carbon sink, both globally and regionally (Friedlingstein et al., 2024, their Fig. 11 and 14). These disparities have fluctuated over time, largely in response to stepwise adjustments made by the GCB team following new reassessments of the magnitude and spatial distribution of this flux in the literature (Fig. 1, Table 1). For instance, a substantial decrease in the global estimate of the pre-industrial outgassing from

**Table 1.** Review of the air-sea carbon outgassing from riverine/burial fluxes of carbon and Alk used in the GCBs. Both the global values and their regional distribution are presented, along with the associated references.

|              | Air-sea carbon o               | utgassing from | n riverine/burial fluxes of carbo | on and Alk |               |           |  |
|--------------|--------------------------------|----------------|-----------------------------------|------------|---------------|-----------|--|
|              | Global                         |                | Distribution                      |            |               |           |  |
| GCB          | Pafaranaa(s)                   | CtC vm 1       | D. C                              | GtC yr-1 ( | %)            |           |  |
|              | Reference(s)                   | GtC yr-1       | Reference                         | South      | Inter-tropics | North     |  |
| 2023 to 2024 | Regnier et al. (2022)          | $0.65\pm0.3$   | Lacroix et al. (2020)             | 0.09 (14)  | 0.42 (64)     | 0.14 (22) |  |
| 2022         | Regnier et al. (2022)          | 0.65           | Aumont et al. (2001)              | 0.32 (49)  | 0.16 (25)     | 0.17 (26) |  |
| 2020 to 2021 | Jacobson et al. (2007)         | 0.61           | Aumont et al. (2001)              | 0.30 (49)  | 0.15 (25)     | 0.16 (26) |  |
|              | & Resplandy et al. (2018)      | 0.01           | Aumont et al. (2001)              | 0.50 (47)  | 0.13 (23)     | 0.10 (20) |  |
| 2018 to 2019 | Resplandy et al. (2018)        | 0.78           | Aumont et al. (2001)              | 0.38 (49)  | 0.19 (25)     | 0.20 (26) |  |
| 2013 to 2017 | 17 Jacobson et al. (2007) 0.45 |                | Not considered at that time       |            |               |           |  |
| 1959-2011    | 1 Not considered at that time  |                |                                   |            |               |           |  |

Figure 1. Schematic illustration of how a bias in evaluating the pre-industrial ocean carbon outgassing affects the assessment of the anthropogenic carbon flux based on  $pCO_2$ -products. A downward revision of the pre-industrial outgassing would decrease anthropogenic carbon flux estimates based on  $pCO_2$ -products, while an upward revision would increase it. This effect applies both globally and regionally.

2019 to 2020 contributed to a notable narrowing of the global observation—model gap. More recently, from 2022 to 2023, a redistribution of the flux between regions, from the southern region to the tropics, led to a reduced southern hemisphere bias and a compensating increase in the inter-tropical discrepancy.

60

Enhancing our understanding of the riverine/burial-driven air-sea carbon flux is critical to achieving more precise estimates of the anthropogenic carbon flux and its distribution from data-driven assessments. Numerical models hold great promise in addressing this challenge, particularly for estimating the spatial distribution of the flux. However, at present, the representation of the pre-industrial air-sea carbon flux remains uncertain in inter-model comparison exercises like CMIP6 (the 6th phase of the Coupled Model Intercomparison Project, Eyring et al., 2016) and the 2024 GCB (Friedlingstein et al., 2024), likely due to differences in model setups and various/incomplete representations of sediment burial and riverine discharge (Terhaar et al.,

2024). The magnitude of this global net flux ranges from -0.73 to 0.38 PgC yr<sup>-1</sup>, while its inter-hemispheric gradient, defined as the difference between its values in the northern and southern hemispheres, ranges from -0.09 to 0.82 PgC yr<sup>-1</sup> (Fig. 2a).

The methods employed thus far to estimate the riverine/burial-driven air-sea carbon flux at the global scale mostly rely on closing the ocean carbon budget. However, they often exhibit limitations in addressing the ocean alkalinity budget. Alkalinity (Alk), defined as the excess of proton acceptors over proton donors, or of positive conservative charges over negative ones, plays a pivotal role in driving air-sea carbon exchanges, which are strongly dependent on the relative balance between Alk and dissolved inorganic carbon (DIC; e.g. Humphreys et al., 2018). Similar to carbon, the Alk budget is controlled by both sources and sinks at the boundaries of the oceanic domain (Middelburg et al., 2020). Conventionally, it is hypothesized that the global Alk inventory has been conserved during the pre-industrial era, with the burial of CaCO<sub>3</sub> balancing the Alk riverine discharge (e.g. Revelle and Suess, 1957; Aumont et al., 2001; Planchat et al., 2023).

Nonetheless, the hypothesis of an imbalanced Alk budget during the pre-industrial era remains plausible, based on estimates of global Alk sources and sinks (e.g. Milliman, 1993; Middelburg et al., 2020). Paleoclimatic studies suggest that such an imbalance could arise from additional CaCO<sub>3</sub> burial (e.g. Cartapanis et al., 2018) or from a carbonate compensation mechanism involving biological processes alongside riverine inputs (Boudreau et al., 2018). Unlike carbon, Alk is not exchanged with the atmosphere, so balancing its budget depends on processes acting over longer timescales, particularly through interactions with the continents (e.g. erosion) and marine biogeochemistry (e.g. sediment dynamics). As a result, Alk budget balancing is slow, yet interactive with the carbon cycle through changes in atmospheric CO<sub>2</sub> and ocean acidity (Hain et al., 2014). An imbalance in the Alk budget would induce an additional air—sea carbon flux beyond that directly inferred from the ocean carbon budget, ultimately resulting in a non-conserved global ocean carbon inventory.

Here, we take a fresh look at the pre-industrial air–sea carbon flux by introducing a new theoretical framework that enables direct estimation of the riverine/burial-driven pre-industrial carbon outgassing, based on both carbon and Alk budgets. We validate this approach using a suite of ocean biogeochemical simulations, which also help identify the key drivers of its regional distribution. We then demonstrate the utility of this framework through two proof-of-concept applications: (i) revisiting the global magnitude of the pre-industrial riverine/burial-driven air—sea carbon flux using existing carbon and Alk budgets; and (ii) reassessing its regional distribution using sensitivity simulations to construct a composite simulated estimate consistent with both budgets.

#### 2 Methods

85

In this study, we use 'steady-state' to refer to the temporal stability of the globally integrated air—sea carbon flux. We describe the carbon and Alk budgets as 'balanced' or 'imbalanced' according to whether fluxes into and out of the ocean are quantitatively balanced. An imbalanced budget drives a deviation in the global inventory: a positive (resp. negative) imbalance leads to an increase (resp. decrease) in the global inventory. We refer to a 'deviation' in a given variable when the system is in steady-state, but a persistent trend is identified for that variable (e.g. the global carbon inventory). In contrast, we use the term 'drift'

**Figure 2.** Pre-industrial air-sea carbon flux from models. The globally integrated pre-industrial flux (x-axis) and its interhemispheric gradient (y-axis) are indicated for (a) CMIP6 Earth system models (ESMs) and GCB Global Ocean Biogeochemical Models (GOBMs), as well as for (b) the NEMO-PISCES sensitivity simulations. (a) The 15 CMIP6 ESMs (10 GCB GOBMs; see Appendix B1) are plotted with red squares (orange circles). The black square and circle refer to the CMIP6 and GCB ensemble means. The CMIP6 and GCB ensemble ranges (line), mean (major tick) and quartiles (minor ticks) are respectively displayed to the top and right in red and orange. The star refers to the reference value used in the GCB 2024 (Table 1; Friedlingstein et al., 2024). The inter-hemispheric air-sea carbon flux gradient is defined as the difference between its values in the northern and southern hemispheres (Sect. 2.1.2).

when a trend in a variable disrupts the steady-state. Unless otherwise stated, all simulations and results refer to pre-industrial conditions.

## 100 2.1 Theoretical framework

#### 2.1.1 Governing equation of the pre-industrial riverine/burial-driven air-sea carbon flux

The collection of surface ocean  $pCO_2$  data, and associated statistical methods, only allow for the direct reconstruction of the contemporary air-sea carbon flux ( $F_{\text{cont.}}^{C, \text{ air-sea}}$ ), which encompasses both an anthropogenic ( $F_{\text{ant.}}^{C, \text{ air-sea}}$ ) and a natural ( $F_{\text{nat.}}^{C, \text{ air-sea}}$ ) component (e.g. Hauck et al., 2020), as follows:

105 
$$F_{\text{cont.}}^{\text{C, air-sea}} = F_{\text{ant.}}^{\text{C, air-sea}} + F_{\text{nat.}}^{\text{C, air-sea}}$$
 (1)

Figure 3. Schematics of the theoretical framework introduced in this manuscript. (a) Conceptual representation of a process X (e.g.  $CaCO_3$  burial), which affects carbon  $(F^{C, X})$  and Alk  $(F^{Alk, X})$ , thereby induces, at steady state, a compensating carbon flux  $(F^{C, air-sea}(X))$  and the resulting deviations in carbon and Alk inventories  $(D^{C/Alk}(X))$ . The equilibrium of the Alk:DIC pair with atmospheric  $CO_2$  is represented by a solid grey line. Carbon (Alk) fluxes are represented through solid (dashed) arrows. (b) Schematic diagram of carbon and Alk budgets by ocean region.  $F^{C/Alk}$  and  $T^{C/Alk}$  respectively refer to the total external fluxes (directed into the ocean) and to the northward transport of carbon and Alk.  $D^{C/Alk}$  corresponds to the regional carbon and Alk inventory deviations in each basin. S, I and N refer to the different ocean regions, respectively the southern hemisphere, the inter-tropical zone, and the northern hemisphere.

where positive fluxes are directed into the ocean (consistent throughout this manuscript). Therefore, it is crucial to determine the natural component to extract the anthropogenic carbon flux from pCO<sub>2</sub>-based products. Within the anthropogenic carbon flux, we incorporate the perturbation of the natural carbon flux in response to climate change, ensuring that  $F_{\rm ant.}^{\rm C, \ air-sea}$  fully reflects the carbon sink resulting from all human-induced disturbances (e.g. Hauck et al., 2020). Accordingly,  $F_{\rm nat.}^{\rm C, \ air-sea}$  is directly defined as the riverine/burial-driven pre-industrial air-sea carbon flux ( $F_{\rm riv./bur.}^{\rm C, \ air-sea}$ ), i.e.:

$$F_{\text{nat.}}^{\text{C, air-sea}} = F_{\text{riv./bur.}}^{\text{C, air-sea}}$$
(2)

The anthropogenic carbon flux can then be derived from  $pCO_2$ -based data as follows:

$$F_{\text{ant.}}^{\text{C, air-sea}} = F_{\text{cont.}}^{\text{C, air-sea}} - F_{\text{riv./bur.}}^{\text{C, air-sea}}$$
(3)

Assuming a steady-state pre-industrial air—sea carbon flux and a balanced Alk budget, the global riverine/burial-driven air—sea carbon flux can be directly inferred by closing the ocean carbon budget (e.g. Regnier et al., 2022):

$$F_{\text{riv./bur.}}^{\text{C, air-sea}} + F^{\text{C, riv./bur.}} = 0$$

$$\tag{4}$$

with:

110

$$F^{C, \text{ riv./bur.}} = F^{C, \text{ riv.}} + F^{C, \text{ bur. org.}} + F^{C, \text{ bur. inorg.}} + F^{C, \text{ minor components}}$$
(5)

where 'riv.' stands for 'riverine discharge', 'bur. org.' for 'OM burial', 'bur. inorg.' for 'CaCO<sub>3</sub> burial', and 'minor components' encompass other minor external fluxes, such as carbon release by mid-ocean ridges and groundwater discharge. Thus, assuming

a conserved pre-industrial global Alk inventory, the riverine/burial-driven air-sea carbon flux is the opposite of the riverine and burial fluxes of carbon:

$$F_{\text{riv./bur.}}^{\text{C, air-sea}} = -F^{\text{C, riv./bur.}}$$
 (6)

However, when considering a pre-industrial imbalanced Alk budget (i.e. a non-conserved global ocean Alk inventory), it becomes necessary to account for the Alk budget explicitly to infer the riverine/burial-driven air-sea carbon flux:

$$F^{\text{Alk, riv./bur.}} = F^{\text{Alk, riv.}} + F^{\text{Alk, bur. org.}} + F^{\text{Alk, bur. inorg.}} + F^{\text{Alk, minor components}}$$
(7)

where 'minor components' encompass this time other minor external fluxes such as anaerobic processes, silicate weathering, and groundwater discharge. Importantly, under the assumption of a steady-state system – that is, with a stable air—sea carbon flux –, any imbalance in the Alk budget induces a compensating carbon flux. To estimate this flux, we extend the conceptual framework introduced by Humphreys et al. (2018) by utilizing a phase diagram (Alk, DIC) in the form of an Alk and DIC flux diagram, while operating on a global scale (Fig. 3a). For any flux affecting carbon and/or Alk, it is possible to derive an air-sea carbon flux and the associated global carbon and Alk inventory imbalances. This approach relies on the equilibrium relationship between the Alk:DIC pair and atmospheric CO<sub>2</sub>. At global scale, at steady-sate, any deviation in Alk is directly proportional to a DIC anomaly, and this proportionality coefficient can be estimated with high precision, as follows:

135 
$$Q_{\text{inv}} \simeq \frac{\text{Alk}}{3 \cdot \text{Alk} - 2 \cdot \text{DIC}}$$
 (8)

where  $Q_{\rm inv}$ , as previously defined in Planchat et al. (2023), represents the inverse of the 'isocapnic quotient' approximation introduced by (Humphreys et al., 2018, see Appendix A). In this study,  $Q_{\rm inv}$  is defined based on the mean surface values of Alk and DIC. In the case of a steady-state air-sea carbon flux (see Fig. B2), every external process X (e.g. riverine discharge) that exerts an impact on carbon ( $F^{\rm C}$ , X) and/or Alk ( $F^{\rm Alk}$ , X), results in a global imbalance, shifting the surface ocean away from equilibrium with the atmosphere. Specifically, this requires an air-sea carbon flux ( $F^{\rm C,air-sea}(X)$ ; Fig. 3a) to maintain global equilibrium with respect to the atmospheric  ${\rm CO}_2$ . This also leads to deviations in global carbon and Alk inventories ( $D^{\rm C}$  and  $D^{\rm Alk}$ , respectively). In summary, for any given process X, we can define:

$$\begin{cases}
F^{C, \text{ air-sea}}(X) = F^{Alk, X} \cdot Q_{\text{inv}} - F^{C, X} \\
D^{C}(X) = F^{Alk, X} \cdot Q_{\text{inv}} \\
D^{Alk}(X) = F^{Alk, X}
\end{cases} \tag{9}$$

Applying this theoretical framework to the total external carbon and Alk fluxes ( $F^{C, riv./bur.}$  and  $F^{Alk, riv./bur.}$ , respectively), we can deduce the riverine/burial-driven air-sea carbon flux ( $F^{C, air-sea}_{riv./bur.}$ ) and the respective deviations in global carbon and Alk inventories as follows:

$$\begin{cases}
F_{\text{riv./bur.}}^{\text{C, air-sea}} = F^{\text{Alk, riv./bur.}} \cdot Q_{\text{inv}} - F^{\text{C, riv./bur.}} \\
D^{\text{C}} = F^{\text{Alk, riv./bur.}} \cdot Q_{\text{inv}} \\
D^{\text{Alk}} = F^{\text{Alk, riv./bur.}}
\end{cases} (10)$$

It is worth noting that this general expression also applies to the specific case where the global Alk inventory is conserved  $(F^{\text{Alk, riv./bur.}} = 0)$ .

#### 2.1.2 Regional proxies for the spatial distribution of the pre-industrial riverine/burial-driven air-sea carbon flux

A direct relationship between carbon and Alk fluxes and the net global air—sea carbon flux can be established under the assumption of a steady-state air-sea carbon flux. However, this approach does not apply directly at the regional scale, where ocean circulation transports both Alk and DIC, and biogeochemical processes also generate regional sources and sinks.

## The concept of the regional carbon: Alk budget imbalance

150

160

To gain a deeper understanding of the factors shaping the spatial distribution of the riverine/burial-driven air-sea carbon flux, we expand upon the theoretical framework previously outlined for the global scale (refer to Section 2.1.1) and adapt it as a proxy for application at regional scales. This is essential to understanding the extent to which specific regional carbon:Alk budget imbalances can drive the global air-sea carbon flux as well as deviations in carbon and Alk inventories.

The air-sea carbon flux calculated by applying Eq. 10 to the riverine and burial fluxes of a specific region can only be considered a potential air-sea carbon flux, i.e. a capacity to generate such a flux at the global scale, without any guarantee that it fully occurs within the same region. Due to ocean circulation and the associated transport of carbon and Alk, regional carbon:Alk budget imbalances in riverine and burial fluxes explain the regional distribution of the drivers of the global air-sea carbon flux. However, they only partially explain the regional distribution of the flux itself.

## The inter-hemispheric flux gradient and transport

Understanding the spatial distribution of the pre-industrial riverine/burial-driven air-sea carbon flux is crucial for understanding the biases between observational and model-based estimates of the anthropogenic carbon sink. Yet, ocean circulation and carbon pumps within the ocean induce an asymmetry in the ocean on either side of the Intertropical Convergence Zone (ITCZ), which serves as an inter-hemispheric transport barrier (e.g. Murnane et al., 1999; Aumont et al., 2001; Resplandy et al., 2018). To assess the significance of this asymmetry on the air-sea carbon flux, particularly its components associated with riverine and burial fluxes, we provide two metrics for large-scale inter-hemispheric fluxes: (i) the inter-hemispheric air-sea carbon flux gradient (*G*), which is defined as the integrated net flux north of 20°N ( $F_N^{\rm C}$ , air-sea) minus that south of 20°S ( $F_S^{\rm C}$ , air-sea):

$$G = F_N^{C, \text{ air-sea}} - F_S^{C, \text{ air-sea}}$$

$$\tag{11}$$

and (ii) the inter-hemispheric ocean transport of carbon ( $T^{\rm C}$ ) and Alk ( $T^{\rm Alk}$ ), both directed northward, defined as the mean transport between 20°N ( $T_N^{\rm C/Alk}$ ) and 20°S ( $T_S^{\rm C/Alk}$ ):

175 
$$T^{\text{C/Alk}} = \frac{1}{2} \cdot \left( T_N^{\text{C/Alk}} + T_S^{\text{C/Alk}} \right)$$
 (12)

These two metrics rely on the subdivision of the ocean into two poleward basins, one south of  $20^{\circ}$ S and the other north of  $20^{\circ}$ N, separated by an intertropical basin (Fig. 3b and see Appendix C). A decomposition of the inter-hemispheric air-sea carbon flux gradient (G) into components associated with carbon transport and with riverine and burial processes is provided in Appendix D2.

Table 2. Summary of the NEMO-PISCES sensitivity simulations with a short description (see Sect. B2.1 and Table B1 for more details).

| Simulation    | Description                                                                                                                  |            |
|---------------|------------------------------------------------------------------------------------------------------------------------------|------------|
| std           | Standard (riverine discharge, as well as OM and CaCO <sub>3</sub> burial simulated)                                          | Balanced   |
| norivbur      | No external fluxes of carbon and Alk, except air-sea carbon fluxes                                                           | Balanced   |
| rivref        | Refractory organic riverine discharge                                                                                        | Balanced   |
| rivorg        | Fully organic riverine discharge                                                                                             | Balanced   |
| rivinorg      | Fully inorganic riverine discharge                                                                                           | Balanced   |
| riv1p5        | Riverine discharge of carbon and Alk multiplied by 1.5                                                                       | Balanced   |
| nosed-resto   | No OM and CaCO <sub>3</sub> burial, but restoration of the Alk content                                                       | Balanced   |
| nosed-diseq   | No OM and CaCO <sub>3</sub> burial                                                                                           | Imbalanced |
| atlpac        | Constrained balance of extra CaCO <sub>3</sub> burial/dissolution between the deep Atlantic/Pacific                          | Balanced   |
| atlpac-diseq  | $Constrained\ imbalance\ of\ extra\ CaCO_3\ burial/dissolution\ between\ the\ deep\ Atlantic/Pacific\ (-0.10\ PgC\ yr^{-1})$ | Imbalanced |
| tropics-diseq | Constrained extra CaCO <sub>3</sub> burial in the shallow tropics (-0.10 PgC yr <sup>-1</sup> )                              | Imbalanced |

### 180 2.2 Model and Simulations

190

195

# 2.2.1 Model and configuration

As part of the NEMO (Nucleus for European Modelling of the Ocean) suite of models, we used here the marine biogeochemical model PISCES (Pelagic Interactions Scheme for Carbon and Ecosystem Studies) to take a fresh look at the pre-industrial air–sea carbon flux. This involved a comprehensive consideration of both the carbon and Alk budgets, with a specific focus on external fluxes, notably CaCO<sub>3</sub> burial. While globally resembling PISCES-v2, as detailed in Aumont et al. (2015) and utilized in IPSL-CM6A-LR (Boucher et al., 2020), we introduced two key modifications in PISCES: (i) an adjustment to the N-fixation parameterization, following Bopp et al. (2022), and (ii) an adaptation of the burial fraction of CaCO<sub>3</sub> to balance the Alk budget and conserve the global Alk inventory without necessitating an Alk restoring scheme – periodically restoring the global Alk inventory to a reference value by adding/removing the required amount, either uniformly or in a weighted manner (see Planchat et al., 2023, their Appendix A2 for details). Our simulations were conducted offline using a tripolar ORCA (orthogonal curvilinear ocean mesh) grid with a nominal resolution of 2° and included 30 vertical levels. The ocean physics were derived from pre-industrial simulations of IPSL-CM5A-LR (Dufresne et al., 2013, based on NEMOv3.2), with a repeated 500-yr period, and a fixed and homogeneous atmospheric CO<sub>2</sub> concentration of 284 ppm at the ocean surface.

To ensure model stability and attainment of a steady state (i.e. stable air-sea carbon flux; e.g. Orr et al., 2017, see Fig. B2), all simulations presented below used the same initial conditions and have been run 2550 yr after an initial 500-yr spin-up using the standard configuration (Sect. 2.2.2). We calculated the carbon and Alk budgets related to their associated external sources/sinks using data from the last 50 yr of the simulations. The carbon and Alk inventory deviations were estimated through linear regression over the same period.

#### 2.2.2 Standard simulation (std) and its riverine/burial component

The standard simulation (referred to as 'std'), based on the standard configuration described above, involves carbon and Alk riverine supply as well as OM and CaCO<sub>3</sub> burial. Riverine supply of carbon and Alk is based on output from the Global Erosion Model (GEM) of Ludwig et al. (1996) and considers both inorganic and organic carbon riverine discharge (0.37 and 0.14 PgC yr<sup>-1</sup>, respectively). Carbon and Alk are added at river mouths using a monthly climatology that is applied recursively. The inorganic fraction is added as bicarbonate ions, thus affecting both DIC and Alk in a similar manner. The organic fraction is assumed to be fully labile and remineralizes instantaneously at the river mouth, thus impacting only DIC. This simulation also includes the burial of OM and CaCO<sub>3</sub> produced by pelagic organisms, which are exported to the ocean interior and only partially remineralized or dissolved in the water column and at the seafloor (e.g. Planchat et al., 2023). These combined fluxes constitute the riverine and burial fluxes (Eq. 5 and 7), which, as introduced in Sect. 2.1.1, lead to the riverine/burial-driven air-sea carbon flux.

To isolate the riverine/burial-driven component of the air-sea carbon flux, a simulation without riverine and burial fluxes was conducted (referred to as 'norivbur'), simulating only the component of the flux associated with the ocean carbon pumps. Indeed, while at the global scale, the net air-sea carbon flux directly corresponds to the riverine/burial-driven air-sea carbon flux (Eq. 2), at the regional scale (N, S, or I, Fig. 3b), the air-sea carbon flux (F<sup>C</sup><sub>nat</sub>. air-sea) can be decomposed at first approximation into two components: one internal component linked to the functioning of the ocean carbon pumps (F<sup>C</sup><sub>pumps</sub> air-sea), and a boundary component associated with the riverine and burial fluxes (F<sup>C</sup><sub>riv</sub>/bur.) – our primary focus – :

$$F_{\text{nat., }N/S/I}^{\text{C, air-sea}} = F_{\text{pumps, }N/S/I}^{\text{C, air-sea}} + F_{\text{riv./bur., }N/S/I}^{\text{C, air-sea}}.$$
(13)

Subsequently, by taking the difference between the std and norivbur simulations, this allows us to determine the riverine/burial-driven air-sea carbon flux of our standard configuration:

$$F_{\text{riv./bur.}, N/S/I}^{\text{C, air-sea}} = F_{\text{nat.}, N/S/I}^{\text{C, air-sea}}(\text{std}) - F_{\text{nat.}, N/S/I}^{\text{C, air-sea}}(\text{norivbur})$$

$$(14)$$

where the 'nat.' label was omitted since the simulations were conducted under pre-industrial conditions, and therefore, no anthropogenic component was included.

# 2.2.3 Sensitivity simulations

225

The set of sensitivity simulations considered covers a broad range of perturbations to the carbon and Alk riverine and burial fluxes. These simulations aim to assess the effects of different assumptions regarding these external fluxes on the riverine/burial-driven air-sea carbon flux (Table 2, Fig. 2b; see Appendix B2.1). Importantly, within the context of our study, the absolute values of the fluxes – whether they align with literature estimates or not – are not of primary concern. What matters are the relative differences between these values across simulations, which reflect the assumptions being tested and briefly outlined below (see Appendix B2.1 for further details).

First, we introduced variations in riverine discharge to account for uncertainties in its magnitude and partly unresolved characteristics (e.g. labile/refractory, organic/inorganic partitioning). By closing the Alk budget, these variations influenced

CaCO<sub>3</sub> burial. In 'rivref', the OM riverine discharge was considered fully refractory (i.e. persisting on a timescale longer than that of ocean circulation), in contrast to the labile assumption in the standard simulation. We explored fully organic and inorganic riverine discharges in 'rivorg' and 'rivinorg', respectively, and also increased riverine discharge by a factor of 1.5 in 'riv1p5', while maintaining the same partitioning as std. Second, to assess the effect of a non-conserved Alk inventory or an Alk restoration scheme, we disabled OM and CaCO<sub>3</sub> burial, artificially restoring Alk in 'nobur-resto', or assuming a non-conserved Alk inventory in 'nobur-diseq'. Third, we varied CaCO<sub>3</sub> burial to address uncertainties in its pre-industrial magnitude and spatial distribution. We added CaCO<sub>3</sub> burial/dissolution between the Atlantic and Pacific, balancing the Alk budget in 'atlpac', or not in 'atlpac-diseq', and we also added CaCO<sub>3</sub> burial in the tropics, resulting in an imbalanced Alk budget in 'tropics-diseq'.

We classify our simulations as either 'equilibrated' (suffix '-eq') or 'disequilibrated' (suffix '-diseq'), based on the conservation of global carbon and Alk inventories. In both cases, the air–sea carbon flux has reached a steady state. Equilibrated simulations are characterized by balanced global carbon and Alk budgets, resulting in conserved global ocean inventories over time. In contrast, disequilibrated simulations exhibit imbalanced budgets, leading to evolving global ocean inventories, which are therefore not conserved. It is important to note that the variations applied in our set of sensitivity simulations directly affected only carbon and Alk fluxes, while nutrient fluxes were held constant in order to avoid perturbing OM and CaCO<sub>3</sub> production. Finally, we report that at the global scale, for the standard simulation,  $Q_{\rm inv} \simeq 0.797$  (Eq. 8), and this coefficient shows minimal variation across all sensitivity simulations considered (

**Figure 4.** Description of the standard NEMO-PISCES simulation (std; see Fig. B3 for additional elements). (a) Map of the pre-industrial air-sea carbon flux, where positive values indicate ocean ingassing. (b) The zonally integrated air-sea carbon flux (dark blue) and the riverine/burial-driven air-sea carbon flux (aquamarine). When the riverine/burial-driven flux exceeds (is less than) the simulated one, the area in between is shaded in rose (cyan). (c) Partitioning of the riverine (orange) and burial (dark gold) fluxes by ocean region (southern, inter-tropical, and northern). The fluxes, in petagrams of carbon per year (PgC yr<sup>-1</sup>) for carbon (in bold) and Alk (in normal font), are directed by arrows, with orientation indicating the sign, and size reflecting the absolute magnitude of the flux. The regional partitioning of the riverine/burial-driven air-sea carbon flux (aquamarine) and of the potential air-sea carbon flux from regional carbon:Alk budget imbalances (light blue) are also provided above. (d) Partitioning of the integrated external sources and sinks of carbon (shaded) and Alk (hatched). The negative impact of OM burial on Alk is attributed to the release of ammonium when OM is remineralized at the seafloor rather than buried. Detailed descriptions of (c) and (d) can be found in Supplementary S1 and S2.

functioning of the ocean carbon pumps. Indeed, regional air-sea carbon fluxes are primarily influenced by these pumps, which establish and sustain vertical and horizontal carbon gradients within the ocean (e.g. Sarmiento and Gruber, 2006; Murnane et al., 1999; Aumont et al., 2001; Resplandy et al., 2018). Thus, both the physical pump (involving ocean circulation and air-sea carbon exchange) and the biological pump (comprising processes like production, export, and the remineralization/dissolution of OM and CaCO<sub>3</sub>) play pivotal roles in elucidating the overall distribution of the air-sea carbon flux. These air-sea carbon fluxes exhibit significant ingassing in the northern hemisphere (+0.67 PgC yr<sup>-1</sup>) and outgassing in the inter-tropical zone (-0.79 PgC yr<sup>-1</sup>), with minimal ingassing in the southern hemisphere (+0.16 PgC yr<sup>-1</sup>; see Table B2). Overall, the air-sea carbon flux associated with the oceanic carbon pumps is expected to be net-zero when integrated at the global scale, although norivbur shows a small residual component (+0.05 PgC yr<sup>-1</sup>; see Fig.B4a). This residual component is attributed to a residual carbon budget imbalance due to internal ocean processes (see Appendix B2.4).

Finally, by taking the difference between our standard simulation and the simulation without riverine and burial fluxes (std minus norivbur), we isolate the component of interest, i.e. that induced by riverine and burial fluxes (Sect. 2.2.2). This riverine/burial-driven air-sea carbon flux results in a net global outgassing of 0.31 PgC yr<sup>-1</sup>, distributed among the northern, inter-tropical, and southern regions as follows: 0.10, 0.12, and 0.10 PgC yr<sup>-1</sup> (Fig. 4c).

## 3.2 The global riverine/burial-driven air-sea carbon flux

#### 3.2.1 Role of sediment burial fluxes

265

285

290

295

Accounting for the riverine carbon input alone in the standard simulation ( $+0.52 \text{ PgC yr}^{-1}$ ) is insufficient to explain the simulated air-sea carbon outgassing of  $0.27 \text{ PgC yr}^{-1}$ . Burial-associated carbon fluxes, from both OM ( $-0.17 \text{ PgC yr}^{-1}$ ) and CaCO<sub>3</sub> ( $-0.04 \text{ PgC yr}^{-1}$ ; Fig. 4d and see Fig. B4a), act to partially offset this input, thereby reducing the net outgassing to  $0.31 \text{ PgC yr}^{-1}$ .

The importance of burial fluxes in driving the riverine/burial-driven air-sea carbon flux is furthermore exemplified by our set of sensitivity simulations (Fig. 5a,b). Increasing the riverine input by a factor of 1.5, while maintaining its partitioning (riv1p5), results in an increase in carbon outgassing of 0.17 PgC yr<sup>-1</sup>. This is less than the increase in riverine carbon discharge (+0.26 PgC yr<sup>-1</sup>), as part of this additional carbon is buried as CaCO<sub>3</sub> (-0.09 PgC yr<sup>-1</sup>) to maintain a balanced Alk budget (see Fig. B4b). Similarly, a change in the partitioning of the riverine input between organic and inorganic forms (rivorg and rivinorg) does not alter the total magnitude of the riverine carbon input compared to the standard configuration, but it does affect the air-sea carbon outgassing. It reaches 0.47 and 0.20 PgC yr<sup>-1</sup>, respectively (see Table B2), as the associated decrease (-0.38 PgC yr<sup>-1</sup>) or increase (+0.14 PgC yr<sup>-1</sup>) in the riverine Alk discharge relative to std (see Fig. B4b) leads to corresponding changes in CaCO<sub>3</sub> burial (+0.19 and -0.07 PgC yr<sup>-1</sup>, respectively) to maintain a balanced Alk budget. This highlights the pivotal role of CaCO<sub>3</sub> burial in shaping the air-sea carbon flux under the assumption of a balanced Alk budget, where riverine Alk inputs are offset by CaCO<sub>3</sub> burial (Fig. 4d).

However, such a carbon budget – which deduces the pre-industrial air-sea carbon flux from riverine and burial fluxes of carbon – is only valid under the condition of a balanced Alk budget (Fig. 5b). When this assumption does not hold, it becomes

Figure 5. The role of riverine and burial fluxes of carbon and Alk in determining the pre-industrial air-sea carbon flux. (continued)

Figure 5. (continued). (a, b) Comparison between the net global air-sea carbon flux and (a) the integrated riverine fluxes of carbon, or (b) the integrated riverine and burial fluxes of carbon. When the net air-sea carbon flux balances the considered external fluxes (on the 1:1 line), simulation names are indicated in black. This applies to (a) simulations that do not account for burial and conserve the global Alk inventory (norivbur and nobur-resto), and (b) all simulations conserving the global Alk inventory (excluding nobur-diseq, atlpac-diseq, and tropics-diseq). (c) Theoretical framework that accounts for Alk and carbon budgets to reconstruct the net air-sea carbon flux. The net air-sea carbon flux (filled contours) is determined by multiplying the integrated riverine and burial fluxes of Alk (x-axis) by  $Q_{inv}$  and then subtracting the integrated riverine and burial fluxes of carbon (y-axis). The deviation of the net air-sea carbon flux from this relationship in the NEMO-PISCES sensitivity simulations is small (less than  $0.01 \text{ PgC yr}^{-1}$  for all, except nobur-resto: less than  $0.03 \text{ PgC yr}^{-1}$ ). Simulations with a conserved global Alk inventory align with the zero x-axis line. The most recent carbon and Alk budgets (Table 3) provide estimates of riverine and burial fluxes of carbon<sup>b</sup> and Alk<sup>c</sup>, as shown at the top and on the right in grey. The net air-sea carbon flux reconstructed from these flux estimates are indicated as grey rectangles, with confidence intervals at 75 %, 50 %, and 25 %, and projected on the color bar.

necessary to account for both the carbon and Alk budgets to correctly assess the riverine/burial-driven air-sea carbon flux (Sect. 2.1.1).

## 3.2.2 Impact of an imbalanced alkalinity budget

The possibility of an imbalanced Alk budget during the pre-industrial era has been hypothesized multiple times over the past three decades (e.g. Milliman, 1993; Middelburg et al., 2020; Cartapanis et al., 2018; Boudreau et al., 2018). The simulations atlpac-diseq and tropics-diseq allow us to assess the implications of such a deviation in the global Alk inventory (Table 2; see also Table B1), by controlling both the magnitude and spatial distribution of CaCO<sub>3</sub> burial in a way that better reflects current paleoceanographic reconstructions (e.g. Cartapanis et al., 2018). Both simulations implement an imbalanced Alk budget (-0.10 PgC yr<sup>-1</sup>) via additional CaCO<sub>3</sub> burial, either in the deep Atlantic (atlpac-diseq) or in the shallow tropics to represent coral reef processes (tropics-diseq). They lead to the same increase in steady-state air-sea carbon outgassing relative to std (+0.07 PgC yr<sup>-1</sup>). This outcome may seem counterintuitive when applying a simple carbon budget, since both simulations prescribe extra carbon removal from the ocean (to the sediments), yet result in enhanced carbon loss to the atmosphere (Fig. 5b). In fact, the associated outgassing leads to a net decrease in the global ocean carbon inventory (-0.16 PgC yr<sup>-1</sup>), which exceeds, in absolute terms, the additional CaCO<sub>3</sub> burial (-0.10 PgC yr<sup>-1</sup>; see Fig. B4b).

# 3.2.3 Validating the governing equation of the pre-industrial riverine/burial-driven air-sea carbon flux

Importantly, even with an imbalanced Alk budget that drives deviations in the global carbon and Alk inventories, the ocean can maintain a steady-state air-sea carbon flux (Fig. B2). Overall, the theoretical framework introduced in Sect. 2.1.1 is fully validated by our set of sensitivity simulations. At pre-industrial steady state, the net air-sea carbon flux ( $F^{C, air-sea}$ ) can be expressed as the product of the integrated riverine and burial Alk flux ( $F^{Alk}$ ) multiplied by  $Q_{inv}$ , minus the integrated riverine

<sup>&</sup>lt;sup>a</sup>The net air-sea carbon flux of the NEMO-PISCES sensitivity simulations was adjusted for their respective residual carbon budget imbalances (see Appendix B2.4).

<sup>&</sup>lt;sup>b</sup>This distribution also includes fluxes from groundwater discharge.

<sup>&</sup>lt;sup>c</sup>This distribution also includes fluxes from anaerobic processes, groundwater discharge, and reverse weathering.

**Figure 6.** Spatial distribution of the riverine/burial-driven air-sea carbon flux. Comparison between the riverine/burial-driven air-sea carbon flux (y-axis, PgC yr<sup>-1</sup>), the fraction of this flux occurring south of 20°S (x-axis, %) and its interhemispheric gradient (color dots, PgC yr<sup>-1</sup>). The fraction of this flux occurring south of 20°S is also shown for Aumont et al. (2001) and Lacroix et al. (2020) (black stars), assuming the same global riverine/burial-driven air-sea carbon flux as our standard simulation (std).

and burial carbon flux ( $F^{C, \text{ bur./riv.}}$ ):

$$F^{\text{C, air-sea}} = \underbrace{Q_{\text{inv}} \cdot F^{\text{Alk}}}_{D^{\text{C}}} - F^{\text{C, bur./riv.}}, \tag{15}$$

where  $D^{\rm C}$  and  $D^{\rm Alk}$  represent the global carbon and Alk inventory deviations, respectively (Fig. 5c). This formulation highlights the critical role of pre-industrial Alk budget assumptions – as well as the persistent uncertainties and unknowns – in estimating the pre-industrial air-sea carbon flux.

#### 3.3 The spatial distribution of the riverine/burial-driven air-sea carbon flux

#### 3.3.1 Contrasting regional fluxes





The inter-hemispheric gradient of the pre-industrial air-sea carbon flux is primarily controlled by ocean interior processes and the functioning of the ocean carbon pumps. Specifically, in an ocean without any riverine and burial carbon fluxes (norivbur), the inter-hemispheric gradient amounts to +0.51 PgC yr<sup>-1</sup> (see Fig. B4a). The biological pump contributes to carbon uptake in the northern hemisphere through surface biological activity and leads to carbon release in the southern hemisphere due to the upwelling of carbon-rich deep waters, as documented in previous studies (e.g. Murnane et al., 1999; Aumont et al., 2001; Resplandy et al., 2018). When subtracting the gradient estimated from simulation norivbur to all other sensitivity simulations, we find that only a fraction of the inter-hemispheric air-sea carbon flux gradient is accounted for by riverine and burial fluxes (ranging from -0.18 to +0.11 PgC yr<sup>-1</sup>).

Our set of sensitivity simulations, which explore various assumptions about riverine and burial fluxes (Sect. 2.2.3), encompass the uncertainty range associated with the inter-hemispheric gradient of the riverine/burial-driven air-sea carbon flux (Fig. 6). The main point of contention regarding this gradient lies in the fraction of the flux occurring in the southern hemisphere, where the largest discrepancies in estimates of the anthropogenic carbon sink between  $pCO_2$ -based methods and model simulations were located in GCBs (from 2018 to 2022; e.g. Hauck et al., 2020; Friedlingstein et al., 2022b), before being mostly shifted to the inter-tropical region (since 2023; e.g. Friedlingstein et al., 2023, 2024). In our simulations, the fraction of the flux occurring in the southern hemisphere ranges from less than 5 % (nobur-diseq) to more than 50 % (rivref). By comparison, it was estimated at 49 % (Aumont et al., 2001) and then revised to 14 % (Lacroix et al., 2020) in the GCBs (Table 1 and Fig. 6), and even as low as 4 % in the literature (Jacobson et al., 2007). This is particularly intriguing, as one might expect this distribution to be primarily governed by the strength of the meridional overturning circulation – and its role in transporting riverine/burial-derived carbon southward – yet our sensitivity simulations, despite identical ocean dynamics, reveal highly contrasting distributions.

#### 3.3.2 Influencing factors

There is no direct correlation between the magnitude of the riverine/burial-driven air-sea carbon flux and the proportion of this flux occurring south of 20°S (Fig. 6). Notably, the substantial uncertainty on the refractory nature of organic riverine discharge (e.g. Aumont et al., 2001; Gruber et al., 2009) is demonstrated to result in a significant shift in the proportion of the riverine/burial-driven air-sea carbon flux occurring in the southern ocean (54 % in rivref vs. 31 % in std; Fig.6), even though the total flux remains the same. Conversely, when the riverine discharge is increased by 50 % (riv1p5), the distribution of the riverine/burial-driven air-sea carbon flux remains unchanged compared to std, while the total outgassing increases from 0.32 PgC yr<sup>-1</sup> (std) to 0.49 PgC yr<sup>-1</sup> (riv1p5; see Table B2).

The decoupling between the magnitude of the net riverine/burial-driven air-sea carbon flux and its inter-hemispheric gradient is primarily linked to the distribution, both horizontally and vertically, of the carbon:Alk budget imbalance resulting from riverine and burial fluxes (Sect. 2.1.2). When an excess of CaCO<sub>3</sub> burial is considered at the bottom of the Atlantic (primarily

in the northern hemisphere; atlpac-diseq), the resulting impact of the carbon:Alk budget imbalance on the riverine/burial-driven air-sea carbon flux occurs remotely, in the southern hemisphere, due to the meridional overturning circulation. This results in a relative outgassing compared to std  $(-0.07 \text{ PgC yr}^{-1})$ , and an increase in the inter-hemispheric riverine/burial-driven air-sea carbon flux gradient  $(+0.07 \text{ PgC yr}^{-1})$ ; see Fig. B4b). On the contrary, when the surplus of CaCO<sub>3</sub> burial is in the shallow tropics (tropics-diseq), the riverine/burial-driven air-sea carbon flux anomaly compared to std is equivalent to the one reported for atlpac-diseq, but the inter-hemispheric gradient is this time nearly not impacted relative to std  $(+0.01 \text{ PgC yr}^{-1})$  since the flux anomaly is concentrated in the shallow tropics, primarily affecting the regional air-sea carbon flux. Similarly, flux anomalies resulting from carbon:Alk budget imbalances with respect to the riverine fluxes tend to manifest regionally (i.e. primarily in the northern hemisphere): (i) a fully organic riverine discharge (rivorg) leads to a relative outgassing compared to std  $(-0.19 \text{ PgC yr}^{-1})$ , aligned with a decrease in the inter-hemispheric gradient  $(-0.18 \text{ PgC yr}^{-1})$ ; and (ii) a fully inorganic riverine discharge (rivinorg) leads to a relative ingassing compared to std  $(+0.07 \text{ PgC yr}^{-1})$ , aligned with an increase in the inter-hemispheric gradient  $(+0.05 \text{ PgC yr}^{-1})$ ; see Fig. B4b).

## 4 Proof-of-concept applications and discussions

## 4.1 The global flux

# 4.1.1 Approach





The theoretical framework introduced in this study (Sect. 2.1.1) has been validated by our set of sensitivity simulations (Sect. 3.2). It is therefore now possible to estimate the global magnitude of the pre-industrial riverine/burial-driven air-sea carbon flux and to investigate the associated global carbon and Alk inventory deviations (Eq. 10) based on existing carbon and Alk budgets, which encompass all external oceanic sources and sinks of carbon and Alk. For consistency with the literature, we rely on the most recent carbon (Regnier et al., 2022) and Alk (Middelburg et al., 2020) budgets, even though they were derived independently and are partly inconsistent (Table 3). We carefully accounted for the uncertainties and extreme values associated with the various external sources/sinks of carbon and Alk (Table 3).

#### 4.1.2 Findings

Using the theoretical framework introduced in this manuscript and literature-based estimates of riverine/burial fluxes of carbon and Alk, based on the most recent carbon and Alk budgets, we derive, from Eq. 15, a pre-industrial riverine/burial-driven air-sea carbon flux estimate of -0.49 [-0.34; -0.70] PgC yr<sup>-1</sup> (Table 3 and Fig. 5c). This pre-industrial riverine/burial-driven air-sea carbon flux is associated with global carbon and Alk inventory deviations of 0.06 [-0.05; 0.11] PgC yr<sup>-1</sup> and 0.07 [-0.06; 0.14] PgC yr<sup>-1</sup>, respectively (see Fig. E1). This estimate is based on an integrated external flux of 0.55 [0.45; 0.65] PgC yr<sup>-1</sup> for carbon and 0.07 [-0.06; 0.14] PgC yr<sup>-1</sup> for Alk.

**Table 3.** Literature-based estimates of riverine/burial fluxes of carbon and Alk, from the most recent carbon and Alk budgets (Sect. 4.2.1 and see Fig. E3), including the calculation of the corresponding air-sea carbon flux, as well as carbon and Alk content deviations. The values are presented in petagrams of carbon per year. Values in brackets represent the uncertainty or extreme range, while the bold value indicates the best estimate or average. The intervals are arranged with the smallest absolute value first, except when both positive and negative values are present in the range.

| Type of                          | Carbon flux                 | Alk flux                       | Associated air-sea                                           | Associated DIC                              | Associated Alk       |
|----------------------------------|-----------------------------|--------------------------------|--------------------------------------------------------------|---------------------------------------------|----------------------|
| sources/sinks                    | (from Regnier et al., 2022) | (from Middelburg et al., 2020) | carbon flux                                                  | deviation                                   | deviation            |
|                                  |                             |                                | $(Q_{\mathrm{inv}} \cdot F^{\mathrm{Alk}} - F^{\mathrm{C}})$ | $(Q_{\mathrm{inv}} \cdot F^{\mathrm{Alk}})$ | $(F^{\mathrm{Alk}})$ |
| Riverine discharge <sup>a</sup>  | [0.650; 1.150]              | [0.578; 0.929]                 | [-0.189; -0.410]                                             | [0.461; 0.740]                              | [0.578, 0.929]       |
|                                  | 0.900                       | 0.756                          | -0.297                                                       | 0.603                                       | 0.756                |
| $\mathbf{OM}\ \mathbf{burial}^b$ | [-0.059; -0.155]            | [0.014; 0.037]                 | [0.070; 0.184]                                               | [0.011; 0.029]                              | [0.014; 0.037]       |
|                                  | -0.107                      | 0.024                          | 0.126                                                        | 0.019                                       | 0.024                |
| CaCO <sub>3</sub> burial         | [-0.141; -0.345]            | [-0.648; -0.828]               | [-0.315; -0.375]                                             | [-0.516; -0.660]                            | [-0.648; -0.828]     |
|                                  | -0.243                      | -0.708                         | -0.321                                                       | -0.564                                      | -0.708               |
| Total                            | [0.450; 0.650]              | [-0.056; 0.138]                | [-0.340; -0.695]                                             | [-0.045; 0.110]                             | [-0.056; 0.138]      |
|                                  | 0.550                       | 0.072                          | -0.493                                                       | 0.057                                       | 0.072                |

<sup>&</sup>lt;sup>a</sup>Including fluxes from groundwater discharge and anaerobic processes.

#### 4.1.3 Discussion




This new estimation of the pre-industrial riverine/burial-driven air-sea carbon flux represents a downward revision of the latest value of  $-0.65 \pm 0.30$  PgC yr<sup>-1</sup> currently adopted in the GCB (Friedlingstein et al., 2024), which was derived from a comprehensive assessment of the global land-to-ocean carbon continuum Regnier et al. (2022). Applying our revised estimate in the calculation of the anthropogenic carbon uptake based on  $pCO_2$ -based methods would reduce the overall discrepancy between observation-based and model-derived oceanic carbon uptake estimates by 0.16 PgC yr<sup>-1</sup> over the historical period, thus alleviating a portion of the present offset (Fig. 1; Friedlingstein et al., 2024).

The discrepancy between our reassessment of the riverine/burial-driven outgassing and the value currently used in the GCB underscores the crucial importance of clearly defining ocean boundary conditions and the pressing need to develop a combined and consistent carbon and Alk budget for the ocean to achieve a robust estimate. Part of this discrepancy arises because atmospheric carbon uptake by continental shelves (0.10 PgC yr<sup>-1</sup>; Regnier et al., 2022) is fully integrated into our net preindustrial riverine/burial-driven air-sea carbon flux as we also consider OM and CaCO<sub>3</sub> burial in these regions, reducing this flux by 0.10 PgC yr<sup>-1</sup>.

The current inconsistencies between the independently developed carbon and Alk budgets make our estimate less robust and highlight the need for a combined revision of both. Beyond the 0.10 PgC yr<sup>-1</sup> reduction in outgassing due to differing ocean boundary definitions relative to GCB, the remaining 0.06 PgC yr<sup>-1</sup> decrease in our new estimate is linked to a slight imbalance in the Alk budget (+0.07 PgC yr<sup>-1</sup> Middelburg et al., 2020). However, the discrepancy in CaCO<sub>3</sub> burial estimates between

<sup>&</sup>lt;sup>b</sup>Including fluxes from reverse weathering.

the most recent carbon and Alk budgets (Regnier et al., 2022; Middelburg et al., 2020) would translate into a 0.22 PgC yr<sup>-1</sup> difference in the Alk budget (Table 3). If the carbon flux associated with CaCO<sub>3</sub> burial were aligned with the Alk budget from Middelburg et al. (2020), the outgassing would decrease by an additional 0.18 PgC yr<sup>-1</sup>. Conversely, aligning the Alk flux associated with CaCO<sub>3</sub> burial with the carbon value from Regnier et al. (2022) would reduce the outgassing by 0.11 PgC yr<sup>-1</sup>. Thus, reconciling CaCO<sub>3</sub> burial fluxes in both carbon and Alk budgets is expected to further reduce the current outgassing offset (Friedlingstein et al., 2024). Establishing a combined and internally consistent carbon and Alk budget is therefore essential to confidently reassess the pre-industrial outgassing within the theoretical framework presented here.

## 4.2 The flux distribution

## 4.2.1 Approach



The set of sensitivity simulations conducted to validate our theoretical framework spans a wide range of assumptions regarding riverine and burial fluxes of carbon and Alk. This provides all the necessary tools to reassess the spatial distribution of the riverine/burial-driven air—sea carbon flux. As in our global estimate (Sect. 4.1), this reassessment strategy is grounded in the most recent global budgets of carbon and Alk. By logically selecting and weighting some of our sensitivity simulations, we construct a composite simulation whose riverine and burial fluxes match those reported in these global budgets (Table 3). In this way, the composite simulation also combines the associated air—sea carbon fluxes, both at the global scale and regionally.

It is this regional aspect that enables a revised estimate of the spatial distribution of the pre-industrial riverine/burial-driven air—sea carbon flux.

First, for each literature-based estimate of the external sources and sinks of carbon and Alk, we constructed a skewed Gaussian probability density function (PDF) that captures the median/mean value and the reported uncertainty range. This was achieved in two steps for each literature estimate of the various external sources/sinks of carbon and Alk. A triangular distribution was first generated using the estimated central value and minimum/maximum bounds via the 'random.triangular' function from the Python library *numpy*. This triangular distribution was then fitted with a skewed normal PDF using the 'stats.skewnorm.fit' function from the *scipy* library. This approach allowed us to preserve the essential characteristics of the literature values (median/mean and extremes) while working with continuous distributions.

Second, we constructed a composite simulation that isolates the effect of riverine and burial fluxes on the air-sea carbon flux (i.e. excluding the influence of internal carbon pumps). This was achieved by linearly combining a subset of our sensitivity simulations. Throughout the remainder of the manuscript, we refer to the composite simulated estimate as the pre-industrial riverine/burial-driven air-sea carbon flux derived from this combined simulation. The following four-step workflow is designed to ensure that the riverine and burial fluxes in the composite simulation are consistent with the latest literature estimates (Middelburg et al., 2020; Regnier et al., 2022, see Fig. E2, and Table 3).

Step 1: We initialized our composite simulation by isolating the effect of riverine and burial fluxes on the air-sea carbon flux, removing the contribution of internal carbon pumps. This was done by subtracting the norivbur simulation from the standard simulation ('std'-'norivbur').

- Step 2: Next, we adjusted the carbon fluxes associated with riverine discharge and OM burial to match literature estimates. This was achieved by weighting the simulation where riverine inputs were increased by a factor of 1.5 ('riv1p5'-'std'), as the overall riverine flux amplitude was the first variable that needed to be tuned. At the end of this stage, the composite simulated estimate was a linear combination of 'std'-'norivbur' and 'riv1p5'-'std'.
  - Step 3: We then adjust the Alk fluxes associated with riverine discharge and OM burial to match literature estimates. This was achieved by weighting the simulation where all riverine discharge was considered inorganic ('rivinorg'-'std'), which did not alter the carbon values already matched in Step 2. The composite simulated estimate became a linear combination of the result from Step 2 and 'rivinorg'-'std'.
  - Step 4: Finally, we ensured that Alk fluxes associated with CaCO<sub>3</sub> burial also matched the literature estimate. This was done using the simulation with enhanced CaCO<sub>3</sub> burial/dissolution and a global Alk imbalance ('atlpac-diseq'-'std'), without affecting the fluxes adjusted in previous steps. Given the 2:1 stoichiometric ratio between Alk and DIC in CaCO<sub>3</sub> processes, this step simultaneously ensured consistency for both the carbon and Alk components of CaCO<sub>3</sub> burial. The final composite estimate was a linear combination of the result from Step 3 and 'atlpac-diseq'-'std'.

Extra step (correction): Due to inconsistencies between the most recent carbon and Alk budgets – specifically in the CaCO<sub>3</sub> burial flux (Table 3) – an additional correction step was required. This correction, applied similarly to Step 3, again uses 'rivinorg'-'std' to consider an increased carbon sink via CaCO<sub>3</sub> burial, while maintaining Alk balance. This adjustment targets only the excess CaCO<sub>3</sub> burial of carbon needed to reconcile our composite simulation with the carbon budget from Regnier et al. (2022). Note that this step would not be necessary if the carbon and Alk budgets were internally consistent.

In summary, this composite simulated estimate, built as a weighted linear combination of targeted sensitivity simulations and constrained by the latest literature estimates of riverine and burial fluxes, provides a spatially explicit representation of the pre-industrial riverine/burial-driven air-sea carbon flux. By design, the integrated value of this flux in the composite simulation is consistent with that obtained by applying the theoretical framework to the most recent carbon and Alk budgets from the literature (Sect. 4.1).

## 4.2.2 Findings






The construction of a composite simulated estimate resulting from a linear combination of our sensitivity simulations to align with the literature-based estimates for carbon and Alk budgets (Fig. 7a; Sect. 4.2.1 and see Fig. E3) enables a reassessment of the distribution of this riverine/burial-driven air-sea carbon outgassing  $(0.15 \pm 0.13, 0.20 \pm 0.10, \text{ and } 0.16 \pm 0.08 \text{ PgC yr}^{-1}$  for the southern, inter-tropical, and northern regions, respectively; Fig. 7e). The uncertainty associated with these values is primarily linked to the uncertainties/extremes in literature-based estimates (see Fig. E1). Such a distribution implies that 29 % of the outgassing occurs in the southern region, 40 % in the inter-tropical region, and 31 % in the northern region.

Figure 7. Description of the composite simulated estimate resulting from a linear combination of the NEMO-PISCES sensitivity simulations and literature-based estimates of riverine/burial fluxes of carbon and Alk (Sect. 4.2.1 and see Fig. E3). (a) PDF illustrating the total riverine and burial fluxes of carbon (shaded) and Alk (hatched) in the composite simulated estimate, along with the associated PDF for the resulting deviation in carbon (solid) and Alk (dashed) content. (b,c) PDFs of the net air-sea carbon flux and the inter-hemispheric air-sea carbon flux gradient. Within each of these sub-panels, the PDF associated with no riverine and burial fluxes of carbon and Alk (norivbur; cyan line) is juxtaposed with the one corresponding to only riverine and burial fluxes of carbon and Alk (composite simulated estimate minus norivbur; aquamarine) to obtain the total value (composite simulated estimate; dark blue). Further details on the residual component where no riverine and burial fluxes are considered are explained in Appendix B2.4. (d, e, f) The associated spatial distribution for the southern, inter-tropical, and northern regions: (d) without riverine and burial fluxes of carbon; (e) exclusively related to riverine and burial fluxes of carbon and Alk; and (f) the overall distribution. In (e), the percentage of each component is provided in brackets.

#### 4.2.3 Discussion

- The distribution we found corresponds to an intermediate distribution compared to those adopted in the GCB over time, falling between the most recent estimate of 14 %, 64 %, and 22 % (Lacroix et al., 2020), and the earlier estimate of 49 %, 25 %, and 26 % (Aumont et al., 2001, Table 1). This would partially confirm the reduction in the discrepancy between *p*CO<sub>2</sub>-based and model estimates in the southern region, while avoiding the introduction of a bias in the inter-tropical region, as noted in GCB 2023 (Friedlingstein et al., 2023) compared to previous GCBs (e.g. Friedlingstein et al., 2022b).
- By summing the fluxes from the composite simulated estimate and the simulation without riverine and burial fluxes (norivbur), the total inter-hemispheric air-sea carbon flux gradient can be obtained. Notably, this amounts to  $0.50\pm0.15$  PgC yr<sup>-1</sup> (Fig. 7c), which aligns with the inter-hemispheric CO<sub>2</sub> concentration gradient in the atmosphere between the South Pole and Mauna Loa during the pre-industrial era. It was historically assessed at +0.82 ppm (Keeling et al., 1989) and more recently reevaluated at  $+0.55\pm0.15$  ppm (Resplandy et al., 2018) through interpolation of atmospheric CO<sub>2</sub> concentration measurements.
- A more comprehensive characterization of riverine and burial fluxes of carbon and Alk remains a critical challenge for accurately constraining the spatial distribution of the riverine/burial-driven air—sea carbon flux. This is particularly true for the fate of terrestrial organic carbon and its associated lability, which remains highly uncertain (Aumont et al., 2001; Jacobson et al., 2007; Gruber et al., 2009). Nevertheless, the approach proposed in this study is flexible and can accommodate future revisions of these external fluxes. Fundamentally, the selection of sensitivity simulations used to construct the composite simulation (Sect. 4.2.1; see also Fig. E2) can be revisited as scientific understanding progresses or as model representations evolve. For instance, in NEMO-PISCES, burial tends to occur predominantly near coastal margins. To counterbalance this biased feature in the composite simulation, we selected the sensitivity simulation with extra CaCO<sub>3</sub> burial in the deep Atlantic basin (atlpacdiseq), rather than the one with increased burial in the shallow tropics (tropics-diseq). A limitation of our approach is that a substantial revision of the spatial distribution of a flux such as riverine inputs would require rerunning a simulation, as it cannot be addressed through our current framework alone.

Nonetheless, the use of sensitivity simulations to build a composite simulation underscores the method's potential for reassessing the distribution of the pre-industrial air—sea carbon flux. By drawing from a set of pre-existing simulations and grounding the reassessment in the theoretical framework developed in this study, the spatial pattern of the flux can be revised in a consistent and coherent manner, without the need for additional model runs. This approach is particularly well suited for model intercomparison exercises, as it allows for efficient re-evaluation of regional fluxes and contributes to reducing biases linked to differences in ocean circulation or biogeochemical parameterizations across models.

## 5 Conclusion and perspectives



We have offered a fresh perspective on the pre-industrial air—sea carbon flux by leveraging the ocean alkalinity budget. The theoretical framework we introduced, validated through sensitivity simulations conducted with NEMO-PISCES, demonstrates both its robustness and practical relevance for assessing the riverine/burial-driven pre-industrial air—sea carbon flux in the context of the GCB.

Through two proof-of-concept applications, we demonstrate the potential of this theoretical framework to identify biases between observation-based and model-derived estimates of the oceanic carbon sink at both global and regional scales, and to partially correct persistent offsets. In the first application, we revisit the global magnitude of the pre-industrial riverine/burial-driven air—sea carbon flux using existing carbon and alkalinity budgets. This yields a simple and rapid method for reassessment whenever these budgets are revised. In the second application, we propose a method to reassess the spatial distribution of the pre-industrial riverine/burial-driven air—sea carbon flux. This is achieved by constructing a composite simulation, based on a linear combination of sensitivity simulations, that aligns with both carbon and alkalinity budgets. This approach is particularly well-suited for model intercomparison exercises, as it enables efficient reassessment of regional fluxes while helping to mitigate biases related to ocean physics or biogeochemical parameterizations.

These flexible applications now call for four key efforts from the community regarding the pre-industrial riverine/burial-driven air-sea carbon flux:

(i) To reduce uncertainty in its global magnitude:





- Clarify the definition of ocean domain boundaries at the coastal interface within the land-to-ocean continuum, where multiple fluxes intersect (riverine discharge, and part of organic matter and CaCO<sub>3</sub> burial).
- Establish a combined and internally consistent carbon and alkalinity budget, as current independently developed estimates remain inconsistent (e.g. CaCO<sub>3</sub> burial).
- (ii) To reduce uncertainty in its regional distribution:
- Improve our understanding of the intrinsic properties of riverine and burial fluxes (e.g. the fate of terrestrial organic matter).
  - Promote intermodel comparison efforts to identify systematic biases and improve robustness across modeling approaches.

#### Appendix A: Theoretical framework

As a complement to the theoretical framework introduced in Sect. 2.1.1, we outline here how to derive  $Q_{\rm inv}$ , the inverse of the 'isocapnic quotient' approximation introduced by Humphreys et al. (2018). Specifically, we develop the method proposed by Planchat et al. (2023), and subsequently demonstrate its full consistency with the approach employed by Humphreys et al. (2018).

For a fixed salinity (S) and temperature (T),  $pCO_2$  – the partial pressure of  $CO_2$  in seawater – can be differentiated as follows:

$$dpCO_2 = \frac{\partial pCO_2}{\partial Alk} \Big|_{DIC,S,T} \cdot dAlk + \frac{\partial pCO_2}{\partial DIC} \Big|_{Alk,S,T} \cdot dDIC$$
 (A1)

Assuming  $pCO_2$  is fixed – for instance, at equilibrium with atmospheric  $CO_2$  – leads to:

$$\frac{\mathrm{dDIC}}{\mathrm{dAlk}}\bigg|_{p\mathrm{CO}_{2},S,T} = -\left. \frac{\partial p\mathrm{CO}_{2}}{\partial \mathrm{Alk}} \right|_{\mathrm{DIC},S,T} \cdot \left. \frac{\partial \mathrm{DIC}}{\partial p\mathrm{CO}_{2}} \right|_{\mathrm{Alk},S,T}$$
(A2)

Yet,  $pCO_2$  is defined by:

$$pCO_2 = \frac{K_2}{K_0 \cdot K_1} \cdot \frac{\left[HCO_3^-\right]^2}{\left[CO_3^{2-}\right]} \tag{A3}$$

where  $K_0$ ,  $K_1$  and  $K_2$  are the stoichiometric equilibrium/dissociation constants of the CO<sub>2</sub> system (e.g. Sarmiento and Gruber, 2006).

We then introduce the simplifying assumption:

$$\begin{cases} [HCO_3^-] \simeq 2DIC - Alk \\ [CO_3^{2-}] \simeq Alk - DIC \end{cases}$$
(A4)

This assumption is reasonable given that  $\left[\mathrm{HCO_3^-}\right]^2$  and  $\left[\mathrm{CO_3^{2-}}\right]$  together typically account for over 99 % of DIC and over 97 % of Alk (Sarmiento and Gruber, 2006; Humphreys et al., 2018). Under this assumption, Eq. A3 can be approximated as:

$$pCO_2 \simeq \frac{K_2}{K_0 \cdot K_1} \cdot \frac{(2DIC - Alk)^2}{Alk - DIC}$$
 (A5)

Accordingly, the partial derivatives of  $pCO_2$  with respect to Alk and DIC at constant  $pCO_2$ , S, and T become:

$$\begin{cases}
\frac{\partial p \text{CO}_2}{\partial \text{Alk}} \Big|_{\text{DIC}, S, T} \simeq \frac{K_2}{K_0 \cdot K_1} \cdot \frac{-\text{Alk} \cdot (2 \text{DIC} - \text{Alk})}{(\text{Alk} - \text{DIC})^2} \\
\frac{\partial p \text{CO}_2}{\partial \text{DIC}} \Big|_{\text{Alk}, S, T} \simeq \frac{K_2}{K_0 \cdot K_1} \cdot \frac{(3 \text{Alk} - 2 \text{DIC}) \cdot (2 \text{DIC} - \text{Alk})}{(\text{Alk} - \text{DIC})^2}
\end{cases}$$
(A6)

Substituting these expressions into Eq. A2 gives:

$$\left. \frac{\text{dDIC}}{\text{dAlk}} \right|_{p \in \mathcal{O}_2, S, T} = \frac{1}{Q} = Q_{\text{inv}} \simeq \frac{\text{Alk}}{3\text{Alk} - 2\text{DIC}}$$
 (A7)

It is worth noting that the same expression can also be derived following the method presented in Humphreys et al. (2018, their Appendix C). Using the same approximation as in Eq. A4, they arrive at the following form for Alk (see their Eq. C.6):

$$Alk \simeq 2DIC + \frac{\beta}{2} - \sqrt{\frac{\beta^2}{4} + \beta DIC}$$
(A8)

with:

$$\beta = \frac{(2\text{DIC} - \text{Alk})^2}{(\text{Alk} - \text{DIC})} = p\text{CO}_2 \cdot \frac{K_0 \cdot K_1}{K_2}$$
(A9)

Differentiating Alk with respect to DIC at constant  $pCO_2$ , S, and T then yields:

$$\frac{\mathrm{dAlk}}{\mathrm{dDIC}}\bigg|_{p\mathrm{CO}_2, S, T} \simeq 2 - \frac{\beta}{2 \cdot \sqrt{\frac{\beta^2}{4} + \beta \mathrm{DIC}}} = 2 - \frac{1}{\sqrt{1 - \frac{4\mathrm{DIC}}{\beta}}}$$
(A10)

After rearrangement, this leads to the same expression for  $Q_{inv}$ :

$$\frac{\mathrm{dAlk}}{\mathrm{dDIC}}\bigg|_{p \in \Omega_2, S, T} = Q = \frac{1}{Q_{\mathrm{inv}}} \simeq \frac{3\mathrm{Alk} - 2\mathrm{DIC}}{\mathrm{Alk}}$$
(A11)

An exact formulation of Q is also provided by Humphreys et al. (2018, their Appendix D).

#### Appendix B: Model and simulations




#### **B1** CMIP6 ESMs and GCB GOBMs

We present an evaluation of the representation of the pre-industrial air-sea carbon flux in ESMs and GOBMS that participated in the CMIP6 exercise (Eyring et al., 2016) and the 2024 GCB exercise (Friedlingstein et al., 2024). This assessment offers valuable insights into the current state of the art regarding the modeling of this flux in the models utilized for intercomparison studies. To ensure comparability, we regridded the CMIP6 data to a regular 1°x1° grid using the distance-weighted average remapping method 'remapdis' provided by the Climate Data Operator (CDO). This step was taken because the data available from the 2024 GCB (Hauck et al., 2022) were already on a regular 1°x1° grid. However, it is important to note that this regridding process introduced a minor error in the integrated air-sea carbon flux values.

We assessed 15 CMIP6 ESMs from 12 different climate modelling centers (Eyring et al., 2016): CanESM5 (r1i1p2f1) and CanESM5-CanOE (r1i1p2f1) from CCCma, with two distinct marine biogeochemical models; CMCC-ESM2 (r1i1p1f1) from CMCC; CNRM-ESM2-1 (r1i1p1f2) from CNRM-CERFACS; ACCESS-ESM1-5 (r1i1p1f1) from CSIRO; IPSL-CM6A-LR (r1i1p1f1) from IPSL; MIROC-ES2L (r1i1p1f2) from MIROC; UKESM1-0-LL (r1i1p1f2) from MOHC; MPI-ESM1-2-LR (r1i1p1f1) and MPI-ESM1-2-HR (r1i1p1f1) from MPI-M, with two different resolutions; MRI-ESM2-0 (r1i2p1f1) from MRI, CESM2-WACCM (r1i1p1f1) from NCAR; NorESM2-LM (r1i1p1f1) from NCC; GFDL-CM4 (r1i1p1f1) and GFDL-ESM4 (r1i1p1f1) from NOAA-GFDL, with two distinct marine biogeochemical models. Only the air-sea CO<sub>2</sub> flux (positive donward, 'fgCO<sub>2</sub>' in kgC m<sup>-2</sup> s<sup>-1</sup>) of the pre-industrial control simulations was considered, from 1850 to 2100, and yearly averaged. Each ESM was weighted in the calculation of the CMIP6 mean, such that each modelling group has the same total contribution. We also assessed the 10 GOBMs used in the 2024 GCB exercise (Friedlingstein et al., 2024): NEMO3.6-PISCESv2-gas (CNRM), NEMO4.2-PISCES (IPSL), MPIOM-HAMOCC6, MRI-ESM2-3, ACCESS, MICOM-HAMOCC (NorESM-OC), MOM6-COBALT (Princeton), FESOM-2.1-REcoM3, NEMO-PlanckTOM12 and CESM-ETHZ). Once again, only the air-sea CO<sub>2</sub> flux (positive donward, 'fgCO<sub>2</sub>' in mol m<sup>-2</sup> s<sup>-1</sup>) of the control simulations (i.e. constant atmospheric CO<sub>2</sub>, no climate change and variability) was considered, from 1959 to 2023, and yearly averaged. We found that the drift in the ESMs and

GOBMs in the net air-sea carbon flux was consistently less than 0.10 PgC (100 yr)<sup>-1</sup>, and as such, it had negligible impact on the related results (see Fig. 2a and D3a).

## **B2** NEMO-PISCES sensitivity simulations

## **B2.1** Configurations







We provide here additional details regarding the various configurations of the sensitivity simulations conducted using NEMO-PISCES (Table B1). In the standard configuration, the slight deviation (-0.02 PgC yr<sup>-1</sup>) between Alk riverine discharge (+0.35 PgC yr<sup>-1</sup>) and inorganic carbon riverine discharge (+0.37 PgC yr<sup>-1</sup>) arises from the supply of inorganic nitrogen by rivers, presumed to be in the form of nitrate, which has a negative impact on Alk (Fig. B4). It is worth noting that the global values and latitudinal distribution of riverine inputs are based on Ludwig et al. (1996) and have recently been revised (Li et al., 2017; Liu et al., 2024), although the human imprint on these fluxes cannot be removed. Lastly, we emphasize that we did not evaluate the implications of partitioning riverine inputs between inorganic and organic components on biological production, and consequently, its effects on the air-sea carbon flux, as we only altered DIC and Alk in the various configurations. Finally, we accounted for atmospheric deposition in our sensitivity simulations, since atmospheric nitrogen deposition is considered a nitrate source, which impacts Alk. This has however a negligible effect, as does the dilution effect (see Supplementary S2).

The manuscript has been crafted to be accessible and comprehensible for both observationalists and modelers. However, the deviations mentioned for the carbon and Alk inventories manifest themselves in model outputs in the form of drifts. Furthermore, all the sensitivity simulations conducted also address modeling issues. In particular, a case that can be encountered in marine biogeochemistry models, both historically and even today, is the consideration, or lack thereof, of the OM and CaCO<sub>3</sub> burial, and the consequences this can have on the carbon flux, depending on whether the global Alk inventory is conserved through a global-scale Alk restoration scheme, or left deviating (nosed-resto, nosed-diseq; Planchat et al., 2023). Finally, the choice of the different configurations, and their resulting impact on the air-sea carbon flux, also serve as a reminder of the importance of carefully considering the global Alk inventory in models, and controlling its potential deviation/drift according to desired hypotheses (e.g. a balanced Alk budget or not).

From a practical standpoint, in NEMO-PISCES, CaCO<sub>3</sub> burial predominantly occurs in coastal areas (Fig. B1a), with limited differentiation in burial at depth between the Atlantic (less acidic) and Pacific (more acidic) regions (Sarmiento and Gruber, 2006; Cartapanis et al., 2018; Ridgwell and Zeebe, 2005, Fig. B1a). To address this limitation, we introduced the configuration 'atlpac' wherein we constrain extra CaCO<sub>3</sub> burial in the deep Atlantic while simulating extra CaCO<sub>3</sub> dissolution in the deep Pacific. This adjustment aims to enhance the representation of CaCO<sub>3</sub> burial while maintaining a balanced Alk budget (i.e. conserving the global Alk inventory; Fig. B1b). Additionally, considering the possibility of an imbalanced Alk budget during the pre-industrial era due to extra CaCO<sub>3</sub> burial at depth (Cartapanis et al., 2018), we created two configurations to account for this extra carbon burial (0.10 PgC yr<sup>-1</sup>): (i) in the deep Atlantic (atlpac-diseq), and (ii) in the shallow tropical regions (tropics-diseq), simulating the accumulation of carbon by coral reefs (Fig. B1b).

Table B1. Summary of the sensitivity simulations led with NEMO-PISCES with their full description and characteristics (supplement to Table 2).

|               | Configuration                                       |            |                                     |           | Parame                     | Parameterization |                                                        |             |
|---------------|-----------------------------------------------------|------------|-------------------------------------|-----------|----------------------------|------------------|--------------------------------------------------------|-------------|
| Simulation    | Description                                         | Alk budget | Riverine discharge                  | OM burial | CaCO <sub>3</sub> burial   | Atmospheric      | Net addition                                           | Alk         |
|               |                                                     |            |                                     |           |                            | deposition       |                                                        | restoration |
| std           | Standard                                            | Balanced   | ×                                   | ×         | To conserve the global Alk | ×                | 1                                                      | 1           |
|               |                                                     |            |                                     |           | inventory                  |                  |                                                        |             |
| norivbur      | No external fluxes of carbon and Alk,               | Balanced   | I                                   | 1         | I                          | ı                | I                                                      | 1           |
|               | except air-sea carbon fluxes                        |            |                                     |           |                            |                  |                                                        |             |
| nivref        | Refractory organic riverine discharge               | Balanced   | Alk and carbon from organic         | ×         | To conserve the global Alk | ×                | I                                                      | ı           |
|               |                                                     |            | riverine discharge are spread all   |           | inventory                  |                  |                                                        |             |
|               |                                                     |            | over the ocean                      |           |                            |                  |                                                        |             |
| rivorg        | Fully organic riverine discharge                    | Balanced   | For every mole of carbon            | ×         | To conserve the global Alk | ×                | I                                                      | 1           |
|               |                                                     |            | discharged by rivers, Alk is        |           | inventory                  |                  |                                                        |             |
|               |                                                     |            | discharged with the N: C ratio of   |           |                            |                  |                                                        |             |
|               |                                                     |            | organic carbon discharge            |           |                            |                  |                                                        |             |
| rivinorg      | Fully inorganic riverine discharge                  | Balanced   | For every mole of carbon            | ×         | To conserve the global Alk | ×                | 1                                                      | 1           |
|               |                                                     |            | discharged by rivers, a mole of Alk |           | inventory                  |                  |                                                        |             |
|               |                                                     |            | is also discharged                  |           |                            |                  |                                                        |             |
| riv1p5        | Riverine discharge of carbon and Alk                | Balanced   | 1.5 times the amount of carbon      | ×         | To conserve the global Alk | ×                | 1                                                      | 1           |
|               | multiplied by 1.5                                   |            | and Alk supplied in std             |           | inventory                  |                  |                                                        |             |
| nosed-resto   | No OM and CaCO <sub>3</sub> burial,                 | Balanced   | ×                                   | ı         | ı                          | ×                | 1                                                      | ×           |
|               | but restoration of the Alk content                  |            |                                     |           |                            |                  |                                                        |             |
| nosed-diseq   | No OM and CaCO <sub>3</sub> burial                  | Imbalanced | ×                                   | ı         | 1                          | ×                | I                                                      | ı           |
| atlpac        | Constrained balance of extra CaCO3                  | Balanced   | ×                                   | ×         | To conserve the global Alk | ×                | Equivalent amount of carbon and Alk                    | 1           |
|               | burial/dissolution between the deep                 |            |                                     |           | inventory, without         |                  | added/removed in the deep Pacific/Atlantic             |             |
|               | Atlantic/Pacific                                    |            |                                     |           | considering 'net addition' |                  | (0.10 PgC yr-1 of carbon and 0.20 PgC yr-1 of Alk)     |             |
| atlpac-diseq  | Constrained imbalance of extra CaCO <sub>3</sub>    | Imbalanced | ×                                   | ×         | To conserve the global Alk | ×                | Twice more carbon and Alk removed in the deep          | 1           |
|               | burial/dissolution between the deep                 |            |                                     |           | inventory, without         |                  | Atlantic compared to the deep Pacific                  |             |
|               | Atlantic/Pacific (-0.10 PgC yr-1)                   |            |                                     |           | considering 'net addition' |                  | (-0.20 PgC yr-1 relative to 0.10 PgC yr-1 of carbon,   |             |
|               |                                                     |            |                                     |           |                            |                  | and -0.40 PgC yr-1 relative to 0.20 PgC yr-1 of Alk)   |             |
| tropics-diseq | tropics-diseq Constrained extra CaCO3 burial in the | Imbalanced | ×                                   | ×         | To conserve the global Alk | ×                | Removal of carbon and Alk in the shallow tropics       | 1           |
|               | shallow tropics (-0.10 PgC yr-1)                    |            |                                     |           | inventory, without         |                  | (-0.10 PgC yr-1 for carbon and -0.20 PgC yr-1 for Alk) |             |
|               |                                                     |            |                                     |           | considering 'net addition' |                  |                                                        |             |
|               |                                                     |            |                                     |           |                            |                  |                                                        |             |

**Figure B1.** Towards a controlled adjustment of extra CaCO<sub>3</sub> burial/dissolution. (a) Map depicting CaCO<sub>3</sub> burial in the standard simulation (std). (b, c) Masks employed to drive (b) a balanced (atlpac) or an imbalanced (atlpac-diseq) additional CaCO<sub>3</sub> burial/dissolution between the deep Atlantic and Pacific, as well as (c) an extra CaCO<sub>3</sub> burial in the tropics. Red (blue) shading represents an addition (removal) of DIC and Alk in the grid cell at a 1:2 ratio. The grid cells considered for this addition/removal are located at 4750 m for the deep Atlantic and Pacific masks, and between 0 and 100 m for the tropics mask.

# B2.2 Spin-up



We track here the evolution of the net air-sea carbon flux during the spin-up for all the NEMO-PISCES sensitivity simulations (Table B1 and Fig. B2a), which are initially branched to a simulation at quasi-steady state equivalent to our standard simulation (std). Two characteristic time-scales emerge (Fig. B2b): (i) a short-term stabilisation over the first 50 yr, and (ii) a long-term stabilisation beyond 50 yr. The short-term (long-term) stabilisation primarily corresponds to the response of the surface (deep) ocean to the modifications associated with the configuration regarding the DIC and Alk external fluxes (Fig. B2c,d). Thus, for the simulation where we constrain extra CaCO<sub>3</sub> burial in the shallow tropics, only a stabilisation of the surface ocean is generally needed, resulting in only a short-term stabilisation. On the contrary, in the case where this extra CaCO<sub>3</sub> burial is constrained in the deep Atlantic, only a stabilisation of the deep ocean is generally needed, resulting in only a long-term stabilisation. Finally, in the case where riverine organic matter input is considered to be entirely refractory (rivref), a significant anomaly in external fluxes is induced at the surface compared to the standard simulation (std), as well as in the deep ocean because this organic carbon input is spread all over the ocean. This results in both short-term and long-term responses.

#### **B2.3** Standard simulation (std)

We provide additional details here regarding the standard simulation (std, Fig. B3) to offer points of comparison with historical modeling studies that have initiated research efforts on this pre-industrial carbon flux (Aumont et al., 2001; Murnane et al., 1999).

Figure B2. Spin-up of the NEMO-PISCES sensitivity simulations. (a) Time series of the net air-sea carbon flux with a 50-yr rolling mean throughout the 2550 yr of the simulations. (b) Same time series in relative to std and without smoothing. The thin black lines refer to the combined exponential fits  $(y = \alpha \cdot e^{-\frac{t}{\tau}} + \beta)$ , where  $\alpha$  is the net air-sea carbon flux offset,  $\tau$  is the time constant, and  $\beta$  is the baseline; using the *curve\_fit* function from the *scipy* python library): (i) one for the short-term considering the first 50 yr; and (ii) one for the long-term, considering the remaining 2500 yr. (c, d) for the short-term (c) and long-term (d) exponential fits, the net air-sea carbon flux offset  $\alpha$  is displayed in function of the time constant  $\alpha$  with their associated uncertainties.

**Figure B3.** Description of the standard NEMO-PISCES simulation (std; continued from Fig. 2). Zonally integrated (a) carbon and (b) Alk fluxes in supplement to Fig. 2b. (c) Latitudinal distribution of the northward transport of carbon (solid) and Alk (dahsed). When the regional imbalance exceeds (falls behind) the simulated air-sea carbon flux, the area in between is shaded in rose (cyan).

#### **B2.4** Residual carbon budget imbalance



A minor imbalance in the carbon budget from external sources/sinks persists without any associated ocean carbon content deviation in our sensitivity simulations. This discrepancy is particularly evident in the standard simulation (std; see 'Total' and 'Drift' in Supplementary S2 and Fig. B4a) but is also observed in other simulations such as rivref, rivorg, rivinorg, riv1p5, atlpac, atlpac-diseq, and tropics-diseq (see Supplementary S2). To understand this counterintuitive result initially, we must examine diazotrophic organisms, which produce OM without altering Alk. Let's imagine a thought experiment where the ocean contains external sources/sinks of carbon and Alk, such as riverine discharge and CaCO<sub>3</sub> burial, but maintains a balanced Alk budget. Then, the ocean carbon balance can be performed independently of Alk to infer the air-sea carbon flux at steady state (see Sect. 3.2.3). Now, let's introduce the production of OM by diazotrophic organisms into this steady-state ocean, assuming that all of it is buried. These organisms will consume carbon in the surface ocean and export it in the sediments without affecting Alk. This leads to a carbon sink in the ocean, which, when brought back to steady state, is counter-balanced by a positive air-sea carbon flux. Therefore, the imbalance in the carbon budget for std results from the OM burial produced by

diazotrophs. In reality, the effect of diazotrophic organisms is more complex, as only a fraction of their OM is buried, and the rest is remineralized, leading to an increase in Alk. However, a similar effect on the air-sea carbon flux would be observed, albeit with a different magnitude. Since we could not determine the distribution of this induced air-sea carbon flux, we could not correct this slight imbalance in the carbon budget from external sources/sinks, except in Fig. 5, where only the total value of the air-sea carbon flux is considered, without its distribution. Please note that this unaccounted-for air-sea carbon flux stemming from external ocean carbon and Alk sources/sinks also contributes to the understanding of the slight discrepancy between the simulated air-sea carbon flux and the one resulting from the carbon:Alk global imbalance (e.g. +0.04 PgC yr<sup>-1</sup> for std; Fig. B4a).

Another type of imbalance in the carbon budget is evident in the simulation without external ocean source/sink (norivbur), accompanied by non-conserved ocean carbon and Alk inventories (see 'Total' and 'Drift' in Supplementary S2 and Fig. B4a). This imbalance arises from the representation of nitrogen reactions in NEMO-PISCES, which includes the restoration of nitrate content in the ocean. An imbalance between nitrification (decreasing Alk) and denitrification (increasing Alk) leads to an internal Alk imbalance (an imbalance stemming from N-reactions is also reported in COBALTv2 Stock et al., 2020). This is not compensated for by the strategy used to conserve the Alk inventory, as  $CaCO_3$  burial is not considered in this simulation (see Sect. 2.2). At steady state, this positive global Alk inventory deviation ( $D^{Alk}$ ) results in an air-sea carbon flux ( $F^{C, air-sea}$ ) and an ocean carbon content deviation ( $D^C$ ) of the same magnitude:  $F^{C, air-sea} = D^C = Q_{inv} \cdot D^{Alk}$  (see Fig. 3b and Sect. 3.2.2). Thus, the imbalance in the carbon budget for norivbur is associated with an air-sea carbon flux resulting from an internal Alk imbalance, also leading to a non-conserved ocean carbon content. As expected, this imbalance is almost equivalent in the simulation without burial and a non-conserved global Alk inventory (nobur-diseq). Once again, as we were unable to access the distribution of this induced air-sea carbon flux, we could not correct this slight imbalance in the carbon budget from external sources/sinks, except in Fig. 5, where only the total value of the air-sea carbon flux is considered.

Very minor residual undesirable disturbances, such as deviations or slight inconsistencies in the budgets over the 50-year period considered, may persist due to the minor non-linearity occurring during the burial of CaCO<sub>3</sub> when the global Alk inventory is constrained to be conserved by the burial of CaCO<sub>3</sub>. Additionally, the modeling scheme of the physical part of NEMO-PISCES induces a slight Alk deviation and a slightly more pronounced carbon content deviation (respectively - 0.002 PgC yr<sup>-1</sup> and +0.01 PgC yr<sup>-1</sup> in std).

## 660 B2.5 NEMO-PISCES sensitivity simulation ensemble





We provide a comprehensive overview of the global-scale carbon and Alk budgets for all NEMO-PISCES sensitivity simulations (Fig. B4). Even more detailed information can be found in Supplementary S1 and S2 (https://doi.org/10.5281/zenodo. 8421898). Finally, we also provide a comprehensive characterization of the air-sea carbon flux in the NEMO-PISCES sensitivity simulations, including both the total flux and the riverine/burial-driven component (Table B2).

Figure B4. Global-scale carbon and Alk budgets for all NEMO-PISCES sensitivity simulations. (continued)

**Figure B4.** (*continued*). Carbon and Alk budgets (a) in absolute values for the standard simulation (std) and the simulation without riverine and burial fluxes of carbon and Alk (norivbur), or (b) relative to std for the other NEMO-PISCES sensitivity simulations. The type of representation is close to the one shared in Fig. 4c, but integrated over the whole ocean. All fluxes, in petagrams of carbon per year (PgC yr<sup>-1</sup>) for carbon (in bold) and Alk (in normal font), are directed by arrows, with orientation indicating the sign, and size reflecting the absolute magnitude of the flux. In (b), only the fluxes (riverine discharge, as well as OM and CaCO<sub>3</sub> burial) with a significant anomaly are displayed, along with their associated changes relative to the standard simulation (std) in brackets, for both carbon (bold) and Alk (normal font). Additionally, values for carbon and Alk deviations (for simulations with a '-diseq' suffix), net addition flux (for atlpac, atlpac-diseq, and tropics-diseq), or the term of Alk restoration (for nosed-resto) is/are also shown when applicable (Table B1). In (a), for the standard simulation (std), a first approximation of the impact of OM and CaCO<sub>3</sub> production in the surface waters is also inferred from POC and PIC export at 100 m (in brackets with a star). Finally, in addition to the air-sea carbon flux (dark blue), the air-sea carbon flux stemming from global imbalance (light blue; Sect. 2.1.2 and Appendix B2.4 for an explanation of the residual imbalance) is also shared, as well as the associated inter-hemispheric air-sea carbon flux gradient (dark and light cyan). In (b), as the values are shown relative to the standard simulation (std), the simulated air-sea carbon flux anomalies are equivalent to the ones of the riverine/burial-driven air-sea carbon flux. A detailed description of the NEMO-PISCES sensitivity simulations can be found in Supplementary S1 and S2.

**Table B2.** Comprehensive description of the net air-sea carbon flux in the NEMO-PISCES sensitivity simulations. The values provided in parentheses are expressed relative to the simulation without riverine and burial fluxes, representing the riverine/burial-driven air-sea carbon flux or carbon transport.

| C!1-4!        | Net air-sea carbon flux |                                             |              | Inter-hemispheric |                            |               |
|---------------|-------------------------|---------------------------------------------|--------------|-------------------|----------------------------|---------------|
| Simulation    | South                   | Inter-tropics                               | North        | Total             | Inter-hemispheric gradient | transport     |
|               | (<20°S)                 | $(20^{\circ}\text{S} - 20^{\circ}\text{N})$ | (>20°N)      | Iotai             | (north - south)            | of carbon     |
| std           | 0.06 (-0.10)            | -0.91 (-0.12)                               | 0.57 (-0.10) | -0.27 (-0.32)     | 0.51 (-0.00)               | -0.35 (-0.09) |
| norivbur      | 0.16                    | -0.79                                       | 0.67         | 0.05              | 0.51                       | -0.26         |
| rivref        | -0.01 (-0.17)           | -0.86 (-0.08)                               | 0.61 (-0.07) | -0.27 (-0.32)     | 0.62 (0.10)                | -0.37 (-0.11) |
| rivorg        | 0.11 (-0.05)            | -1.03 (-0.24)                               | 0.45 (-0.23) | -0.47 (-0.52)     | 0.33 (-0.18)               | -0.24 (0.02)  |
| rivinorg      | 0.05 (-0.11)            | -0.86 (-0.08)                               | 0.61 (-0.06) | -0.20 (-0.25)     | 0.56 (0.05)                | -0.38 (-0.12) |
| riv1p5        | 0.01 (-0.15)            | -0.97 (-0.19)                               | 0.52 (-0.15) | -0.44 (-0.49)     | 0.51 (0.00)                | -0.40 (-0.15) |
| nobur-resto   | -0.09 (-0.25)           | -0.95 (-0.16)                               | 0.54 (-0.14) | -0.50 (-0.55)     | 0.62 (0.11)                | -0.40 (-0.14) |
| nobur-diseq   | 0.15 (-0.01)            | -0.93 (-0.14)                               | 0.59 (-0.08) | -0.18 (-0.23)     | 0.44 (-0.07)               | -0.31 (-0.05) |
| atlpac        | 0.05 (-0.11)            | -0.90 (-0.12)                               | 0.57 (-0.10) | -0.28 (-0.33)     | 0.52 (0.01)                | -0.36 (-0.10) |
| atlpac-diseq  | -0.01 (-0.17)           | -0.89 (-0.11)                               | 0.57 (-0.11) | -0.34 (-0.39)     | 0.58 (0.06)                | -0.36 (-0.11) |
| tropics-diseq | 0.03 (-0.14)            | -0.92 (-0.13)                               | 0.55 (-0.13) | -0.34 (-0.39)     | 0.52 (0.01)                | -0.35 (-0.09) |

## Appendix C: Ocean regions, and boundary conditions

The boundaries chosen to demarcate the southern, inter-tropical, and northern regions at 20°S and 20°N (see Fig. 3a, 4b, as well as Tables B2, E1, and E2) have indeed been previously employed in the literature (e.g. Aumont et al., 2001; Resplandy et al., 2018). These boundaries primarily align with physical features of the ocean, especially concerning air-sea carbon fluxes. It is in, or very close, to these latitudes that the air-sea carbon flux resulting from regional carbon:Alk budget imbalance reconciles with the simulated one (see Fig. 4c). By employing these boundaries, the air-sea carbon flux from regional carbon:Alk budget imbalances (see Sect. 2.1.2) closely matches the simulated values for each oceanic region (see Supplementary S1). This alignment deteriorates when the boundaries are shifted away from 20°S and 20°N. Consequently, we have opted for a consistent approach, maintaining the 20°S/N boundary to delineate distinct oceanic regions, despite the shift to 30°S/N boundaries in the GCB, primarily to correspond with terrestrial biomes (Friedlingstein et al., 2024). However, for potential use in the GCB, we share values of the spatial distribution with boundaries at 30°S/N in Table E2.

## Appendix D: Inter-hemispheric air-sea carbon flux gradient

#### D1 Partitioning between the northern and southern components

We share additional insights regarding the inter-hemispheric air-sea carbon flux gradient, which is crucial for the global carbon cycle in its connection with the atmosphere and land (e.g. Keeling et al., 1989; Resplandy et al., 2018). It is thus valuable to distinguish in this inter-hemispheric gradient the component associated with the net air-sea carbon flux in both the southern and northern regions (Fig. D1).

#### D2 Partitioning between carbon transport and riverine and burial processes

#### **D2.1** Expression




In addition to defining the inter-hemispheric air–sea carbon flux gradient (G), and the inter-hemispheric oceanic transports of carbon  $(T^{C})$  and Alk  $(T^{Alk})$ , we propose here a decomposition of G into contributions associated with carbon transport and with riverine and burial processes.

To this end, we recall that the total regional fluxes of carbon and Alk can be expressed as follows:

$$\begin{cases} F_{N/S/I}^{\text{C}} = F_{N/S/I}^{\text{C, air-sea}} + F_{N/S/I}^{\text{C, riv./bur.}} \\ F_{N/S/I}^{\text{Alk}} = F_{N/S/I}^{\text{Alk, riv./bur.}} \end{cases}$$
(D1)

Specifically, by considering these fluxes  $(F_{N,S,I}^{C/Alk})$  along with regional carbon and Alk deviations  $(D_{N,S,I}^{C/Alk})$  and assuming a steady-state ocean (see Fig. B2), we derive two expressions for the ocean transport of carbon and Alk through their respective

**Figure D1.** Decomposition of the inter-hemispheric air-sea carbon flux gradient (supplement to Fig. 2). Decomposition of the inter-hemispheric air-sea carbon flux gradient into the net southern and northern air-sea carbon fluxes for (a) CMIP6 and GCB, and (b) the NEMO-PISCES sensitivity simulations. Filled contours correspond to the inter-hemispheric air-sea carbon flux gradient. (a) The 15 CMIP6 ESMs (10 GCB GOBMs) are plotted with red squares (orange circles). The black square and circle refer to the CMIP6 and GCB ensemble means. In (b), secondary axes have been added to characterize the implied changes for the southern/northern air-land carbon flux relative to std, if the inter-hemispheric gradient is considered as well-represented. Then, a decrease in the net sourthern (northern) air-sea carbon flux relative to std entails an increase of the same magnitude in the net southern (northern) air-land carbon flux relative to std, and conversely.

budget closure equations  $(T_{N/S}^{C/Alk})$ :

$$\begin{cases} T_{N/S}^{\text{C/Alk}} + F_{\geq 20^{\circ} N/20^{\circ} S}^{\text{C/Alk}} + D_{\geq 20^{\circ} N/20^{\circ} S}^{\text{C/Alk}} = 0 \\ -T_{N/S}^{\text{C/Alk}} + F_{< 20^{\circ} N/20^{\circ} S}^{\text{C/Alk}} + D_{< 20^{\circ} N/20^{\circ} S}^{\text{C/Alk}} = 0 \end{cases}$$
 (D2)

Hence, we define the ocean transport of carbon and Alk as the average of its two expressions (Eq. D2; Fig. 3b):

$$T_{N/S}^{\text{C/Alk}} = \frac{1}{2} \cdot \left[ \left( F_{<20^{\circ}N/20^{\circ}S}^{\text{C/Alk}} + D_{<20^{\circ}N/20^{\circ}S}^{\text{C/Alk}} \right) - \left( F_{\geq20^{\circ}N/20^{\circ}S}^{\text{C/Alk}} + D_{\geq20^{\circ}N/20^{\circ}S}^{\text{C/Alk}} \right) \right] \tag{D3}$$

In particular:

$$\begin{cases} T_{N}^{\text{C/Alk}} = \frac{1}{2} \cdot \left[ \left( F_{S}^{\text{C/Alk}} + D_{S}^{\text{C/Alk}} + F_{I}^{\text{C/Alk}} + D_{I}^{\text{C/Alk}} \right) - \left( F_{N}^{\text{C/Alk}} + D_{N}^{\text{C/Alk}} \right) \right] \\ T_{S}^{\text{C/Alk}} = \frac{1}{2} \cdot \left[ \left( F_{S}^{\text{C/Alk}} + D_{S}^{\text{C/Alk}} \right) - \left( F_{I}^{\text{C/Alk}} + D_{I}^{\text{C/Alk}} + F_{N}^{\text{C/Alk}} + D_{N}^{\text{C/Alk}} \right) \right] \end{cases}$$
(D4)

from which an expression of the inter-hemispheric transport of carbon and Alk (Eq. 12) can be derived:

$$T^{\text{C/Alk}} = \frac{1}{2} \cdot \left[ \left( F_S^{\text{C/Alk}} + D_S^{\text{C/Alk}} \right) - \left( F_N^{\text{C/Alk}} + D_N^{\text{C/Alk}} \right) \right] \tag{D5}$$

Specifically, using Eq. D1, the inter-hemispheric transport of carbon can be rewritten as follows:

$$T^{\mathcal{C}} = \frac{1}{2} \cdot \left[ \left( F_S^{\mathcal{C}, \text{ air-sea}} + F_S^{\mathcal{C}, \text{ riv./bur.}} + D_S^{\mathcal{C}} \right) - \left( F_N^{\mathcal{C}, \text{ air-sea}} + F_N^{\mathcal{C}, \text{ riv./bur.}} + D_N^{\mathcal{C}} \right) \right]$$

$$(D6)$$

After rearrangement, the inter-hemispheric air-sea carbon flux gradient can be expressed as:

$$G = \underbrace{-2 \cdot T^{C}}_{\text{Transport component}} \underbrace{-\left(F_{N}^{C, \text{ riv./bur.}} - F_{S}^{C, \text{ riv./bur.}}\right) - \left(D_{N}^{C} - D_{S}^{C}\right)}_{\text{Riverine and burial component}}$$
(D7)

At first glance, it may appear that this expression is exclusively formulated in terms of carbon, seemingly without any consideration of Alk. However, Alk plays a subtle yet integral role in this equation. Firstly, because  $T^C$  depends on both the southern and northern air-sea carbon fluxes (Eq. D5 and D1), and these regional fluxes are chemically driven by the relative imbalance between Alk and DIC. Secondly, the deviations in the carbon content of the northern and southern oceans ( $D_N^C$  and  $D_S^C$ , respectively) are directly linked to the deviations in Alk content (Eq. 10). Thus, the role of Alk is intricately interwoven within the formulation of G (Eq. D7).

# D2.2 Results


It is possible to decompose the inter-hemispheric gradient of the riverine/burial-driven air-sea flux into a component associated with the inter-hemispheric transport of a carbon:Alk budget imbalance and a component associated with a carbon:Alk budget imbalance stemming from riverine and burial fluxes (including inventory deviations; Eq. D7, and Fig. D2, D3, and E1). Focusing solely on the effect of riverine and burial fluxes (i.e. relative to norivbur), the component associated with these external fluxes (-0.07 PgC yr<sup>-1</sup>) is offset by the transport-related component (+0.07 PgC yr<sup>-1</sup>) in the standard configuration, resulting in a null inter-hemispheric riverine/burial-driven air-sea carbon flux gradient for std (Fig. 5b and Fig. D2). In the case of a surplus of CaCO<sub>3</sub> burial in the deep Atlantic (atlpac-diseq), the increase in the inter-hemispheric riverine/burial-driven air-sea carbon flux gradient relative to std (+0.07 PgC yr<sup>-1</sup>; see Fig. B4b) is primarily attributed to the transport of a carbon:Alk budget imbalance (Fig. D2). Conversely, when the riverine discharge is entirely organic (rivorg), it is mostly the external flux component that causes the decrease in the inter-hemispheric air-sea carbon flux gradient relative to std (-0.18 PgC yr<sup>-1</sup>; see Fig. B4b and Fig. D2), and the same outcome occurs when the riverine discharge is entirely inorganic (rivinorg). This emphasizes that the spatial distribution of the carbon:Alk budget imbalance stemming from external fluxes, in conjunction with oceanic transport, plays a significant role in shaping the pre-industrial inter-hemispheric riverine/burial-driven air-sea carbon flux gradient.

# **Appendix E: Applications**

#### 725 E1 Literature review

Here, we provide a literature review on: (i) the evolution of the assessment and characterization of the air-sea carbon flux since the late 1990s (Table E1); and (ii) the evolution of the estimation and characterization of the riverine/burial-driven air-sea

Figure D2. Drivers of the spatial distribution of the riverine/burial-driven air-sea carbon flux. Inter-hemispheric gradient of the riverine/burial-driven air-sea carbon flux (filled contours) and its two components, from carbon:Alk budget imbalances (see Sect. 2.1.2 and 2.1.2). One component (x-axis) is associated with the inter-hemispheric gradient of air-sea carbon flux driven by northern and southern carbon:Alk budget imbalances (and inventory deviations), while the other component (y-axis) corresponds to the inter-hemispheric gradient of air-sea carbon flux associated to the inter-hemispheric transport of the carbon:Alk budget imbalance (Eq. D7 and Fig. D1). The deviation in the simulated inter-hemispheric gradient in NEMO-PISCES sensitivity simulations as compared to the reconstructed ones using the two components is minimal ( $

**Figure D3.** Partitioning of the inter-hemispheric air-sea carbon flux gradient from inter-hemispheric carbon:Alk budget imbalances (supplement to D2). (a, b) Visual construction of the inter-hemispheric air-sea carbon flux gradient resulting from regional carbon:Alk budget imbalances due to (a) riverine and burial fluxes (including inventory deviations), and (b) inter-hemispheric transport, defining the values used in Fig. D2. The reference was set on the simulation without riverine and burial fluxes (norivbur), so that the combination of the arrows of (a) and (b) results in the inter-hemispheric riverine/burial-driven air-sea carbon flux gradient. (c) Synthetic characterization for the whole set of NEMO-PISCES sensitivity simulations of the inter-hemispheric air-sea carbon flux gradient: with a southern/northern decomposition (as in Fig. D1b), and the partitioning resulting from regional carbon:Alk budget imbalances due to riverine and burial fluxes (including inventory deviations) – constructed in (a) –, and inter-hemispheric transport – constructed in (b).

carbon flux in comparison with our composite simulated estimate (Table E2). We also provide the PDFs of the literature-based estimates for the ocean's external sources/sinks of carbon and Alk, derived from the most recent carbon and Alk budgets (Regnier et al., 2022; Middelburg et al., 2020), which were used to construct the composite simulated estimate (Fig. E1, see Table 3 as well as Sect. 4.2.1 and 4.2.2).

a All the ocean biogeochemistry model used in the following studies both account for the physical carbon pump and the biological one (soft-tissue and carbon ade).

Boundaries at 15°5/N. Boundaries at the equator.

Table E1. Literature review of the net air-sea carbon flux and its characterization. Bold lines are those accounting for external fluxes and boundaries at 20°S/N for the spatial distribution.

|                                | Describeron of the method                                                                                     |                                 |               |                                                           |                                   |                                                       |                                               |                                    |
|--------------------------------|---------------------------------------------------------------------------------------------------------------|---------------------------------|---------------|-----------------------------------------------------------|-----------------------------------|-------------------------------------------------------|-----------------------------------------------|------------------------------------|
| Source                         | and further characterization                                                                                  |                                 | South (<20°S) | Inter-tropics $(20^{\circ}\text{S} - 20^{\circ}\text{N})$ | North (>20°N)                     | Total                                                 | Inter-hemispheric gradient<br>(north - south) | transport of carbon                |
|                                | PPOM                                                                                                          | Moddeling approach <sup>a</sup> |               |                                                           |                                   |                                                       |                                               |                                    |
| Murnane et al. (1999)          | No external fluxes of carbon and Alk. The CaCO3 reaching the seafloor is redissolved                          | Princeton model                 | 09.0          | -1.20 <sup>b</sup>                                        | 0900                              | 0.00                                                  | 0.00 <sup>b</sup>                             |                                    |
|                                | at the surface to conserve the global Alk inventory.                                                          |                                 |               |                                                           |                                   |                                                       |                                               |                                    |
| Samiento et al. (2000)         | Intercomparison study of three ocean biogeochemistry models with the same                                     | Princeton model                 | -0.55°        |                                                           | -0.09€                            | -0.64                                                 |                                               |                                    |
|                                | implicit riverine discharge of 0.64 PgC yr <sup>-1</sup> split between the northern and southern              |                                 |               |                                                           |                                   |                                                       |                                               |                                    |
|                                | hemispheres (respectively -0.21 and -0.43 PgC yr <sup>-1</sup> for the associated outgassing).                |                                 |               |                                                           |                                   |                                                       |                                               |                                    |
|                                |                                                                                                               | IPSL model                      | -0.53°        |                                                           | -0.11°                            | -0.64                                                 |                                               |                                    |
|                                |                                                                                                               | MPI model                       | -0.39°        |                                                           | -0.25°                            | -0.64                                                 |                                               |                                    |
| Aumont et al. (2001)           | No external fluxes of carbon and Alk. The CaCO3 reaching the seafloor is redissolved                          | IPSL model                      | 0.73          | -1.40                                                     | 0.67                              | 0.00                                                  | -0.06                                         | -0.10                              |
|                                | at depth to conserve the global Alk inventory.                                                                |                                 |               |                                                           |                                   |                                                       |                                               |                                    |
|                                | Carbon and Alk riverine discharge of respectively 0.81 and 0.40 PgC yr <sup>-1</sup> , the global             | IPSL model 0.43                 | 0.43          | -1.55                                                     | 0.51                              | -0.61                                                 | 80.0                                          | -0.25                              |
|                                | Alk inventory being conserved through an equivalent CaCO3 burial. Regarding the                               |                                 |               |                                                           |                                   |                                                       |                                               |                                    |
|                                | riverine carbon (0.41 PgC yr¹), 1/3 is injected as DIC at the river mouth and the rest                        |                                 |               |                                                           |                                   |                                                       |                                               |                                    |
|                                | is injected as DOC with an oxidation time-scale of 100 yr.                                                    |                                 |               |                                                           |                                   |                                                       |                                               |                                    |
| Lacroix et al. (2020)          | No external fluxes of carbon and Alk. The CaCO3 reaching the seafloor is redissolved                          | MPI model                       |               |                                                           |                                   | -0.05                                                 |                                               |                                    |
|                                | homogeneously at the surface to conserve the global Alk inventory, same for organic                           |                                 |               |                                                           |                                   |                                                       |                                               |                                    |
|                                | mater. This results in an equivalent implicit riverine discharge of 0.314 PgC yr¹for                          |                                 |               |                                                           |                                   |                                                       |                                               |                                    |
|                                | carbon and 0.208 PgC yr <sup>-1</sup> for Alk.                                                                |                                 |               |                                                           |                                   |                                                       |                                               |                                    |
|                                | Constrained riverine discharge (0.603 PgC yr <sup>-1</sup> for carbon and 0.366 PgC yr <sup>-1</sup> for Alk) | MPI model                       |               |                                                           |                                   | 0.18                                                  |                                               |                                    |
|                                | based on a hierarchy of weathering and terrestrial organic matter export models,                              |                                 |               |                                                           |                                   |                                                       |                                               |                                    |
|                                | while identifying regional hotspots of the riverine exports. OM and CaCO <sub>3</sub> burial are              |                                 |               |                                                           |                                   |                                                       |                                               |                                    |
|                                | considered and not constrained (respectively 0.582 and 0.188 PgC yr <sup>-1</sup> ), making free              |                                 |               |                                                           |                                   |                                                       |                                               |                                    |
|                                | the global Alk inventory.                                                                                     |                                 |               |                                                           |                                   |                                                       |                                               |                                    |
| Present study                  | Linear combination of sensitivity simulations to match literature estimates of the                            | IPSL model                      | $0.01\pm0.13$ | IPSL model $0.01\pm0.13$ $-0.99\pm0.10$                   | $\textbf{0.51} \pm \textbf{0.08}$ | $0.51 \pm 0.08  \text{-0.46} \pm 0.24  0.50 \pm 0.15$ | $0.50\pm0.15$                                 | $\textbf{-0.16} \pm \textbf{0.08}$ |
| (composite simulated estimate) | (composite simulated estimate) external sources/sinks of carbon and AIk                                       |                                 |               |                                                           |                                   |                                                       |                                               |                                    |

(continued)

|                                              | Decombestion of the mothed                                                                                                           |                                   |                                    | Net air-sea carbon flux           | carbon flux                                                                                          |                                   | International             |
|----------------------------------------------|--------------------------------------------------------------------------------------------------------------------------------------|-----------------------------------|------------------------------------|-----------------------------------|------------------------------------------------------------------------------------------------------|-----------------------------------|---------------------------|
| Source                                       | Describing of the Highlian                                                                                                           | South                             | Inter-tropics                      | North                             |                                                                                                      | Inter-hemispheric gradient        | The remains place in      |
|                                              | and turther characterization                                                                                                         | (<20°S)                           | (20°S - 20°N)                      | (>20°N)                           | Iotal                                                                                                | (north - south)                   | transport of carbon       |
|                                              |                                                                                                                                      |                                   |                                    |                                   |                                                                                                      |                                   |                           |
|                                              | Combined atmosphere-ocean inversion                                                                                                  | ersion                            |                                    |                                   |                                                                                                      |                                   |                           |
| Sarmiento and Sundquist (1992)               | Use of an ocean general circulation model (Sarmiento et al., 1992) to substract the simulated anthropogenic                          | 0.82 <sup>b</sup>                 | -1.70 <sup>b</sup>                 | 0.28 <sup>b</sup>                 | -0.60                                                                                                | 0.54 <sup>b</sup>                 |                           |
|                                              | carbon uptake to the revised carbon budget by Tans et al. (1990) from a combination of an atmospheric transport                      |                                   |                                    |                                   |                                                                                                      |                                   |                           |
|                                              | model with a compilation of observations of air-sea CO <sub>2</sub> difference.                                                      |                                   |                                    |                                   |                                                                                                      |                                   |                           |
| Jacobson et al. (2007)                       | Use of a combination of transport inversions of atmospheric (constraining fluxes into the atmosphere from both                       |                                   |                                    |                                   | $-0.39 \pm 0.19$                                                                                     |                                   |                           |
|                                              | land and ocean) and oceanic observations (constraining only air-sea fluxes), using multiple circulation models to                    |                                   |                                    |                                   |                                                                                                      |                                   |                           |
|                                              | assess the effects of errors in simulated transport. The ocean inversion estimates are corrected to remove the                       |                                   |                                    |                                   |                                                                                                      |                                   |                           |
|                                              | riverine carbon discharge $(0.45~\mathrm{PgC~yr^{-1}})$ .                                                                            |                                   |                                    |                                   |                                                                                                      |                                   |                           |
|                                              | Ocean inversion                                                                                                                      |                                   |                                    |                                   |                                                                                                      |                                   |                           |
| Gloor et al. (2003) <sup>e</sup>             | Use of an ocean general circulation model as well as observations of DIC and other tracers. The values were                          | 0,32 <sup>d</sup>                 | -1.35 <sup>d</sup>                 | 0.63 <sup>d</sup>                 | -0.69                                                                                                | 09.00 <sup>d</sup>                |                           |
|                                              | corrected to remove the riverine carbon discharge with the estimate (0.45 PgC yr <sup>-1</sup> ) from Jacobson et al. (2007).        |                                   |                                    |                                   |                                                                                                      |                                   |                           |
| Mikaloff Fletcher et al. (2007) <sup>f</sup> | Use of ten different ocean general circulation models (to quantify the error arising from uncertainties in the                       | $\textbf{0.21} \pm \textbf{0.16}$ | $\textbf{-1.10} \pm \textbf{0.16}$ | $\textbf{0.40} \pm \textbf{0.14}$ | $0.40 \pm 0.14$ -0.49 $\pm 0.27$ 0.19 $\pm 0.21$                                                     | $\textbf{0.19} \pm \textbf{0.21}$ | $\textbf{-0.19} \pm 0.09$ |
|                                              | modeled transport) as well as observations of DIC and other tracers. The values were corrected to remove the                         | 9                                 |                                    |                                   |                                                                                                      |                                   |                           |
|                                              | riverine carbon discharge with the estimate (0.45 PcC vr <sup>-1</sup> ) from Tacobson et al. (2007).                                |                                   |                                    |                                   |                                                                                                      |                                   |                           |
|                                              | itectine carroon discharge with the commune (see 1 gc 3). ) from disconsistive an (2007).                                            |                                   |                                    |                                   |                                                                                                      |                                   |                           |
|                                              | pCO <sub>2</sub> -based                                                                                                              |                                   |                                    |                                   |                                                                                                      |                                   |                           |
| Resplandy et al. (2018)                      | Reconstruction of the modern-day air-sea carbon flux from the global surface ocean Takahashi et al. (2009)                           |                                   | -0.16 $\pm$ 0.38 -1.09 $\pm$ 0.22  | $0.41\pm0.23$                     | $-0.84 \pm 0.50  0.57 \pm 0.44$                                                                      | $0.57 \pm 0.44$                   |                           |
|                                              | pCO <sub>2</sub> field. The anthropogenic carbon flux is then extracted using an observationnally                                    |                                   |                                    |                                   |                                                                                                      |                                   |                           |
|                                              | based reconstruction with a Green's function estimated from tracer data                                                              |                                   |                                    |                                   |                                                                                                      |                                   |                           |
|                                              | (Khatiwala et al., 2009). The values were corrected to remove the riverine carbon                                                    |                                   |                                    |                                   |                                                                                                      |                                   |                           |
|                                              | discharge with the estimate (0.45 PgC yr <sup>-1</sup> ) from Jacobson et al. (2007).                                                |                                   |                                    |                                   |                                                                                                      |                                   |                           |
|                                              | Wanninkhof et al. (2013)                                                                                                             | 8                                 |                                    |                                   | $\textbf{-1.00} \pm 0.67$                                                                            |                                   |                           |
|                                              | Landschützer et al. (2014)                                                                                                           | t) $-0.04 \pm 0.38$               | $-1.11 \pm 0.22$                   | $0.45\pm0.23$                     | $-0.70 \pm 0.50$                                                                                     | $0.49 \pm 0.44$                   |                           |
|                                              | Same as above, except that riverine carbone discharge is constrained to ensure the Rödenbeck et al. (2013)                           | 3) $0.00 \pm 0.38$                | $\textbf{-1.29} \pm 0.22$          | $0.51\pm0.22$                     | $-0.78 \pm 0.50$                                                                                     | $0.51 \pm 0.44$                   | -0.43                     |
|                                              | closing of the ocean carbon budget (only considering air-sea fluxes and riverine                                                     |                                   |                                    |                                   |                                                                                                      |                                   |                           |
|                                              | carbon discharge).                                                                                                                   |                                   |                                    |                                   |                                                                                                      |                                   |                           |
| Regnier et al. (2022)                        | Reconstruction of the modern-day air-sea carbon flux from the global surface ocean pCO2 field                                        | 0.20                              | -1.47                              | 0.61                              | $-0.65 \pm 0.30$                                                                                     | 0.41                              |                           |
|                                              | (Landschützer et al., 2014). The anthropogenic carbon flux is then extracted using three different methods: global                   |                                   |                                    |                                   |                                                                                                      |                                   |                           |
|                                              | ocean biogeochemistry models, ocean circulation inverse model, and pCO2-based flux mapping models - with a                           |                                   |                                    |                                   |                                                                                                      |                                   |                           |
|                                              | post-correction applied to the latter for the rivers (0.6 PgC yr <sup>-1</sup> ) – (DeVries et al., 2019). The values were corrected |                                   |                                    |                                   |                                                                                                      |                                   |                           |
|                                              | to remove the riverine carbon discharge with the estimate (0.45 PgC yr <sup>-1</sup> ) from Jacobson et al. (2007).                  |                                   |                                    |                                   |                                                                                                      |                                   |                           |
|                                              | Mean ±                                                                                                                               | $Mean \pm std^g  0.00 \pm 0.22$   | $\textbf{-1.24} \pm \textbf{0.13}$ | $\textbf{0.46} \pm \textbf{0.13}$ | $\textbf{0.46} \pm \textbf{0.13}  \textbf{-0.74} \pm \textbf{0.29}  \textbf{0.46} \pm \textbf{0.25}$ | $\textbf{0.46} \pm \textbf{0.25}$ |                           |
|                                              |                                                                                                                                      |                                   |                                    |                                   |                                                                                                      |                                   |                           |

<sup>b</sup> Boundaries at 15°SN. <sup>d</sup> Boundaries at 13°SN. <sup>c</sup> The values were extracted from the Supplementary Figure of Mikaloff Fleicher et al. (2007). <sup>f</sup> Values from Resplandy et al. (2018). <sup>g</sup> Accounting for all the pCO<sub>2</sub>-based estimates, except the one by Wanninkhof et al. (2013), which is incomplete.

and CaCO<sub>3</sub> burial. The values obtained from the composite simulated estimate are also shared in grey. Note that its riverine discharge, OM burial and CaCO<sub>3</sub> fluxes Table E2. Literature review of the riverine/burial-driven air-sea carbon flux estimates, sharing also values and a description of the riverine discharge, OM burial, (see Table 3) differ from those considered in the GCB and derived from Regnier et al. (2022), as we also account for flux on continental shelves.

|                                                                   |                                   |                                 |                                                                                                                               |                                |                                       |                                 | Air-sea carbon outgassing from       | Partitioning                                                  |                                                 |
|-------------------------------------------------------------------|-----------------------------------|---------------------------------|-------------------------------------------------------------------------------------------------------------------------------|--------------------------------|---------------------------------------|---------------------------------|--------------------------------------|---------------------------------------------------------------|-------------------------------------------------|
|                                                                   | Riverine discharge                |                                 | OM burial                                                                                                                     |                                | CaCO <sub>3</sub> burial              |                                 | riverine and burial                  | (south; inter-tropics; north)                                 | north)                                          |
|                                                                   |                                   |                                 |                                                                                                                               |                                |                                       |                                 | fluxes of carbon and Alk             | with boundaries at 20°S/N                                     | 0°S/N                                           |
| Source                                                            | Comment                           | Flux<br>(PgC yr <sup>-1</sup> ) | Comment                                                                                                                       | Flux<br>(PgC yr <sup>1</sup> ) | Comment                               | Flux<br>(PgC yr <sup>-1</sup> ) | (PgC yr <sup>-1</sup> )              | Comment                                                       | Flux<br>(PgC yr <sup>4</sup> )                  |
| Present study                                                     | Linear combination of sensitivity |                                 | $0.90\pm0.26$ Linear combination of sensitivity $-0.11\pm0.05$ Linear combination of sensitivity $-0.25\pm0.05$ $0.51\pm0.24$ | $-0.11 \pm 0.05$               | Linear combination of sensitivity     | $-0.25 \pm 0.05$                | $0.51 \pm 0.24$                      | Partitioning shared with boundaries at 20°S/N                 | $(0.15 \pm 0.13; 0.20 \pm 0.10; 0.16 \pm 0.08)$ |
| (composite simulated estimate)                                    | simulations to match literature   |                                 | simulations to match literature                                                                                               |                                | simulations to match literature       |                                 |                                      | Partitioning shared with boundaries at 30°S/N                 | $(0.13\pm0.13;0.24\pm0.12;0.13\pm0.07)$         |
|                                                                   | estimates                         |                                 | estimates                                                                                                                     |                                | estimates                             |                                 |                                      |                                                               |                                                 |
| Regnier et al. (2022)                                             | Mass balance calculation through  |                                 | $0.80 \pm 0.30$ From literature                                                                                               | $\textbf{-0.04} \pm 0.02$      | -0.04 $\pm$ 0.02 From literature      | $-0.13\pm0.02$                  | $-0.13 \pm 0.02$ $0.65 \pm 0.30^{a}$ | Using Lacroix et al. (2020) partitioning;                     | (0.09; 0.42; 0.14)                              |
|                                                                   | the land-to-ocean aquatic         |                                 |                                                                                                                               |                                |                                       |                                 |                                      | e.g. Friedlingstein et al. (2023, 2024)                       |                                                 |
|                                                                   | continuum loop                    |                                 |                                                                                                                               |                                |                                       |                                 |                                      | Using Aumont et al. (2001) partitioning;                      | (0.32; 0.16; 0.17)                              |
|                                                                   |                                   |                                 |                                                                                                                               |                                |                                       |                                 |                                      | e.g. Friedlingstein et al. (2022b)                            |                                                 |
| Friedlingstein et al. (2020, 2022a) Mean of Murnane et al. (1999) | Mean of Murnane et al. (1999)     | 0.74                            | Mean of Resplandy et al. (2018) -0.05                                                                                         | -0.05                          | Mean of Resplandy et al. (2018) -0.08 | -0.08                           | 0.61                                 | Using Aumont et al. (2001) partitioning;                      | (0.30; 0.15; 0.16)                              |
|                                                                   | and Jacobson et al. (2007)        |                                 | and Jacobson et al. (2007)                                                                                                    |                                | and Jacobson et al. (2007)            |                                 |                                      | e.g. Friedlingstein et al. (2020, 2022a)                      |                                                 |
| Lacroix et al. (2020)                                             | From a hydrological discharge     | 09'0                            | From an ocean biogeochemistry                                                                                                 | -0.19                          | From an ocean biogeochemistry         | -0.58                           | 0.23 <sup>b</sup>                    | Partitioning at the equator between southern                  | (0,11; 0.12)                                    |
|                                                                   | model                             |                                 | model                                                                                                                         |                                | model                                 |                                 |                                      | and northern hemispheres                                      |                                                 |
| Resplandy et al. (2018)                                           | Scaled-up river flux of           | $0.78\pm0.41$                   |                                                                                                                               |                                |                                       |                                 | $0.78 \pm 0.41$                      | Using IPSL GOBM partitioning;                                 | (0.38; 0.19; 0.20)                              |
|                                                                   | Jacobson et al. (2007) from heat  |                                 |                                                                                                                               |                                |                                       |                                 |                                      | e.g. Le Quéré et al. (2018b) and Friedlingstein et al. (2019) |                                                 |
|                                                                   | based constraint                  |                                 |                                                                                                                               |                                |                                       |                                 |                                      | Using Jacobson et al. (2007) partitionning;                   | $(0.03\pm0.02;0.38\pm0.19;0.36\pm0.17)$         |
|                                                                   |                                   |                                 |                                                                                                                               |                                |                                       |                                 |                                      | e.g. Resplandy et al. (2018)                                  |                                                 |
| Jacobson et al. (2007)6                                           | From global erosion model         | 0.71                            | From literature                                                                                                               | -0.10                          | From literature                       | -0.16                           | $0.45 \pm 0.18^{d}$                  | Partitioning shared in Resplandy et al. (2018)                | $(0.02\pm0.01;0.22\pm0.11;0.21\pm0.10)$         |
| Aumont et al. (2001)                                              | From a global erosion model       | 0.81                            |                                                                                                                               |                                | From an ocean biogeochemistry         | -0.2                            | 190                                  |                                                               | (0.30; 0.15; 0.16)                              |
|                                                                   |                                   |                                 |                                                                                                                               |                                | model to conserve the global          |                                 |                                      |                                                               |                                                 |
|                                                                   |                                   |                                 |                                                                                                                               |                                | Alk inventory                         |                                 |                                      |                                                               |                                                 |

 $^a$  Rounded at 0.05 PgC yr  $^4$  in their paper.  $^b$  Imbalance due to the region as well.  $^d$  e.g. Le Quéré et al. (2016, 2018a).

**Figure E1.** Literature-based estimates of the riverine discharge<sup>a</sup>, OM burial<sup>b</sup> and CaCO<sub>3</sub> burial, with their associated uncertainties/extremes through normalized PDFs.

# E2 Reassessing the regional distribution of the riverine/burial-driven air-sea carbon flux

We provide a schematic of the practical framework introduced in this manuscript (Sect. 4.2.1), which follows a four-step workflow to construct a composite simulation aligning with the most recent carbon and Alk budgets (Fig. E2).

In addition, we also share (Fig. E3) the various components of the composite simulated estimate creation process as described in Sect. 4.2.1, and the results of which are presented in Sect. 4.2.2 (see Fig. 7 and Table E2).

<sup>&</sup>lt;sup>a</sup>Including groundwater discharge for both carbon and Alk, and anaerobic processes Alk.

<sup>&</sup>lt;sup>b</sup>Including reverse weathering for Alk.

Figure E2. Schematic of the four-step workflow to construct the composite simulation. The composite simulated estimate of the riverine/burial-driven air-sea carbon flux is built from a linear combination of our sensitivity simulations to match the most recent carbon and Alk budgets (riverine discharge, OM burial, and  $CaCO_3$  burial). An additional step, equivalent to Step 3 but not shown in this general workflow, was necessary to adjust the  $CaCO_3$  burial of carbon in excess (Regnier et al., 2022), compared to the values accounted for Alk (Middelburg et al., 2020). In the schematic, the exponents indicate the source of each variable/parameter: E denotes the composite simulated estimate, E refers to the various sensitivity simulations, and E stands for literature estimates.

**Figure E3.** Components of the composite simulated estimate (supplement to Fig. 7). Each of the components represents the elements added at the different stages of the composite simulated estimate construction process (Steps 1, 2, 3, 4, and the extra step; Fig. E2). (a, b) Decomposition of the composite simulated estimate PDF associated with the total (a) carbon and (b) Alk external fluxes. (c, d, e, f) Characterization of the riverine/burial air-sea carbon flux in the composite simulated estimate, showing the various components for (c) the total value, as well as the (d) southern, (e) inter-tropical, and (f) northern regions. The black solid lines represent the total values for the composite simulated estimate, while the black dotted lines (a, b) correspond to the total carbon and Alk external fluxes from literature estimates (Fig. E1).

Code availability. NEMO is released under the terms of the CeCILL licence. The standard NEMO-PISCES version (PISCESv2; Aumont et al., 2015) slightly modified in this study (see Sect. 2.2.1) is accessible through https://www.nemo-ocean.eu (last access: January 2025). The other NEMO-PISCES versions are available on request from the corresponding author.

Data availability. All the CMIP ensemble data were available on at least one of the Earth System Grid Federation (ESGF) nodes https://esgf-node.ipsl.upmc.fr/projects/esgf-ipsl/ (last access: January 2025). The 2024 release of GCB data is currently available via their SFTP server upon request but is expected to become directly accessible soon through their data browser platform https://mdosullivan.github.io/GCB/ (last access: January 2025). The code of all the various configurations of NEMO-PISCES used in this study is accessible on Zenodo (https://doi.org/10.5281/zenodo.8421951), as well as two supplementary figures, S1 and S2 (https://doi.org/10.5281/zenodo.8421898).

Author contributions. This work is in the framework of the OMIP-BGC group, which contributed collectively to this study through the organization and execution of the CMIP exercises and the sharing of simulation outputs. AP: conceptualization, investigation, methodology, formal analysis, visualization, writing – original draft preparation – and project administration. LB and LK: supervision, funding acquisition, methodology, resources, conceptualization, and writing – original draft preparation.

750 Competing interests. The contact author has declared that none of the authors has any competing interests.




*Disclaimer.* This article reflects only the authors' views; the funding agencies and their executive agencies are not responsible for any use that may be made of the information that the article contains.

Acknowledgements. We would like to express our gratitude to Pierre Regnier and an anonymous reviewer for their careful and constructive review which greatly contributed to improving the quality and clarity of our work. We are also grateful to Jack Middelburg for being the associate editor. Thanks to Olivier Aumont and Olivier Sulpis for the discussions we had, respectively, regarding the evolution of the riverine/burial-driven air-sea carbon flux assessment, and the burial of CaCO<sub>3</sub>, as well as the global Alk inventory. We are grateful to the World Climate Research Programme's Working Group on Coupled Modelling, which is responsible for the CMIP exercises. For CMIP, the U.S. Department of Energy's Program for Climate Model Diagnosis and Intercomparison provided coordinating support and led the development of software infrastructure in partnership with the Global Organization for Earth System Science Portals. This study benefited from the ESPRI (Ensemble de Services Pour la Recherche à l'IPSL) computing and data centre (https://mesocentre.ipsl.fr, last access: January 2025), which is supported by CNRS, Sorbonne Université, École Polytechnique, and CNES and through national and international grants. We are grateful to the GCP for leading such a community effort around the carbon cycle, especially with yearly inter-model comparison exercises. We thank the PISCES community for actively using, developing, sharing knowledge around the marine biogeochemical model NEMO-PISCES (https://www.pisces-community.org, last access: January 2025). I would like to express my appreciation to the CEA (Commissariat

à l'Énergie Atomique et aux énergies alternatives) and James C. Orr, the project coordinator of 'gen0040', for granting us access to the TGCC (*Très Grand Centre de Calcul*), particularly allowing us to conduct all our simulations on their 'Irène Rome' machine. Finally, we are also grateful to the administrative and technical staff at École Normale Supérieure/PSL.

Financial supports. Alban Planchat, Laurent Bopp and Lester Kwiatkowski are grateful to the ENS-Chanel research chair.

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
