# Peer review of "A fresh look at the pre-industrial air-sea carbon flux using the alkalinity budget"

_EGUsphere, 2025_

## Referee Comment (RC2)

In their manuscript, Planchat et al. investigate the pre-industrial $CO_2$ outgassing in the ocean, with a particular focus on constraints from processes that affect the ocean alkalinity budget. They thereby combine a theoretical model, which budgets alkalinity from riverine source and burial sink and also accounts for a potential drift, model simulations, including a great number of sensitivity simulations, as well as a practical framework which aims to reconcile the sensitivity simulations with current river input and burial estimates. The authors have obviously put a great amount of thought and effort into the development of these tools and simulations, and the work generally seems seems and robust from a technical stand point. However, the limitations of the model and framework (and I assume there are many), are not transparently discussed. The paper is also extremely hard to follow in its current presentation, and would not be well suited for general BG readers (and arguably even experts on the topic), in its current form. I therefore recommend major revisions, which however mostly involve streamlining and improvements in the explanations (especially regarding methodology). I would also recommend potentially re-thinking the amount of frameworks and results the authors wish to present. I made some suggestions and comments, which I hope can help improve the paper.

**Major Comments**

- The paper and especially individual sections need a better structure and streamlining. The paper is covering a large amount of novel material: a new-ish theoretical and a new practical framework, almost uncountable amounts of sensitivity experiments, GCB results, CMIP7 results, a literature review, and then new theories in addition popping up in the results section.It would be important that the methods and results are presented in a way that the general reader can somewhat follow and understand. For individual sections, it's often not really clear what the main message of the section is (often multiple complex and only sparsely explained points, also often unconnected results). Sensitivity experiments are introduced in the results section without having been properly explained. I also believe certain aspects could be simplified (maybe inspired at how previous studies explain the alkalinity/carbon ocean budget, interhemispheric transport), but would leave it up to the authors to decide.

- I think it would be very helpful for the general reader if the paper were to introduce the "geologically-known" balance of carbonate alkalinity from river inputs to burial in the ocean (see e.g. Hartmann et al., 2013), I guess early in the methods section:

$$Ca^{2+} + 2HCO_3^- \rightarrow CaCO_3 + CO_2 + H_2O$$

With the left-hand side the river transports, and the right the exports from the ocean. This is essential to understand and might make it easier to follow the equations described in section 2.1. I think this is in essence what is meant throughout the paper when referring to a "simple budget" in the results section, although it wasn't 100% clear how this differs to the theoretical framework explained in the methods. The equations (atleast to me) explain in a simpler way (atleast for me) the difference of balance of alkalinity:carbon from the river inputs and burial, and also explains why e.g. CO2 outgassing increases with CaCO3 burial (to me these explanations in the results section will be hard to follow for a general reader).

- The methods section is lacking a general overview on the strategy, which I think is needed due to the numerous frameworks/simulations and complexity of the topic. I would clearly explain from the start why there is the need to develop a a theoretical framework, then run simulations with sensitivity experiments, then another practical framework (and to which purposes the different frameworks are needed)? Why not just improve the model to perform better?

- In the methodology and even after reading the whole text, it is not really clear what the exact purposes of the individual sensitivity experiments needed. I understand they are needed for the practical framework, but why these specifically ones, would the results be different actually if you had chosen other sensitivity experiments / left out some? The results from the sensitivity experiments are brought up every now and then in the paper, but the general thinking behind these experiments, and clear explanations and reasonings behind the simulations would help a lot from the methods section.

- Regarding the strong focus of the paper on Global Carbon Budget numbers and history (and a call for potential revision?): I am not sure improving the observations to model discrepency of the Global Carbon Budget should really be a central evaluation metric for the results. There are now a few other major issues that have been identified that could explain (a large part?) of the bias between observational and model products (model biases: Terhaar et al., 2022 observation biases: Landschützer et al. 2023). In the abstract it seems like the authors are strongly calling for a revision of GCB estimates based on their numbers, but in the main text and especially in the conclusion, the authors don't seem quite as confident in these mean numbers, highlighting large uncertainties. Personally, the study does an excellent job at highlighting uncertainties in current knowledge and estimates, but the limitations in terms of the model used and the observational basis for burial/rivers are the same/similar as of previous work. I think would probably make more sense to try to motivate for better models and observations for reducing these uncertainties.

- Limitations: The limitations from the both simulations and practical should be made more transparent. For instance, the model likely very poorly represents the processes in the coastal ocean (CaCO3 production there?) and its transport to the open ocean, doesn't represent terrestrial organic matter etc. . I think the practical framework doesn't distinguish between more realistic sensitivity simulation assumptions, but only aims to reduce the bias to the observations, for me this seems kind of an important caveat (so might be getting things "right-er" for wrong reasons).

- Regarding the impacts of an alkalinity imbalance: The paper really lays a large focus on the potential impacts of an alkalinity imbalance, and largely motivates a major part of the frameworks introduced. But if I understand correctly, this imbalance is only a possibility and its even uncertain in which direction this imbalance would be?

- Regarding the spatial variability of the preindustrial outgassing: The authors acknowledge that the fate of terrestrial organic matter plays a major role in explaining spatial patterns of river-induced outgassing in the ocean. However, since this paper doesn't really investigate constraints or uncertainties regarding the fate of terrestrial organic carbon in the ocean I wouldn't really be particularly confident in the spatial distribution derived from the framework presented here. While this seems to reduce biases in the Southern Ocean between model and observational means, I would argue this is a region that is both one of lacking observational data, and complex region for models to simulate. Recent estimates of OM lability generally seem to suggest rapid degradation, even already in the coastal ocean, which would go against significant transport to and outgassing in the Southern Ocean (Aarnos et al., 2019; Lacroix et al., 2021; Powley et al., 2024).

**Specific Comments**

**Abstract**

L1 "The disparities in estimates of the ocean carbon sink, whether derived from observations or models…"

Did the authors not mean: "The disparities in the estimates of the ocean carbon sink between observations and models ..

"raise questions about our ability to accurately assess its magnitude and trend over recent decades. "

For me this is too strong of a statement.

L17 "Addressing the current inconsistencies between the combined carbon and alkalinity budgets.." What these inconsistencies are hasn't been mentioned yet, so I would tend to leave this out here (or briefly explain what is meant beforehand).

L19 "… and intermodel comparisons are required to constrain its regional distribution." I would argue that improving the models regarding the fate of terrestrial carbon in the ocean would also be needed (maybe even as a more urgent priority).

**Introduction**

L21-L26 As mentioned, I'm not sure I would make the discrepency between observations and model estimates the central motivation here, since many other factors have been put forward recently that may also explain the difference.

L40 "The spatial distribution of this riverine/burial-driven air-sea carbon flux is also highly uncertain and depends on the assumptions and methods used for its assessment (see Table B2). "

I would expand on which important assumptions are referred to here.

L41 "The historical value.."

This term is kind of confusing here, historical=pre-industrial ?

L48 "These disparities have emerged and disappeared without apparent reason.."

L49-L54 I believe these strategic changes were made using the best and most recent science (and not with the primary goal of reducing the discrepency). I don't think this qualifies as "without apparent reason".

L57-L60 I would maybe add information whether the CMIP6 models even consider riverine fluxes. I would guess not, and in a probably very simplified manner.

**Methods**

2.1 -> I guess this section describes the theoretical framework mentioned previously? I would make this more clear (not really sure overview and definitions is really a fitting subtitle).

L140 "this conceptual tool only permits estimates of a potential air-sea carbon flux resulting from a local/regional carbon:Alk imbalance"

Please clarify what is meant by "potential air-sea carbon flux".

L142-143 "There is indeed no guarantee of its local/regional applicability due to the transport of the induced carbon:Alk local/regional imbalance. "

Could you maybe explain its applicability then, or how exactly you are dealing with this uncertainty?

L208 "The inorganic fraction is supposed to be in the form of bicarbonate ions and thus affects both DIC and Alk in a similar manner. "

Is it supposed to be or is it added as such?

L210 This simulation also includes the burial of OM and CaCO3 produced by pelagic organisms, which is exported to the ocean interior and only partially remineralized or dissolved in the water column and at the seafloor.

Since this paper focusses on alkalinity fluxes and its burial, I would recommend giving a brief overview of how these processes affect the alkalinity balance (and especially how the burial is parametrized).

L231-244 The sensitivity experiments are described here, but it is not really clear why they are (individually) needed.

L248 "For instance, the gap in the CaCO3 burial (0.24 versus 0.35 PgC yr$^{-1}$) would drive a 0.22 PgC yr$^{-1}$ difference in the Alk budget. "

Isn't this a difference in the carbon budget? (0.22 Pg *C* yr-1)

Figure 4: "The negative impact of OM burial on Alk is attributed to the release of ammonium when OM is remineralized at the seafloor rather than buried."

I am not sure this information belongs in the caption, but in the text.

L302 "This residual component is attributed to a residual carbon budget imbalance due to internal ocean processes."

I guess this would be a model drift?

L309 "Role of sediment burial fluxes"

This section was really tough to read and understand, it has a lot of unconnected (complex and not really explained) information put together, it is really unclear in terms of streamlining -> maybe try to revise?

L310 "At global scale, the riverine carbon flux in the standard simulation ($+0.52$ PgC yr$^{-1}$) is insufficient to fully account for the air-sea carbon outgassing ($0.27$ PgC yr$^{-1}$) alone (Fig. 4d and 5a). "

I think most readers will not understand what is meant here (actually I am also not sure), since 0.52 PgC yr-1 is larger than 0.27 PgC yr-1.

L315 "Increasing the river input by a factor of 1.5 (riv1p5)"

Increasing all species of carbon? Wasn't clear to me from the methods, and would be quite important to know to understand the increase in carbon outgassing?

L324 "This underscores the significance of considering CaCO3 burial when constructing the ocean carbon budget, especially when evaluating the net air-sea carbon flux. "

Yes but I would argue understanding the fate of terrestrial OM and its burial is equally important, in terms of magnitudes atleast?

L325 "The total carbon outgassing obtained from a simple carbon budget only holds however for our simulations in which the global Alk inventory is in equilibrium"

I think you should explain which of these fluxes above are the simple carbon budget? To me they seem coming from the simulation.

L334 Why are the regional burial fluxes actually discussed in the "imbalanced" alkalinity section? Is it not possible that these large regional burial fluxes are actually in quasi-equilibrium with the river fluxes?  If you are removing alkalinity in these places artificially, would you not have less alkalinity burial in other places as a result in the model?

L373 "Part of this discrepancy arises because atmospheric carbon uptake by continental shelves ($0.10$ PgC yr$^{-1}$ Regnier et al., 2022) is fully integrated into our net pre- industrial

riverine/burial-driven air-sea carbon flux as we also consider OM and CaCO3 burial in these regions, reducing this flux by 0.10 PgC yr$^{-1}$."

The OM and CaCO3 burial on continental shelves will likely not be well resolved at the applied model resolution (+ not taking into account coastal specific processes such as benthic CO3 producers etc. ).

L426-L440 This section was really heavy and hard to understand.

L485 "Additionally, a better understanding of the intrinsic properties of these external fluxes is needed, such as whether the organic carbon brought by rivers is refractory or not (Aumont et al., 2001; Jacobson et al., 2007; Gruber et al., 2009)."

Improving the understanding on the fate of the terrestrial organic carbon is probably equally important. I don't think its impacts are resolved by degrading the organic matter at the river mouths as in such models.

References:

Aarnos, H., Gélinas, Y., Kasurinen, V., Gu, Y., Puupponen, V.-M., & Vähätalo, A. V. (2018). Photochemical mineralization of terrigenous DOC to dissolved inorganic carbon in ocean. Global Biogeochemical Cycles, 32, 250–266. https://doi.org/10.1002/2017GB005698

Hartmann, J., A. J. West, P. Renforth, P. Köhler, C. L. De La Rocha, D. A. Wolf-Gladrow, H. H. Dürr, and J. Scheffran (2013), Enhanced chemical weathering as a geoengineering strategy to reduce atmospheric carbon dioxide, supply nutrients, and mitigate ocean acidification, Rev. Geophys., 51, 113–149, doi:10.1002/rog.20004.

Hauck, J., Nissen, C., Landschützer, P., Rödenbeck, C., Bushinsky, S., & Olsen, A. (2023). Sparse observations induce large biases in estimates of the global ocean CO2 sink: An ocean model subsampling experiment.

Lacroix, F., Ilyina, T., Laruelle, G. G., & Regnier, P. (2021). Reconstructing the preindustrial coastal carbon cycle through a global ocean circulation model: was the global continental shelf already both autotrophic and a CO2 sink?. Global Biogeochemical Cycles, 35, e2020GB006603. https://doi.org/10.1029/2020GB006603

Powley, H. R., Polimene, L., Torres, R., Al Azhar, M., Bell, V. A., Cooper, D., ... & Artioli, Y. (2023). Modelling terrigenous DOC across the north west European Shelf: Fate of riverine input and impact on air-sea CO2 fluxes. Science of The Total Environment, 912, 168938.

Terhaar, J., Goris, N., Müller, J. D., DeVries, T., Gruber, N., Hauck, J., et al. (2024). Assessment of global ocean biogeochemistry models for ocean carbon sink estimates in RECCAP2 and recommendations for future studies. Journal of Advances in Modeling Earth Systems, 16, e2023MS003840. https://doi.org/10.1029/2023MS003840

---

## Author Comment (AC1)

**Manuscript revision**
**A fresh look at the pre-industrial air-sea carbon flux using the alkalinity budget**

Alban Planchat, Laurent Bopp, and Lester Kwiatkowski

*We have carefully addressed the reviewers' comments in our revision and would like to express our gratitude to both reviewers for their careful and constructive evaluations. Their insights have greatly contributed to improving the quality and clarity of our work. Below, we provide a detailed point-by-point response to all reviewers' comments.*

Before addressing each comment individually, we would first like to provide a general response outlining the major revisions made to the manuscript.

The paper has been extensively restructured. The 'Methods' section has been significantly shortened and clarified. Additionally, we have introduced a new 'Results' section, separate from a new 'Proof-of-concept applications and discussions' section, in order to clearly distinguish the results – particularly the validation of the theoretical framework through our sensitivity simulations – from the proof-of-concept applications that follow. The revised paper structure is:

1. Introduction
2. Methods
   2.1. Theoretical framework
      2.1.1. Governing equation of the pre-industrial riverine/burial-driven air-sea carbon flux
      2.1.2. Regional proxies for the spatial distribution of the pre-industrial riverine/burial-driven air-sea carbon flux
         *The concept of the regional carbon:Alk budget imbalance*
         *The inter-hemispheric flux gradient and transport*
   2.2. Model and simulations
      2.2.1. Model and configuration
      2.2.2. Standard simulation (std) and its riverine/burial component
      2.2.3. Sensitivity simulations
3. Results
   3.1. Pre-industrial air-sea carbon flux and its riverine/burial-driven component
   3.2. The global riverine/burial-driven air-sea carbon flux
      3.2.1. Role of sediment burial fluxes
      3.2.2. Impact of an imbalanced alkalinity budget
      3.2.3. Validating the governing equation of the pre-industrial riverine/burial-driven air-sea carbon flux
   3.3. The spatial distribution of the riverine/burial-driven air-sea carbon flux
      3.3.1. Contrasting regional fluxes
      3.3.2. Influencing factors
4. Proof-of-concept applications and discussion
   4.1. The global flux
      4.1.1. Approach
      4.1.2. Findings
      4.1.3. Discussion
   4.2. The flux distribution
      4.2.1. Approach
      4.2.2. Findings
      4.2.3. Discussion
5. Conclusion and perspectives

**Reviewer #1**

The ms. By Planchat et al. provides a new theoretical framework to analyze the magnitude and spatial distribution of the pre-industrial ocean outgassing. As explained by the authors, the topic is important because it allows to compare and eventually reconcile the magnitude of the anthropogenic ocean sink from models and observations and, in particular, the role of the inter-hemispheric carbon transport in shaping the spatial pattern of the outgassing. The topic fits well within the scope of Biogeosciences, the research is a novel contribution and important contribution to the field and, overall, the approach appears sound and robust. Yet, I have two main comments that I believe need to be addressed (I would say, moderate revisions):

**Main comments:**

- If I am overall convinced about the new framework, I am less convinced about the quantification that results from its application. This is important as there is a risk that the community will mostly remember the one sentence in the abstract that provides the new quantitative assessment of the pre-industrial ocean outgassing (0.49 PgC yr-1). I would thus suggest toning down this statement to better reflect the uncertainties associated with the quantification. Indeed, as explained by the authors in the main text, the assessment is  broadly consistent (by construction) with the one by Regnier et al. (0.65 – 0.1 for coastal) = 0.55 PgC yr-1, the remaining difference (0.06 PgC yr-1) being the contribution of the Alk imbalance. However, taken the fact that external sources/sinks of Alk (and their fate) are poorly constrained together with a C -Alk budget currently partly inconsistent, the impact of the disequilibrium on the air-sea flux remains highly uncertain in quantitative terms.Probably one of the best example is the uncertain contribution of the river PIC contribution to ocean Alk (as acknowledged by Middelburg, 2020), a flux that has for instance been significantly reduced in the recent work by Liu et al., 2024 (Nature Geoscience, mean value of 2.5 Tmol/yr). I am not suggesting here that the overall budget should be revised with new C-Alk external numbers (as I do acknowledge that this does not question the value of the framework), but I would be less assertive regarding the quantification and the fact that it helps reconciling observations and models, including in the abstract. I further elaborate on this point later in the review.

Thank you for this comment, which echoes some of the concerns we had during the preparation of the submitted manuscript.

   We fully agree on the importance of ensuring that the community does not focus primarily on the numerical estimates we present. As you point out, our core contribution lies in the validation of the theoretical framework we introduce, followed by two proof-of-concept applications.

   In response to this concern – and in addition to a major restructuring of the manuscript – we have rewritten both the 'Abstract' and the 'Conclusion and perspectives' sections. In these key parts of the paper, we now emphasize the strength and flexibility of the theoretical framework and its applications, rather than the specific results of our reassessments. Our primary objective is to encourage the community to adopt and build upon this methodological approach.

- I do very much appreciate complex science, but on this occasion, I found that it was sometimes hard to follow the argumentation developed in the paper. I hope that I have properly understood (most of) the proposed framework, but remain concerned about the overall 'accessibility' of the research to the broad readership of Biogeosciences. Part of the issue likely stems from choices made at the beginning of the research project, which were then adjusted later on (e.g., choice of external fluxes for the std simulation which are then 'corrected' through a series of steps that are not fully transparent). This makes the overall ms. highly complex and I recommend that the authors make an extra effort to improve the clarity of their ms. This is important in the context of my first major comment (readers might focus on key information in the abstract, without getting into the details of the complex ms.). Several of my comments that follow are suggestions in this direction.

We fully acknowledge the difficulty we initially encountered in conveying the results of this study in a

clear manner. We agree with your assessment that the submitted version of the manuscript was, at times, difficult to follow.

In response, we undertook a substantial restructuring of the manuscript, accompanied by an in-depth rewriting effort. This revision has greatly improved both the clarity and accessibility of the paper. While we recognize that the topic remains inherently complex, we now feel confident – thanks in large part to your constructive feedback – that the core ideas and associated messages of the study are now clearly articulated and accessible to the broad readership of *Biogeosciences*.

**Specific comments:**

- Line 47: "without apparent reasons". This statement is not true. Reassessments have essentially followed new evidences from the literature.
We agree that the original wording was incorrect and have accordingly revised the paragraph:
*"Uncertainties in estimating the riverine/burial-driven pre-industrial outgassing may contribute to the persistent discrepancies between observation-based and model-derived estimates of the anthropogenic ocean carbon sink, both globally and regionally (Friedlingstein et al., 2024, their Fig. 11 and 14). These disparities have fluctuated over time, largely in response to stepwise adjustments made by the GCB team following new reassessments of the magnitude and spatial distribution of this flux in the literature (Fig. 1, Table 1). For instance, a substantial decrease in the global estimate of the pre-industrial outgassing from 2019 to 2020 contributed to a notable narrowing of the global observation–model gap. More recently, from 2022 to 2023, a redistribution of the flux between regions, from the southern region to the tropics, led to a reduced southern hemisphere bias and a compensating increase in the inter-tropical discrepancy."*

- Fig.1: define GOBM in figure caption
It is now mentioned.

- Lines 58-60: a few words of explanation regarding the possible underlying reasons for the major discrepancies in global air-sea flux and its spatial distribution across CMIP models would be useful.
Only a few words were added as GOBMs and CMIP6 ESMs were not investigated in depth and solely used for an illustrative purpose: *"likely due to differences in model setups and various/incomplete representations of sediment burial and riverine discharge (Terhaar et al., 2024)".*

- Lines 68-75 The Alk disequilibrium is central to this research but the introduction does not provide much explanation as to why the budget would have been outside equilibrium under pre-industrial times. I suggest elaborating on this aspect briefly.
The content of the paragraph was adjusted to clarify the message and briefly explain the specificity of the Alk budget:
*"Nonetheless, the hypothesis of an imbalanced Alk budget during the pre-industrial era remains plausible, based on estimates of global Alk sources and sinks (e.g. Milliman, 1993; Middelburg et al., 2020). Paleoclimatic studies suggest that such an imbalance could arise from additional CaCO3 burial (e.g. Cartapanis et al., 2018) or from a carbonate compensation mechanism involving biological processes alongside riverine inputs (Boudreau et al., 2018). Unlike carbon, Alk is not exchanged with the atmosphere, so balancing its budget depends on processes acting over longer timescales, particularly through interactions with the continents (e.g. erosion) and marine biogeochemistry (e.g. sediment dynamics). As a result, Alk budget balancing is slow, yet interactive with the carbon cycle through changes in atmospheric CO2 and ocean acidity (Hain et al., 2014). An imbalance in the Alk budget would induce an additional air–sea carbon flux beyond that directly inferred from the ocean carbon budget, ultimately resulting in a non-conserved global ocean carbon inventory."*

- Line 79: what is a "practical framework" ?
This final paragraph of the introduction was fully rewritten to align with the revised structure of the manuscript. We no longer refer to a 'practical framework' which we agree was confusing terminology.

- Line 88: should it not be (by convention) "positive fluxes are directed into the ocean"
This has been adjusted.

- Line 92-93: This is not true, I believe. The values (at least for C) selected later on in the ms. are for pre-industrial times, including burial.
You are right that this is not true. Assuming a steady state in pre-industrial times, recent changes in riverine and burial fluxes could, to a first approximation, be associated with human-induced disturbances. In this manuscript, however, we focus solely on the pre-industrial period for the sake of clarity. We have therefore reformulated two sentences in this paragraph accordingly:
*"Within the anthropogenic carbon flux, we incorporate the perturbation of the natural carbon flux in response to climate change, ensuring that F C, air–sea ant. fully reflects the carbon sink resulting from all human-induced disturbances (e.g. Hauck et al., 2020). Accordingly, F C, air–sea nat. is directly defined as the riverine/burial-driven pre-industrial air-sea carbon flux (F C, air–sea riv./bur.), i.e.:"*
    It is worth noting that the content of this manuscript, particularly the theoretical framework, could also be relevant/discussed in the context of human-induced disturbances.

- Line 98: "When assuming a global Alk inventory equilibrium". I am not sure that this is the best way to introduce the concept. In reality, the key assumption here is rather that the approach assumed no accumulation of C in the ocean (i.e., C inputs and outputs are balanced), without any assumption regarding Alk. Rather, I understand the Alk constrain as one used to account for the possibility of a time variant ocean C inventory during pre-industrial times.
Thank you for sharing this perspective. We have taken this comment into account while maintaining a causal logic centered on the Alk budget, which governs whether the ocean carbon budget is balanced or imbalanced. Accordingly, we have reformulated the sentence as follows:
*"Assuming a steady-state pre-industrial air–sea carbon flux and a balanced Alk budget, the global riverine/burial-driven air–sea carbon flux can be directly inferred by closing the ocean carbon budget (e.g. Regnier et al., 2022):"*
    In addition, based on your comments, we clarified at the very beginning of the 'Methods' section how we define the notions of 'steady state', '(im)balance', and '(dis)equilibrium'. From that point on, we consistently rely on this framework throughout the manuscript.
*"In this study, we use 'steady-state' to refer to the temporal stability of the globally integrated air–sea carbon flux. We describe the carbon and Alk budgets as 'balanced' or 'imbalanced' according to whether fluxes into and out of the ocean are quantitatively balanced. An imbalanced budget drives a deviation in the global inventory: a positive (resp. negative) imbalance leads to an increase (resp. decrease) in the global inventory. Unless otherwise stated, all simulations and results refer to pre-industrial conditions."*
    We further clarify it in the context of our sensitivity simulations at the end of Sect. 2.2.3:
*"We classify our simulations as either 'equilibrated' (suffix '-eq') or 'disequilibrated' (suffix '-diseq'), based on the conservation of global carbon and Alk inventories. In both cases, the air–sea carbon flux has reached a steady-state. Equilibrated simulations are characterized by balanced global carbon and Alk budgets, resulting in conserved inventories over time. In contrast, disequilibrated simulations exhibit imbalanced budgets, leading to evolving global inventories, which are therefore not conserved."*

- Line 105: this is an example where it would be useful to have the comment on line 88 implemented
Comment line 88 has been implemented.

- Figure 3 caption: theoretical (typo). Why primes for the variables here ?
The primes on the variables were removed, as we have completely revised and simplified our treatment of potential fluxes to reduce the manuscript's complexity and enhance clarity.

- Figure 3 caption (line 2): "associated local imbalance". Imbalance of what ? Check this here and everywhere else in the text

Beyond being restructured, the 'Methods' section has also been partially rewritten and simplified. We now take the time, in a dedicated subsection (Sect. 2.1.2), to clearly define the concept of the regional carbon:Alk imbalance.

- Figure 3 caption (line 3): disequilibrium in Alk and DIC inventories. I suggest adding "inventories"
This has been adjusted.

- lines 113-118 and Figure 3 caption (line 6): "CO2 concentration, and requires an air-sea carbon flux to restore equilibrium" "… to achieve equilibrium". I am here confused with the use of "equilibrium". By virtue of the pre-industrial outgassing, the ocean is not in a state of equilibrium with the atmosphere. The whole concept is thus hard to grasp (do you mean steady-state ?). And is it straightforward that the ocean as a whole would have reached a state of "equilibrium" with the atmosphere ? Is it an assumption ? See also the following comment.
We have rephrased one central sentence to clarify:
*"Importantly, under the assumption of a steady-state system – that is, with a constant air–sea carbon flux –, any imbalance in the Alk budget induces a compensating carbon flux."*
        See also response to 'Line 98' comment.

- lines 118-122: I glanced through both Humphreys and Planchat et al. (2023) and could not derive the proposed "model". I recommend adding (in SI) more details elaborating on the conceptual model (including any underlying assumption needed to derive the key equations), taken its central role for the ms. In particular, is the Humphrey's approach (chemistry only) applicable to a framework where the ocean physics is explicitly incorporated ?
A full 'Appendix' section (Appendix A) is now dedicated to developing the method proposed by Planchat et al. (2023) to define Qinv. We also demonstrate its full consistency with the approach employed by Humphreys et al. (2018).
        The expression of Qinv naturally varies locally, depending on Alk and DIC values, and therefore on both physical and biogeochemical processes. However, as shown in Planchat et al. (2023, our Fig. 12a), this coefficient proves extremely robust across all CMIP5 and CMIP6 ESMs. This is also something we noted among our sensitivity simulations. At the global scale, it thus appears entirely reasonable to adopt a mean Qinv value, as we do in this study.

- line 125: This I suspect implies a framework where the land-atmosphere and ocean-atmosphere were both in steady state (along with a time invariant atmospheric C inventory) during pre-industrial times, yet with a disequilibrium state in the land and ocean C inventories through the land-ocean route (as materialized by the Alk imbalance ?). If correct, I believe that this aspect should then be better presented and discussed (e.g., are there any observational evidences for a time variant C inventory on land compensating the one in the ocean) ?
Our study is conducted within the framework of an invariant atmospheric carbon inventory and a steady-state ocean (i.e. stable air–sea carbon flux; e.g. Orr et al., 2017, see Fig. B2). Within this context, any carbon release from the ocean to the atmosphere must be balanced by an equivalent carbon uptake by the land. This principle is illustrated in Resplandy et al. (2018, their Fig. 5b).
        Interestingly, if we model the system with only two boxes for the land, ocean, and atmosphere (the northern hemisphere (lat>0°) and the southern hemisphere (lat<0°)), then knowing the inter-hemispheric atmospheric CO2 gradient (and thus the inter-hemispheric flux gradient,

defined as $G^{atmo} = F_N - F_S$, where $F_{N/S} = F_{N/S}{}^{land} + F_{N/S}{}^{ocean}$), and assuming an invariant atmospheric carbon inventory ($F_N + F_S = 0$), we can derive the following relationships:

$$G^{atmo} = -2F_S = 2F_N$$

This allows us to express the northern (resp. southern) air-land carbon flux as a function of the northern (resp. southern) air-sea carbon flux:

$$F_N^{land} = \frac{G^{atmo}}{2} - F_N^{ocean} \quad \text{and} \quad F_S^{land} = -\frac{G^{atmo}}{2} - F_S^{ocean} \; .$$

However, this system of equations becomes underdetermined if we introduce a third, intertropical box for the ocean, land and atmosphere.

- line 142: again here a sentence where the notion of an "equilibrium" constrain remains unclear.
See response to 'Line 98' comment.

- line 150: typo "carbon flux as well as carbon and Alk disequilibria ". Please clarify what you mean by "potential" fluxes.
The concept of "potential" fluxes occupe maintenant une place très marginale dans le papier. Nous préférons principalement parler de regional carbon:Alk imbalance, un concept less confusing et défini dans la Sect. 2.1.2:
*"To gain a deeper understanding of the factors shaping the spatial distribution of the riverine/burial-driven air-sea carbon flux, we expand upon the theoretical framework previously outlined for the global scale (refer to Section 2.1.1) and adapt it as a proxy for application at regional scales. This is essential to understanding the extent to which specific regional carbon:Alk budget imbalances can drive the global air-sea carbon flux as well as deviations in carbon and Alk inventories.*
*The air-sea carbon flux calculated by applying Eq. 10 to the riverine and burial fluxes of a specific region can only be considered a potential air–sea carbon flux, in the sense that it represents the capacity to generate such a flux at the global scale, with no guarantee that this flux is fully expressed within the exact same region. Due to ocean circulation and the associated transport of carbon and Alk, regional carbon:Alk budget imbalances in riverine and burial fluxes explain the regional distribution of the drivers of the global air–sea carbon flux. However, they only partially explain the regional distribution of the flux itself."*

- line 168: "and assuming an ocean at equilibrium" (fig. B1). Fig B1 rather shows that the air-sea fluxes are in steady-state (in contrast to the C / ALK inventories). I think that this central idea (if correct) is not well introduced in general in the ms. Also why "assuming" ? Is this something that you have imposed in your simulations or is it arising from the model behavior ?
See response to 'Line 98' comment.

- Line 178: I could not derive eq. 19 in a straightforward way (I understand it, though).
This is now part of the 'Appendix' to simplify the 'Methods' section, and a step was added to share more details regarding the calculation.

- Line 207: your std simulations clearly rely on not up to date (and very low) river C inputs. What about POC ? I understand that you attempt a correction for this later on, but the whole procedure remains unclear and not sufficiently transparent to me (see comment section 2.3 below).
It is important to note that our goal was not to run a simulation using an ideal configuration. Rather, our objective – building on the theoretical framework we establish and validate through our sensitivity simulations – was precisely to enable the estimation of the riverine/burial-driven air–sea carbon flux, along with its spatial distribution, without the need for new simulations. Our two proof-of-concept applications illustrate the flexibility of this theoretical framework, making it a powerful tool to efficiently address future re-evaluations of riverine and burial fluxes, as well as associated processes (e.g. the fate of terrestrial organic matter).
We clarified the purpose and objectives of our simulations in the first paragraph of Sect. 2.2.3:
*"The set of sensitivity simulations considered covers a broad range of perturbations to the carbon and Alk riverine and burial fluxes. These simulations aim to assess the effects of different assumptions regarding these external fluxes on the riverine/burial-driven air-sea carbon flux (Table 2, Fig. 2b; see Appendix B2.1). Importantly, within the context of our study, the absolute values of the fluxes –*

*whether they align with literature estimates or not – are not of primary concern. What matters are the relative differences between these values across simulations, which reflect the assumptions being tested."*

- Line 239: here you use 'steady-state' for the atmosphere, which I believe is clearer (see also comments on lines 113-122).
You are right, and the manuscript was adjusted to take this remark into account.

- section 2.3 and associated figure: I do not follow well the composite simulation strategy and its potential implications.
The structure of the paper has been thoroughly revised to avoid losing the reader, especially within the lengthy 'Methods' section, and to clearly distinguish the 'Results' from the 'Proof-of-concept applications and discussions'. It is in the latter that we now introduce the strategy for constructing the composite simulation, which makes much more sense in this context (Sect. 4.2.1). Besides, we now give more details to clarify the strategy (see also Fig. E2).

- Figure 4 caption: CaCO should be CaCO3. Please correct here and elsewhere.
Most certainly an issue with the shared pdf from EGU Sphere. No issue in our manuscript.

- Figure 4 caption:  I do not understand to what exactly the contribution from local (or regional as in the figure) imbalance is referring to and how these values were constrained (it is not discussed in the text. Or is it lines 298-300 ? But I don't get the math here). Does the "imbalance" always refer to the Alk:DIC imbalance ? If yes, this has to be specified here and elsewhere.
Fig. 4b was partly modified to only show the simulated and riverine/burial-driven air-sea carbon fluxes. In addition, the concept of the regional carbon:Alk budget imbalance and the associated potential air-sea carbon flux is now clarified and better introduced in the 'Methods', Sect. 2.1.2.

- Lines 323-324: the statement appears obvious to me (is it not textbook knowledge that CaCO3 burial impacts the C budget and air-sea CO2 exchange ?). I am not sure what is precisely meant here.
The wording of the entire paragraph has been improved for clarity, and the final sentence has been modified to better reflect the underlying concept: *"This highlights the pivotal role of CaCO3 burial in shaping the air-sea carbon flux under the assumption of a balanced Alk budget, where riverine Alk inputs are offset by CaCO3 burial (Fig. 4d)."*

- Line 337: the description of the two disequilibria sensitivity runs induce an extra outgassing (due to a higher CaCO3 burial; I suspect that you should have a negative Alk disequilibrium here ?). These sensitivity runs thus operate in the opposite direction to the one in section 3.2.3 that follows (where the Alk imbalance induces a reduction of the outgassing). I found this really confusing. I suggest elaborating also on a sensitivity run that goes in the same direction as in 3.2.3 or at least - if I understood correctly-, I would stress this aspect more clearly (i.e. the Alk disequilibrium can generate increasing or decreasing air-sea CO2 exchange) – see also comment on lines 377-80 below.
Indeed, an imbalanced Alk budget can lead to either an increase or a decrease in the pre-industrial air-sea carbon flux, depending on the sign of the imbalance. The equation governing the pre-industrial riverine/burial-driven air-sea carbon flux remains valid regardless of the Alk imbalance (positive, negative, or even neutral).
        Furthermore, restructuring the paper by separating the 'Results' from the 'Proof-of-concept applications and discussions' allows us, first, to validate this equation using our sensitivity simulations, and then to apply it independently of those simulations. The value of the tool we introduce in this manuscript becomes evident through the proof-of-concept applications.

Line 338-39: why counterintuitive ? – Is this not the effect of the carbonate counter pump (more carbonate precipitation and preservation inducing a CO2 source to the atmosphere) ?
Absolutely! However, this remains counterintuitive for the vast majority of people, including many within the marine biogeochemistry community. We believe it is important to emphasize this

counterintuitive aspect, as the role of the carbonate pump and Alk is still too often overlooked or insufficiently considered. The whole paragraph was rewritten to enhance clarity, especially the end:
*"This outcome may seem counterintuitive when applying a simple carbon budget, since both simulations prescribe extra carbon removal from the ocean (to the sediments), yet result in enhanced carbon loss to the atmosphere (Fig. 5b). In fact, the associated outgassing leads to a net decrease in the global ocean carbon inventory (–0.16 PgC yr-1), which exceeds, in absolute terms, the additional CaCO3 burial (0.10 PgC yr-1; see Fig. B4b)."*

Line 353-55: This central idea should appear much earlier in the ms.
We believe that this central idea is an integral part of the results and therefore rightfully belongs in this section. Introducing it earlier would be premature, even if it is implicitly addressed in the 'Methods' section through the presentation of the theoretical framework.

Lines 372-275: These are important statements which are not properly reflected in the abstract (see main comment 1 above).
We agree that this issue deserves to be explicitly raised in the GCB framework. However, with the new structure of the manuscript, we believe that this point now finds its appropriate place in the 'Discussion' section of the first proof-of-concept application we develop (Sect. 4.1). This issue is also mentioned in the new version of our 'Conclusion and perspectives'.

Lines 377-380: These statements are true but potentially misleading as they give the impression that the Alk imbalance will always lead to a reduction in the ocean outgassing. In fact, the opposite effect could also be obtained, if the burial terms are kept as they are, but if the Alk riverine input (see major comment 1 above) is decreased. Then the Alk imbalance (negative) should generate an additional outgassing compared to a budget based on an equilibrated Alk budget. I would reflect on this more transparently.
Yes, as mentioned in a previous comment, we agree that an imbalanced Alk budget could lead to either an increase or a decrease in the pre-industrial outgassing. This point has now been clarified in the revised manuscript. However, we believe it is important to emphasize that, even though it may appear surprising, adjusting either the carbon-side or the Alk-side of the $CaCO_3$ burial flux (based on Middelburg et al. (2020) and Regnier et al. (2022) respectively), leads in both cases to a further reduction of the pre-industrial outgassing by 0.11 and 0.18 PgC yr$^{-1}$, according to our theoretical framework. This highlights the importance of addressing the issue of a combined carbon and Alk budget when reassessing the pre-industrial air-sea carbon flux. A dedicated and reorganized 'Discussion' section has been added to the manuscript to specifically address this reevaluation.

Line 400: Typo Table YY. Should it not be Fig 6a ?
This has been corrected.

Line 422: "tend to manifest locally, primarily in the northern hemisphere" Why, especially for the rivinorg run?
In the standard configuration of NEMO-PISCES – identical to the one used in our study – riverine discharge is assumed to be fully labile, thereby directly impacting both Alk and DIC at the river mouths. Consequently, modifying the partitioning between organic and inorganic components of the riverine input (rivorg and rivinorg) only affects the Alk riverine input (see Fig. B4b). The resulting changes in Alk occur locally, at the river mouths, and therefore tend to affect the air-sea carbon flux locally (i.e. mainly with a regional effect). It is worth noting that this change in the Alk budget (in rivorg and rivinorg versus std) induces a compensation through CaCO3 burial (see Fig. B4b). However, as this compensation is distributed over the global ocean, it is expected to have only a negligible influence on the spatial distribution of the air-sea carbon flux.
        The simulation that significantly alters the spatial distribution of the riverine/burial-driven air-sea carbon flux by modifying the riverine discharge is rivref, in which the riverine organic discharge is assumed to be fully refractory (see Sect. 3.3.2).

Line 430-431: would "local" or "within hemisphere" help clarify the statement ? That is, …."with a carbon:Alk imbalance stemming from within-hemisphere (or local) riverine and burial fluxes"
This whole complicated paragraph was pushed in 'Appendix', in addition to the associated subpanel of the previous Fig. 6.

**Reviewer #2**

In their manuscript, Planchat et al. investigate the pre-industrial outgassing in the ocean, with a particular focus on constraints from processes that affect the ocean alkalinity budget. They thereby combine a theoretical model, which budgets alkalinity from riverine source and burial sink and also accounts for a potential drift, model simulations, including a great number of sensitivity simulations, as well as a practical framework which aims to reconcile the sensitivity simulations with current river input and burial estimates. The authors have obviously put a great amount of thought and effort into the development of these tools and simulations, and the work generally seems seems and robust from a technical stand point. However, the limitations of the model and framework (and I assume there are many), are not transparently discussed. The paper is also extremely hard to follow in its current presentation, and would not be well suited for general BG readers (and arguably even experts on the topic), in its current form. I therefore recommend major revisions, which however mostly involve streamlining and improvements in the explanations (especially regarding methodology). I would also recommend potentially re-thinking the amount of frameworks and results the authors wish to present. I made some suggestions and comments, which I hope can help improve the paper.

**Main Comments:**

- The paper and especially individual sections need a better structure and streamlining. The paper is covering a large amount of novel material: a new-ish theoretical and a new practical framework, almost uncountable amounts of sensitivity experiments, GCB results, CMIP7 results, a literature review, and then new theories in addition popping up in the results section.It would be important that the methods and results are presented in a way that the general reader can somewhat follow and understand. For individual sections, it's often not really clear what the main message of the section is (often multiple complex and only sparsely explained points, also often unconnected results). Sensitivity experiments are introduced in the results section without having been properly explained. I also believe certain aspects could be simplified (maybe inspired at how previous studies explain the alkalinity/carbon ocean budget, interhemispheric transport), but would leave it up to the authors to decide.

As mentioned for Reviewer #1, we fully acknowledge the difficulty we initially encountered in conveying the results of this study in a clear and accessible manner. We agree with your assessment that the submitted version of the manuscript was, at times, difficult to follow and, in places, challenging to understand.

In response, we undertook a substantial restructuring of the manuscript, accompanied by an in-depth rewriting effort. This revision has greatly improved both the clarity and accessibility of the paper. While we recognize that the topic remains inherently complex, we now feel confident – thanks in large part to your constructive feedback – that the core ideas and associated messages of the study are now clearly articulated and accessible to the broad readership of *Biogeosciences*.

- I think it would be very helpful for the general reader if the paper were to introduce the "geologically-known" balance of carbonate alkalinity from river inputs to burial in the ocean (see e.g. Hartmann et al., 2013), I guess early in the methods section:
With the left-hand side the river transports, and the right the exports from the ocean. This is essential to understand and might make it easier to follow the equations described in section 2.1. I think this is in essence what is meant throughout the paper when referring to a "simple budget" in the results section, although it wasn't clear how this differs to the theoretical framework explained in the methods. The equations (atleast to me) explain in a simpler way (atleast for me) the difference of balance of alkalinity:carbon from the river inputs and burial, and also explains why e.g. $CO_2$ outgassing increases with $CaCO_3$ burial (to me these explanations in the results section will be hard to follow for a general reader).

The approach you suggest, referring for instance to Hartmann et al. (2013), is indeed interesting, but we believe it lies beyond the scope of this manuscript, which is already dense and conceptually rich.

Our intention was to focus on communicating the complexity of the riverine/burial-driven air–sea carbon flux, in continuity with Hauck et al. (2020), without reducing this paleo-alkalinity balance to a simplified relationship. Such a simplification would mask key degrees of freedom, particularly the partial independence between riverine inputs and burial fluxes.

As for the specific aspect of carbonate compensation, which can be counterintuitive even to many within the marine biogeochemistry community, we have clarified its role in the revised manuscript wherever relevant, while ensuring it does not create confusion.

- The methods section is lacking a general overview on the strategy, which I think is needed due to the numerous frameworks/simulations and complexity of the topic. I would clearly explain from the start why there is the need to develop a a theoretical framework, then run simulations with sensitivity experiments, then another practical framework (and to which purposes the different frameworks are needed)? Why not just improve the model to perform better?

The 'Methods' section has been thoroughly restructured. It has also been simplified and clarified, resulting in a more streamlined manuscript. While the content remains dense, the revised structure should ensure that it no longer overwhelms the reader.

- In the methodology and even after reading the whole text, it is not really clear what the exact purposes of the individual sensitivity experiments needed. I understand they are needed for the practical framework, but why these specifically ones, would the results be different actually if you had chosen other sensitivity experiments / left out some? The results from the sensitivity experiments are brought up every now and then in the paper, but the general thinking behind these experiments, and clear explanations and reasonings behind the simulations would help a lot from the methods section.

We have improved the description of the sensitivity simulations, which are also thoroughly described in the 'Appendix' to avoid overloading the main body of the manuscript. We believe that the major restructuring of the paper now provides clearer context for understanding the purpose of these simulations, which span a wide range of hypotheses regarding riverine and burial fluxes. Specifically, these sensitivity simulations are used in the 'Results' section to validate the theoretical framework and to identify the main factors influencing the regional distribution of the riverine/burial-driven air–sea carbon flux. Subsequently, in the 'Proof-of-concept applications and discussions' section, they serve as the basis for constructing a composite simulation – built from a linear combination of selected sensitivity simulations – which enables a reassessment of the regional distribution of the riverine/burial-driven air–sea carbon flux and demonstrates its practical implementation.

- Regarding the strong focus of the paper on Global Carbon Budget numbers and history (and a call for potential revision?): I am not sure improving the observations to model discrepency of the Global Carbon Budget should really be a central evaluation metric for the results. There are now a few other major issues that have been identified that could explain (a large part?) of the bias between observational and model products (model biases: Terhaar et al., 2022 observation biases: Landschützer et al. 2023). In the abstract it seems like the authors are strongly calling for a revision of GCB estimates based on their numbers, but in the main text and especially in the conclusion, the authors don't seem quite as confident in these mean numbers, highlighting large uncertainties. Personally, the study does an excellent job at highlighting uncertainties in current knowledge and estimates, but the limitations in terms of the model used and the observational basis for burial/rivers are the same/similar as of previous work. I think would probably make more sense to try to motivate for better models and observations for reducing these uncertainties.

With the revised structure of the manuscript, substantial changes have been made that directly address your comment.

We have maintained a strong focus on the GCB, both in the 'Introduction' and in the discussion subsections of 'Proof-of-concept applications and discussions', in relation to the proof-of-concept applications we develop (i.e. our reassessments of the riverine/burial-driven air–sea carbon flux, in terms of both its global magnitude and regional distribution). The GCB remains the most comprehensive and relevant synthesis report to frame this issue, and it was also the original motivation behind the development of our proof-of-concept applications. These applications, by design, are flexible and reproducible, and we see them as opening new avenues for future iterations of the GCB.

We no longer report specific reassessment values in the 'Abstract' or the 'Conclusion and perspectives' sections. In these key parts of the paper, we now emphasize the strength and flexibility of the theoretical framework and its applications, rather than the specific results of our reassessments. Our primary objective is to encourage the community to adopt and build upon the methodological approach we propose, rather than to focus on the quantitative outcomes of our illustrative applications.

Finally, in the 'Conclusion and perspectives' section, we now explicitly highlight the key challenges and priorities for the community concerning the riverine/burial-driven air–sea carbon flux:

*"These flexible applications now call for four key efforts from the community regarding the pre-industrial riverine/burial-driven air–sea carbon flux:*

*(i) To reduce uncertainty in its global magnitude:*

*- Clarify the definition of ocean domain boundaries at the coastal interface within the land-to-ocean continuum, where multiple fluxes intersect (riverine discharge, and part of organic matter and CaCO3 burial).*

*- Establish a combined and internally consistent carbon and alkalinity budget, as current independently developed estimates remain inconsistent (e.g. CaCO3 burial).*

*(ii) To reduce uncertainty in its regional distribution:*

*- Improve our understanding of the intrinsic properties of riverine and burial fluxes (e.g. the fate of terrestrial organic matter).*

*- Promote intermodel comparison efforts to identify systematic biases and improve robustness across modeling approaches."*

- Limitations: The limitations from the both simulations and practical should be made more transparent. For instance, the model likely very poorly represents the processes in the coastal ocean (CaCO3 production there?) and its transport to the open ocean, doesn't represent terrestrial organic matter etc. . I think the practical framework doesn't distinguish between more realistic sensitivity simulation assumptions, but only aims to reduce the bias to the observations, for me this seems kind of an important caveat (so might be getting things "right-er" for wrong reasons).

In the submitted version of the manuscript, the applications of our theoretical framework were not sufficiently developed and led to confusion. The revised manuscript structure now clearly separates our two proof-of-concept applications (Sect. 4.1 and 4.2), without conflating them with the core 'Results' section of the paper.

In both applications, we followed the strategy most consistent with the current state of knowledge regarding riverine and burial fluxes, as well as the present capabilities of NEMO-PISCES in representing these processes. In addition, we now explicitly discuss the limitations and potential improvements associated with each application.

For instance, in the first application (Sect. 4.1), aimed at reassessing the global magnitude of the pre-industrial air–sea carbon flux, we write:

"The current inconsistencies between the independently developed carbon and Alk budgets make our estimate less robust and highlight the need for a combined revision of both budgets. Beyond the 0.10 PgC yr-1 reduction in outgassing due to differing ocean boundary definitions relative to GCB, the remaining 0.06 PgC yr-1 decrease in our new estimate is linked to a slight imbalance in the Alk budget (+0.07 PgC yr-1 Middelburg et al., 2020). However, the discrepancy in CaCO3 burial estimates between the most recent carbon and Alk budgets (Regnier et al., 2022; Middelburg et al., 2020) would translate into a 0.22 PgC yr-1 difference in the Alk budget (Table 3). If the carbon flux associated with CaCO3 burial were aligned with the Alk budget from Middelburg et al. (2020), the

outgassing would decrease by an additional 0.18 PgC yr-1. Conversely, aligning the Alk flux associated with CaCO3 burial with the carbon value from Regnier et al. (2022) would reduce the outgassing by 0.11 PgC yr-1. Thus, reconciling CaCO3 burial fluxes in both carbon and Alk budgets is expected to further reduce the current outgassing offset (Friedlingstein et al., 2024). Establishing a combined and internally consistent carbon and Alk budget is therefore essential to confidently reassess the pre-industrial outgassing within the theoretical framework presented here."

And in the second application (Sect. 4.2), focused on reassessing the spatial distribution of this flux, we write:

"A more comprehensive characterization of riverine and burial fluxes of carbon and Alk remains a critical challenge for accurately constraining the spatial distribution of the riverine/burial-driven air–sea carbon flux. This is particularly true for the fate of terrestrial organic carbon and its associated lability, which remains highly uncertain (Aumont et al., 2001; Jacobson et al., 2007; Gruber et al., 2009). Nevertheless, the approach proposed in this study is flexible and can accommodate future revisions of these external fluxes. Fundamentally, the selection of sensitivity simulations used to construct the composite simulation (Sect. 4.2.1; see also Fig. E2) can be revisited as scientific understanding progresses or as model representations evolve."

- Regarding the impacts of an alkalinity imbalance: The paper really lays a large focus on the potential impacts of an alkalinity imbalance, and largely motivates a major part of the frameworks introduced. But if I understand correctly, this imbalance is only a possibility and its even uncertain in which direction this imbalance would be?

Absolutely, there remains significant uncertainty regarding the Alk budget, including even the direction of the imbalance, as rightly noted. From a paleoclimatic perspective, one might expect a negative Alk imbalance. However, the most recent assessment that we adopt in this study (Middelburg et al., 2020) suggests a positive imbalance. It is therefore likely that future revisions of the Alk budget will emerge. This is precisely where the theoretical framework we introduce and validate in this manuscript proves valuable: the two proof-of-concept applications we develop based on this framework are inherently flexible and can be readily updated to accommodate such revisions in the Alk budget.

- Regarding the spatial variability of the preindustrial outgassing: The authors acknowledge that the fate of terrestrial organic matter plays a major role in explaining spatial patterns of river-induced outgassing in the ocean. However, since this paper doesn't really investigate constraints or uncertainties regarding the fate of terrestrial organic carbon in the ocean I wouldn't really be particularly confident in the spatial distribution derived from the framework presented here. While this seems to reduce biases in the Southern Ocean between model and observational means, I would argue this is a region that is both one of lacking observational data, and complex region for models to simulate. Recent estimates of OM lability generally seem to suggest rapid degradation, even already in the coastal ocean, which would go against significant transport to and outgassing in the Southern Ocean (Aarnos et al., 2019; Lacroix et al., 2021; Powley et al., 2024).

While the fate of terrestrial organic matter is not the central focus of the manuscript, the issue is explicitly addressed. Among the various hypotheses we tested regarding riverine discharge, one specifically concerns the lability of terrestrial OM. In the standard configuration (std) of NEMO-PISCES – which was used in major intercomparison exercises such as CMIP6 and the GCB – riverine organic discharge is treated as fully labile, directly affecting both DIC and Alk at the river mouths. We explored the opposite assumption by running a simulation where the entire riverine organic discharge was considered refractory, thereby homogeneously distributing its effect for DIC and Alk across the ocean interior.

Moreover, as you rightly point out, recent studies tend to support a predominantly labile nature for terrestrial OM. In light of this emerging consensus, we ensured that the composite simulation (Sect. 4.2) – used to reassess the spatial distribution of the riverine/burial-driven air–sea carbon flux – was built exclusively from sensitivity simulations assuming labile organic discharge.

*Once again, our second application strategy illustrates the core strength of our approach: its flexibility. The method allows for rapid and effective updates in response to evolving understanding or revised estimates of riverine and burial processes, without the need to rerun model simulations.*

**Specific Comments:**

*Abstract*

L1 "The disparities in estimates of the ocean carbon sink, whether derived from observations or models..." Did the authors not mean: "The disparities in the estimates of the ocean carbon sink between observations and models ..
"raise questions about our ability to accurately assess its magnitude and trend over recent decades. " For me this is too strong of a statement.
*The first few sentences of the abstract have been adjusted to ensure a more balanced and accurate language:*
*"Disparities between observational and model-based estimates of the ocean carbon sink persist, highlighting the need for improved understanding and methodologies to reconcile differences in both magnitude and trends over recent decades. A potential key source of uncertainty lies in the pre-industrial air–sea carbon flux, which is essential for isolating the anthropogenic component from observations. This flux, thought to result globally from an imbalance between riverine discharge and sediment burial of carbon, remains highly uncertain, limiting the confidence in impactful applications such as the Global Carbon Budget (GCB)."*

L17 "Addressing the current inconsistencies between the combined carbon and alkalinity budgets.." What these inconsistencies are hasn't been mentioned yet, so I would tend to leave this out here (or briefly explain what is meant beforehand).
L19 "... and intermodel comparisons are required to constrain its regional distribution." I would argue that improving the models regarding the fate of terrestrial carbon in the ocean would also be needed (maybe even as a more urgent priority).
*The second part of the abstract was also fully rewritten to align with the updated structure of the manuscript, clearly distinguishing the results from the proof-of-concept applications we share:*
*"In this study, we present a new theoretical framework that enables direct estimation of the riverine/burial-driven pre-industrial carbon outgassing using both carbon and alkalinity budgets. This approach is validated with a series of ocean biogeochemical simulations, which also highlight the main factors influencing its regional distribution. We then demonstrate the utility of the framework through two proof-of-concept applications. The first revisits the pre-industrial riverine/burial-driven air–sea carbon flux using existing carbon and alkalinity budgets, offering a simple method for reassessment as these budgets are updated. The second application leverages sensitivity simulations to construct a composite simulated estimate that aligns with both carbon and alkalinity budgets to assess the regional distribution of the pre-industrial riverine/burial-driven air–sea carbon flux. This approach is well suited for model intercomparisons, enabling efficient reassessment of regional flux patterns and helping to reduce biases related to ocean model physics or biogeochemical parameterizations."*
*We emphasize, in particular, that one of the key advantages of intermodel comparisons lies in the ability to account for the diversity in representation of ocean physics and key biogeochemical processes (e.g. the fate of terrestrial carbon). This diversity helps highlight the variability these choices induce, and may ultimately support the informed selection of model simulations that align best with current scientific understanding.*

*Introduction*

L21-L26 As mentioned, I'm not sure I would make the discrepency between observations and model estimates the central motivation here, since many other factors have been put forward recently that may also explain the difference.

It is true that other studies have attempted to explain this discrepancy, but it remains likely that it results from a combination of factors – which is precisely where our study becomes particularly relevant. A full paragraph was added in the 'Introduction' to address this point:

*"Recent studies have begun to investigate the origins of discrepancies in both the magnitude and trend of observational versus model-based estimates of the ocean anthropogenic carbon sink. Analyses of GOBM-derived estimates (Terhaar et al., 2024) and pCO2-based products (Ford et al., 2024) point to multiple sources of uncertainty, including methodological differences, model biases, and data limitations. On the modeling side, GOBMs have been shown to underestimate the global sink magnitude (Terhaar et al., 2022), as well as decadal variability, especially in the Southern Ocean (Mayot et al., 2023, 2024). On the observational side, the sparse and uneven spatial and temporal coverage of surface ocean pCO2 measurements remains a major limitation (Hauck et al., 2023; Landschützer et al., 2023; Dong et al., 2024). While the causes of these mismatches are likely multifaceted, one less-discussed contributor is the uncertainty surrounding the pre-industrial air–sea carbon flux and its influence on pCO2-based estimates."*

L40 "The spatial distribution of this riverine/burial-driven air-sea carbon flux is also highly uncertain and depends on the assumptions and methods used for its assessment (see Table B2). " I would expand on which important assumptions are referred to here.
L41 "The historical value.."
This term is kind of confusing here, historical=pre-industrial ?
We briefly expanded on the key underlying assumptions and refined the wording throughout the paragraph for clarity:

*"The spatial distribution of this riverine/burial-driven air-sea carbon flux is also highly uncertain. It strongly depends on the assumptions and methods used to assess it, including how sediment burial processes are represented, and both the magnitude and characteristics of riverine carbon inputs – particularly the balance between organic and inorganic forms, as well as the lability of terrestrial organic matter – (see Table E1). The earlier estimate, derived from a modeling analysis (Aumont et al., 2001) distributed this flux as follows: 49 % in the southern region, 25 % in the intertropical region, and 26 % in the northern region. In contrast, the most recent estimate, currently used in the GCB and also based on a modeling study (Lacroix et al., 2020, Table 1), suggests a very different partitioning: 14 %, 64 % and 22 %, respectively, reshaping our understanding of the regional distribution of this flux."*

L48 "These disparities have emerged and disappeared without apparent reason.."
L49-L54 I believe these strategic changes were made using the best and most recent science (and not with the primary goal of reducing the discrepency). I don't think this qualifies as "without apparent reason".
We agree that the original wording was inappropriate and have accordingly revised the paragraph:

*"Uncertainties in estimating the riverine/burial-driven pre-industrial outgassing may contribute to the persistent discrepancies between observation-based and model-derived estimates of the anthropogenic ocean carbon sink, both globally and regionally (Friedlingstein et al., 2024, their Fig. 11 and 14). These disparities have fluctuated over time, largely in response to stepwise adjustments made by the GCB team following new reassessments of the magnitude and spatial distribution of this flux in the literature (Fig. 1, Table 1). For instance, a substantial decrease in the global estimate of the pre-industrial outgassing from 2019 to 2020 contributed to a notable narrowing of the global observation–model gap. More recently, from 2022 to 2023, a redistribution of the flux between regions, from the southern region to the tropics, led to a reduced southern hemisphere bias and a compensating increase in the inter-tropical discrepancy."*

L57-L60 I would maybe add information whether the CMIP6 models even consider riverine fluxes. I would guess not, and in a probably very simplified manner.
While it is true that not all CMIP6 ESMs include riverine carbon inputs, most do, albeit with varying assumptions (see Planchat et al., 2023, our Supplementary Table). However, to keep the manuscript focused and avoid unnecessary complexity, we prefer not to elaborate on this point in the main text,

as the use of CMIP6 ESMs in our study is essentially illustrative. A simple comment has been added here: *"likely due to differences in model setups and various/incomplete representations of sediment burial and riverine discharge (Terhaar et al., 2024)"*.

*Methods*

2.1 -> I guess this section describes the theoretical framework mentioned previously? I would make this more clear (not really sure overview and definitions is really a fitting subtitle).
The 'Methods' section was restructured and simplified. In particular, 'Theoretical framework' is now a stand alone section to increase clarity.

L140 "this conceptual tool only permits estimates of a potential air-sea carbon flux resulting from a local/regional carbon:Alk imbalance" Please clarify what is meant by "potential air-sea carbon flux".
L142-143 "There is indeed no guarantee of its local/regional applicability due to the transport of the induced carbon:Alk local/regional imbalance. " Could you maybe explain its applicability then, or how exactly you are dealing with this uncertainty?
The concept of a "potential" air-sea carbon flux is complex to define and understand. We have simplified the way we introduce it, while clarifying it in Sect. 2.1.2:
*"To gain a deeper understanding of the factors shaping the spatial distribution of the riverine/burial-driven air-sea carbon flux, we expand upon the theoretical framework previously outlined for the global scale (refer to Section 2.1.1) and adapt it as a proxy for application at regional scales. This is essential to understanding the extent to which specific regional carbon:Alk budget imbalances can drive the global air-sea carbon flux as well as deviations in carbon and Alk inventories.*
*The air-sea carbon flux calculated by applying Eq. 10 to the riverine and burial fluxes of a specific region can only be considered a potential air–sea carbon flux, i.e. a capacity to generate such a flux at the global scale, without any guarantee that it fully occurs within the same region. Due to ocean circulation and the associated transport of carbon and Alk, regional carbon:Alk budget imbalances in riverine and burial fluxes explain the regional distribution of the drivers of the global air–sea carbon flux. However, they only partially explain the regional distribution of the flux itself."*

L208 "The inorganic fraction is supposed to be in the form of bicarbonate ions and thus affects both DIC and Alk in a similar manner. " Is it supposed to be or is it added as such?
This was indeed imprecise and has been revised accordingly: it is added as such.

L210 This simulation also includes the burial of OM and CaCO 3 produced by pelagic organisms, which is exported to the ocean interior and only partially remineralized or dissolved in the water column and at the seafloor. Since this paper focusses on alkalinity fluxes and its burial, I would recommend giving a brief overview of how these processes affect the alkalinity balance (and especially how the burial is parametrized).
We believe it is unnecessary to revisit these foundational aspects within the manuscript, which is already quite dense. We have instead added a reference to Planchat et al. (2023), which discusses the soft-tissue and carbonate pumps in considerable detail, particularly through the lens of Alk.

L231-244 The sensitivity experiments are described here, but it is not really clear why they are (individually) needed.
Without adding excessive detail, we took care to briefly highlight the relevance of the different types of simulations in this paragraph. The hypotheses underlying these simulations are also clearly presented and discussed in the 'Results' section.

L248 "For instance, the gap in the CaCO 3 burial ( 0.24 versus  ) would drive a  difference in the Alk budget. " Isn't this a difference in the carbon budget?
This was moved to the discussion of the first application section (Sect. 4.1.3). See also the reply to your major comment starting with "- Limitations:". We confirm that our sentence was correct. The carbon flux associated with CaCO3 burial was estimated at 0.24 PgC yr-1 by Regnier et al. (2022)

and 0.35 PgC yr-1 by Middelburg et al. (2020). This corresponds to a difference in the alkalinity flux of 0.22 PgC yr-1 (0.11x2).

Figure 4: "The negative impact of OM burial on Alk is attributed to the release of ammonium when OM is remineralized at the seafloor rather than buried." I am not sure this information belongs in the caption, but in the text.
In theory, yes, but this information plays only a very minor role in the study and manuscript. We believe that a brief clarification in the figure caption is sufficient. Otherwise, the information is likely to be missed during reading and could cause confusion.

*Results and discussion*

L302 "This residual component is attributed to a residual carbon budget imbalance due to internal ocean processes." I guess this would be a model drift?
This is not a model drift and a detailed explanation is provided in Appendix B2.4.

L309 "Role of sediment burial fluxes" This section was really tough to read and understand, it has a lot of unconnected (complex and not really explained) information put together, it is really unclear in terms of streamlining -> maybe try to revise?
The wording of the entire section has been improved for clarity.

L310 "At global scale, the riverine carbon flux in the standard simulation  is insufficient to fully account for the air-sea carbon outgassing (  ) alone (Fig. 4d and 5a)." I think most readers will not understand what is meant here (actually I am also not sure), since 0.52 PgC yr-1 is larger than .
This has been adjusted: *"Accounting for the riverine carbon input alone in the standard simulation (+0.52 PgC yr–1) is insufficient to explain the simulated air-sea carbon outgassing (0.27 PgC yr–1). Burial-associated carbon fluxes, from both OM (–0.17 PgC yr–1) and CaCO3 (–0.04 PgC yr–1; Fig. 2d and see Fig. B4a), act to partially offset this input, thereby reducing the net outgassing to 0.31 PgC yr–1."*

L315 'Increasing the river input by a factor of 1.5 (riv1p5)"
Increasing all species of carbon? Wasn't clear to me from the methods, and would be quite important to know to understand the increase in carbon outgassing?
This is now mentioned in the 'Methods' (Sect. 2.2.3), that we increased riverine input by a factor of 1.5, while maintaining the same partitioning as in the standard configuration (std).

L324 "This underscores the significance of considering CaCO 3 burial when constructing the ocean carbon budget, especially when evaluating the net air-sea carbon flux. "Yes but I would argue understanding the fate of terrestrial OM and its burial is equally important, in terms of magnitudes atleast?
The final sentence of the paragraph has been modified to better reflect the underlying concept: *"This highlights the pivotal role of CaCO3 burial in shaping the air-sea carbon flux under the assumption of a balanced Alk budget, where riverine Alk inputs are offset by CaCO3 burial (Fig. 4d)."*
        As a reply to the second part of your comment, no, the fate of terrestrial OM in the ocean does not affect the global magnitude of the air-sea carbon flux. This is exactly what we demonstrate with the theoretical framework we introduce in this manuscript  and validate through our simulations. However, it does influence the spatial distribution of the riverine/burial-driven air-sea carbon flux, and this is addressed in Sect. 3.3.

L325 "The total carbon outgassing obtained from a simple carbon budget only holds however for our simulations in which the global Alk inventory is in equilibrium" I think you should explain which of these fluxes above are the simple carbon budget? To me they seem coming from the simulation.
The small paragraph has been adjusted: *"However, such a carbon budget – which deduces the pre-industrial air-sea carbon flux from riverine and burial fluxes of carbon – is only valid under the condition of a balanced Alk budget (Fig. 5b). When this assumption does not hold, it becomes*

*necessary to account for both the carbon and Alk budgets to correctly assess the riverine/burial-driven air-sea carbon flux (Sect. 2.1.1)."*

L334 Why are the regional burial fluxes actually discussed in the "imbalanced" alkalinity section? Is it not possible that these large regional burial fluxes are actually in quasiequilibrium with the river fluxes? If you are removing alkalinity in these places artificially, would you not have less alkalinity burial in other places as a result in the model?

The rationale behind designing simulations where we manipulate CaCO3 burial stems from the difficulty of accurately representing this flux in ESMs. It is generally not a priority for modeling groups, which tend to focus more on surface processes that respond more directly and rapidly to climate change. However, from a paleoclimatic perspective, this flux – although also challenging to constrain with data-based products – is of real interest. The aim of the three specific configurations we developed ('atlpac', 'atlpac-diseq', and 'tropics-diseq') is precisely to control changes in $CaCO_3$ burial both in amplitude and spatial distribution, while keeping perturbations to other biogeochemical processes negligible. This allows us to explicitly control the Alk budget: it is balanced only in 'atlpac', and deliberately imbalanced in 'atlpac-diseq' and 'tropics-diseq' (see Table B1). The whole section was rewritten to enhance clarity.

L373 "Part of this discrepancy arises because atmospheric carbon uptake by continental shelves Regnier et al., 2022) is fully integrated into our net pre- industrial
riverine/burial-driven air-sea carbon flux as we also consider OM and CaCO 3 burial in these regions, reducing this flux by ." The OM and CaCO3 burial on continental shelves will likely not be well resolved at the applied model resolution (+ not taking into account coastal specific processes such as benthic CO3 producers etc. ).

Observations are expected to account for the coastal ocean. Therefore, any product used to derive the oceanic carbon sink from observations (e.g. the riverine/burial-driven air-sea carbon flux, whether inferred from our theoretical framework or from modeling) should logically include the coastal component as well. Otherwise, this introduces an inconsistency.

Moreover, we acknowledge the limitations of global models in representing the coastal ocean, but the fact that coastal processes are imperfectly represented does not mean they are entirely absent. For instance, in NEMO-PISCES, burial fluxes are predominantly concentrated in the coastal domain. We argue that imperfect representation should not justify omitting these processes altogether.

We also emphasize that the definition of ocean domain boundaries at the coastal interface, within the land-to-ocean continuum where multiple fluxes intersect (e.g. riverine discharge, and part of organic matter and CaCO3 burial), should be explicitly addressed by the GCB). This point is specifically mentioned in the 'Conclusion and perspectives' section.

L426-L440 This section was really heavy and hard to understand.

This complicated paragraph was moved to an 'Appendix', in addition to the associated subpanel of the previous Fig. 6.

*Conclusion and perspectives*

L485 "Additionally, a better understanding of the intrinsic properties of these external fluxes is needed, such as whether the organic carbon brought by rivers is refractory or not (Aumont et al., 2001; Jacobson et al., 2007; Gruber et al., 2009)." Improving the understanding on the fate of the terrestrial organic carbon is probably equally important. I don't think its impacts are resolved by degrading the organic matter at the river mouths as in such models.

The 'Conclusion and perspectives' section has been entirely rewritten to reflect the new structure of the manuscript. Once again, we highlight the fate of terrestrial organic matter as one of the key challenges facing the community. However, we argue that the likely resolution to this issue lies between two end-member hypotheses: one assuming that the riverine organic discharge is entirely labile (as in the std simulation), and the other assuming it is entirely refractory (rivref). Both extremes

are explored in this study and could, in the future, be integrated into the construction of the composite simulation – used to reassess the spatial distribution of the riverine/burial-driven air–sea carbon flux – by assigning appropriate weights to each sensitivity scenario. This is precisely the kind of flexibility offered by the second proof-of-concept application developed in this work

---

## Author Response (AR3)

**Manuscript revision (2nd round)**

A fresh look at the pre-industrial air-sea carbon flux using the alkalinity budget Alban Planchat, Laurent Bopp, and Lester Kwiatkowski

We have carefully addressed the reviewers' comments in our revision and would like, once again, to express our gratitude to both reviewers for their careful and constructive evaluations. Over the course of these two rounds of review, their insights have greatly contributed to improving the quality and clarity of our work. Below, we provide a detailed point-by-point response to all reviewers' comments.

In addition, we have updated Fig. 4b, Fig. B2 and Fig. B3c using a colour palette from <a href="https://sronpersonalpages.nl/~pault/">https://sronpersonalpages.nl/~pault/</a>, to ensure that readers with colour vision deficiencies can correctly interpret our findings.

**Reviewer #1**

The authors have made a remarkable job revising their ms. (I enjoyed reading this new version!) and at this stage, I only have a few very minor comment (technical corrections):

- Table 1: If I recall well, the underlying assumption in Jacobson et al. (2007) was that the distribution of outgassing mirrored the regional inputs/outputs (no inter-hemispheric transport). Thus, why 'not applicable' ? Please check.

You are absolutely right. However, this Table is simply a summary regarding the GCB manuscripts. At the time (from GCB 2013 to GCB 2017), the distribution of the air-sea carbon outgassing from riverine/burial carbon and alkalinity fluxes was not used. In fact, there was no regional distribution of the ocean anthropogenic carbon flux shown back then. For clarity, we have changed 'Not applicable' to 'Not considered at that time' in the Table.

- Line 67: Fig1a should be 2a (please check references to figures throughout the text)
  This has been corrected. References for the Figures and Tables have been checked throughout the manuscript.
- Figure 3 (caption): "and the colors of the arrows is consistent with what is used throughout the manuscript (e.g. Fig. 2)". Not sure to what this statement refers to.

  Remove?

This has been removed.

- Figure 3 (caption): "deviations" with respect to what?

We have added two sentences in the first paragraph of the Methods to make it as clear as possible:

"We refer to a 'deviation' in a given variable when the system is in steady-state, but a persistent trend is identified for that variable (e.g. the global carbon inventory). In contrast, we use the term 'drift' when a trend in a variable disrupts the steady-state."

- Line 122: add "ocean" to "a non-conserved global Alk inventory" This has been added.
- Line 186: explain in a few words (for the broad readership) what an "Alk restoring scheme" means

This has been added: "periodically restoring the global Alk inventory to a reference value by adding/removing the required amount, either uniformly or in a weighted manner"

- Line 200: Here (or somewhere else in 2.2.2 or 2.2.3 if you find it more appropriate), I suggest adding a reference to Liu et al. (Nature Geoscience, 2024) to highlight that your river input in your std simulation is on the low side.

Such a reference has been included in Appendix B2.1: "It is worth noting that the global values and latitudinal distribution of riverine inputs are based on (Ludwig et al., 1996) and have recently been revised (Li et al., 2017; Liu et al., 2024) although the human imprint on these fluxes cannot be removed."

- Lines 237&238: add "ocean" next to inventories? This has been added.
- Line 271-72: Should these numbers not be reported with a negative sign? Add that the difference between 0.27 and 0.31 is due to the slight "residual component"?

Since we are referring here to a net global outgassing (i.e. 'a net global outgassing of ...'), we report the value without the negative sign. The positive or negative sign is used only when the values are presented in brackets (see previous paragraph in the manuscript).

Regarding the clarification of this residual component offset, we have decided not to include the reference at this point in the manuscript, as it is already explicitly mentioned at the end of the previous paragraph in the manuscript. Moreover, it is too early in the Results section to introduce the expected value of  $0.27 \, \text{PgC/yr.}$

- Line 277: Should be Fig.4d (not 2d) This has been corrected.
- Section 3.2.3: Add a reference somewhere to fig.5c ? This has been added.

**Reviewer #2**

In their revised manuscript, Planchat et al. strongly improved their manuscript, especially in their restructuring and improving the contexts and explanations of their results, and their work remains significant and a huge effort. Both introduced frameworks are much clearer, and I especially appreciate that the second theoretical framework was introduced into a new section, which helps differentiate the results from the simulations obtained directly from the model, to those derived from making offline corrections to the simulations through combination of sensitivity experiments. The introduction of the second framework is also mostly clear through the step-wise description of the approach. I would still have a few points to clarify, but would otherwise recommend the manuscript for publication.

**Major Comments**

- Limitations: The manuscript would, especially regarding the theoretical framework, benefit from a brief limitations paragraph. While the methods introduced are definitely interesting and computationally efficient, they are not correcting for all physical and biogeochemical biases of the model (I don't think it is really what the authors mean, but it is not really stated otherwise). While this will likely work for corrections of model river inputs and potentially burial fluxes, structural model errors present in all simulations, which affect the redistribution of the carbon in the ocean, will not be corrected from combining the sensitivity simulations together to fit (globally evaluated?) observational values.

Regarding the global pre-industrial riverine/burial-driven air-sea carbon flux, it can now be assessed without relying on models, based solely on theoretical considerations (see Sect. 4.1). The physical and biogeochemical biases of models have no impact on these results. What remains limiting is our knowledge of, and confidence in, the values of riverine and burial fluxes of carbon and Alk (see Sect. 4.1.3).

As for the spatial distribution of the flux (Sect. 4.2), we explicitly discuss the limitations associated with model use, particularly in the final paragraph of Sect. 4.2.3 (also in Conclusion and perspectives). It is true that the approach (Sect. 4.2.1) does not resolve the potential impact of

biases in model physics or biogeochemistry on the spatial distribution of the flux. These biases are inherent to the use of a single model. However, if we aim to mitigate at least part of these model-intrinsic biases, an intercomparison exercise is needed, and our approach (Sect. 4.2.1), being highly adaptable, is particularly well suited for that purpose.

- Importance of terrestrial organic carbon discussion: The (equal?) importance of terrestrial organic carbon inputs for riverine-induced outgassing, which is really only very coarsely taken into consideration here, is still not clearly enough stated. Estimating the terrestrial organic contribution by correcting the simulations with burial observations as presented here would, in my opinion, not be robust as the regional outgassing is not necessarily coupled to regional burial fluxes associated with terrestrial organic carbon inputs (as one could make the case for terrestrial alkalinity/DIC). It is fine to not address this in the manuscript, but limitations regarding especially spatial patterns need to be acknowledged clearly (I am aware there is was single sentence introduced in the discussion section on this).

We fully agree on the importance of terrestrial organic carbon and on the need to better understand, represent, and constrain it in the future. This is, in fact, a central theme throughout the manuscript. From the introduction, we emphasize the significance of riverine organic matter and its lability. In our set of sensitivity simulations, we included a panel of four experiments specifically designed to investigate and illustrate the consequences of changing the partitioning of the riverine input (rivorg and rivinorg), its lability (rivref), and the magnitude of the total riverine flux. The assumptions related to riverine discharge, particularly its organic component, are at the heart of the present study.

We detail the influence of these assumptions on the flux distribution in Sect. 3.3.2, and discuss it in the penultimate paragraph of Sect. 4.2.3. We further highlight the critical need to improve our understanding of the fate of terrestrial organic matter in the 'Conclusion and perspectives' section.

While we do not resolve the open questions surrounding riverine terrestrial organic matter, we do illustrate the range of its potential impacts through our sensitivity simulations. Furthermore, we propose a flexible framework (Sect. 4.2.1) for combining sensitivity simulations, which can readily accommodate future updates reflecting revised understanding of the fate of this terrestrial organic carbon flux (e.g. magnitude, lability).

- Clarifications regarding the theoretical framework: The framework is now much more clearly explained! Could you however clarify in the text if the corrections applied are based on global observations, or are you taking into account the broad region definitions addressed in the manuscript (North-Tropics-South)? If I understand correctly, it is the latter, but if the former, I am wondering if the corrections applied might actually be introducing regional biases due to biases in the river inputs and modelled spatial distributions of burial fluxes. I am also wondering to which degree the application of the different correction steps are overriding each other (if I understand correctly, it is the air-sea CO2 flux that is being corrected again from steps 2-4?). Some further clarifications could be helpful here.

We have revised the first paragraph of Sect. 4.2.1 to better clarify our approach:

"The set of sensitivity simulations conducted to validate our theoretical framework spans a wide range of assumptions regarding riverine and burial fluxes of carbon and Alk. This provides all the necessary tools to reassess the spatial distribution of the riverine/burial-driven air—sea carbon flux. As in our global estimate (Sect. 4.1), this reassessment strategy is grounded in the most recent global budgets of carbon and Alk. By logically selecting and weighting some of our sensitivity simulations, we construct a composite simulation whose riverine and burial fluxes match those reported in these global budgets (Table 3). In this way, the composite simulation also combines the associated air—sea carbon fluxes, both at the global scale and regionally. It is this regional aspect that enables a revised estimate of the spatial distribution of the pre-industrial riverine/burial-driven air—sea carbon flux."

Thank you also for the comment, which has invited us to discuss an additional limitation of our approach in Sect. 4.2.3, not mentioned previously:

"A limitation of our approach is that a substantial revision of the spatial distribution of a flux -such as riverine inputs -- would require rerunning a simulation, as it cannot be addressed through our current framework alone."

Finally, and as expected, the effects of the successive steps on the global air–sea carbon flux, and its regional distribution, can accumulate or partially offset each other as the steps progress (see Fig. E3).

**Specific Comments**

L13 enabling efficient reassessment -> an efficient reassessment? This has been modified accordingly.

L41 "literature estimates of riverine and burial fluxes" I would remove "literature" here, since some of these studies derive the river flux magnitudes themselves.

This has been modified accordingly.

L209 "Indeed, while at the global scale, the net air-sea carbon flux directly corresponds to the riverine/burial-driven air-sea carbon f lux (Eq. 2), at the regional scale (N, S, or I, Fig. 3b), the air-sea carbon flux (FC, air-sea nat. components: one associated with the functioning of the ocean carbon pumps (FC, air-sea pump riverine and burial fluxes (FC, air-sea riv./bur.), which is our primary focus:" Why not just call this the "natural air-sea carbon flux without/prior to river fluxes" or something like this? (I mean the carbon pump processes also affect riverine carbon?) I would also add "..to the theoretically-calculated riverine/burial-driven air-sea carbon flux?"

We agree that, at the regional scale, there is an interaction between the effect of the ocean carbon pumps and the riverine/burial carbon and Alk fluxes on the air-sea carbon flux. However, we chose to retain the terminology we originally used, as it helps to highlight the source of the driver: internal to the ocean for the carbon pumps, and at the system boundaries for the riverine and burial fluxes. We believe this distinction is both more relevant and more accessible for readers. That said, to take your comment into account, we have slightly refined some of the phrasing to be more accurate.

"Indeed, while at the global scale, the net air-sea carbon flux directly corresponds to the riverine/burial-driven air-sea carbon flux (Eq. 2), at the regional scale (N, S, or I, Fig. 3b), the air-sea carbon flux (FC,air-sea nat.) can be decomposed at first approximation into two components: one internal component linked to the functioning of the ocean carbon pumps (FC,air-sea pumps), and a boundary component associated with the riverine and burial fluxes (FC,air-sea riv./bur.) – our primary focus – :"

L222 Importantly, within the context of our study, the absolute values of the fluxes—whether they align with literature estimates or not—are not of primary concern. What matters are the relative differences between these values across simulations, which reflect the assumptions being tested. Could the authors maybe be precise here which assumptions are specifically being tested?

These are specified in the following paragraph, with further details provided in Appendix B2.1. We have clarified the end of this paragraph to ensure continuity for the reader.

"What matters are the relative differences between these values across simulations, which reflect the assumptions being tested and briefly outlined below (see Sect. B2.1 for further details)."

**Table 3 This Table should be explained in more detail in the text.**

We believe that this Table is already sufficiently explained in the main text. Going further would require detailing the numbers it contains (as in the first version of the manuscript), but this would be verbose and reduce readability. The Table thus serves to highlight in the text only the key elements and figures related to the approach (Sect. 4.1.1), the findings (Sect. 4.1.2), and the discussion (Sect. 4.1.3), without overwhelming the reader.